



# Assessing the potential for non-turbulent methane escape from the East Siberian Arctic Shelf

Matteo Puglini[1,2], Victor Brovkin[1], Pierre Regnier[2], and Sandra Arndt[2]

[1]Land in the Earth System, Max Planck Institute for Meteorology, Hamburg, Germany
[2]BGeosys, Department Geoscience, Environment & Society (DGES), Université Libre de Bruxelles, Brussels, Belgium

**Correspondence:** Matteo Puglini (matteo.puglini@mpimet.mpg.de)

**Abstract.** East Siberian Arctic Shelf (ESAS) hosts large, yet poorly quantified reservoirs of subsea permafrost and associated gas hydrates. It has been suggested the global-warming induced thawing and dissociation of these reservoirs is currently releasing methane to the shallow shelf ocean and ultimately the atmosphere. However, the exact contribution of permafrost thaw and methane gas hydrate destabilization to benthic methane efflux from the warming shelf and ultimately methane-
climate feedbacks remains controversial. A major unknown is the fate of permafrost and/or gas hydrate-derived methane as it migrates towards the sediment-water interface. In marine sediments, (an)aerobic oxidation reactions generally act as extremely efficient biofilters that often consume close to 100% of the upward migrating methane. However, it has been shown that a number of environmental conditions can reduce the efficiency of this biofilter, thus allowing methane to escape to the overlying ocean. Here, we used a reaction-transport model to assess the efficiency of the benthic methane filter and, thus, the potential
for permafrost and/or gas hydrate derived methane to escape shelf sediments under a wide range of environmental conditions encountered on East Siberian Arctic Shelf. Results of an extensive sensitivity analysis show that, under steady state conditions, anaerobic oxidation of methane (AOM) acts as an efficient biofilter that prevents the escape of dissolved methane from shelf sediments for a wide range of environmental conditions. Yet, high $CH_4$ escape comparable to fluxes reported from mud-volcanoes is simulated for rapidly accumulating (sedimentation rate $> 0.7$ cm yr$^{-1}$) and/or active (active fluid flow $> 6$ cm
yr$^{-1}$) sediments and can be further enhanced by mid-range organic matter reactivity and/or intense local transport processes, such as bioirrigation. In active settings, high non-turbulent methane escape of up to 19 $\mu$mol$CH_4$ cm$^{-2}$ yr$^{-1}$ can also occur during a transient, multi-decadal period following the sudden onset of $CH_4$ flux triggered by, for instance, permafrost thaw or hydrate destabilization. This "window of opportunity" arises due to the time needed by the microbial community to build up an efficient AOM biofilter. In contrast, seasonal variations in environmental conditions (e.g. bottom water $SO_4^{2-}$, $CH_4$ flux)
exert a negligible effect on $CH_4$ efflux through the Sediment-Water Interface (SWI). Our results indicate that present and future methane efflux from ESAS sediments is mainly supported by methane gas and non-turbulent $CH_4$ efflux from rapidly accumulating and/or active sediments (*e.g.* coastal settings, portions close to river mouths or submarine slumps). In particular active sites on the ESAS may release methane in response to the onset or increase of permafrost thawing or $CH_4$ gas hydrate destabilization rates. Model results also reveal that AOM generally acts as an efficient biofilter for upward migrating $CH_4$
under environmental conditions that are representative for the present-day ESAS with potentially important, yet unquantified implications for the Arctic ocean's alkalinity budget and, thus, $CO_2$ fluxes. The results of the model sensitivity study are





used as a quantitative framework to derive first-order estimates of non-turbulent, benthic methane efflux from the Laptev Sea. We find that, under present day conditions, AOM is an efficient biofilter and non-turbulent methane efflux from Laptev Sea sediments does not exceed $1\,\mathrm{GgCH_4}\,\mathrm{yr}^{-1}$. As a consequence, we state that previously published estimates of fluxes from ESAS water into atmosphere cannot be supported by non-turbulent methane escape from the sediments, but require the build-up and

preferential escape of benthic methane gas from the sediments to the atmosphere that matches or even exceeds such estimated fluxes.

# 1 Introduction

The Siberian Shelf represents the largest shelf on Earth ($\sim 3$ millions km$^2$ Wegner et al. (2015)) and spreads from the Kara Sea to the Laptev, the East Siberian and the Chuckhi Sea. The East Siberian Arctic Shelf (ESAS) corresponds to the broad area

beneath the shallow ($\sim 45$ m water depth, James et al. (2016)) Laptev and East Siberian Arctic Sea (Romanovskii et al., 2004; Shakhova et al., 2010a) and represents the largest region on the Siberian Shelf (Romanovskii et al., 2005), covering about 25% of the total Arctic shelf (Shakhova et al., 2010a).

Although similar in many aspects to other shelf environments, a distinguishing feature of the ESAS is the presence of subsea permafrost and associated gas hydrates buried in the sediment (Sloan Jr and Koh, 2007; Romanovskii et al., 2005). Subsea

permafrost is a terrestrial relict that mainly formed during glacial periods, when Arctic shelves were exposed due to sea level retreating, down to a minimum of 120 m below the current level around the Last Glacial Maximum (Fairbanks, 1989; Bauch et al., 2001). Under these conditions, permafrost aggraded on the shelf and was subsequently submersed when rising sea level flooded the shelf during the Holocene sea transgression (12 and 5 kyr BP). Little is known about he total amount of carbon stored in subsea permafrost, as well as its partitioning between subsea permafrost itself, gas hydrates and free gas. Published

estimates of carbon reservoir sizes diverge by orders of magnitude. For instance Shakhova et al. (2010a) estimate that 1175 PgC are locked in subsea permafrost on the ESAS alone, while McGuire et al. (2009) calculate that, across the entire Arctic shelf, 9.4 PgC reside in upper sediments and 1.5-49 PgC (2-65 PgCH$_4$ ) in methane gas hydrates. Thus, the size of the Arctic subsea permafrost reservoir, its spatial distribution, as well as its biogeochemical and physical characteristics remain poorly known.

These knowledge gaps are critical as climate change is amplified in polar regions. The Arctic is currently warming at a rate twice as fast as the global mean (Trenberth et al., 2007; Bekryaev et al., 2010; Jeffries and Richter-Menge, 2012; Christensen et al., 2013). Recent observations indicate that bottom water temperatures in the coastal and inner shelf regions of the ESAS (water depth < 30 m, Dmitrenko et al. (2011)) are rising, while the central shelf sea may be subject to intense episodic warming (Janout et al., 2016). The increasing influx of warmer Atlantic water into the Arctic Ocean - the so-called

Atlantification - will not only further enhance this warming, but will also influence circulation and salinity patterns on the shelf (Zhang et al., 1998; Biastoch et al., 2011). At the same time, it has been long recognized that the Arctic is a potential hotspot for methane emissions. Extensive methane gas bubbling has been observed in the Laptev Sea and has been directly linked to these environmental changes (Shakhova et al., 2010b, 2014). Shakhova et al. (2014) suggest that warming induced subsea permafrost





thaw and hydrate destabilization may support methane emissions of up to 17 TgCH$_4$ yr$^{-1}$ from the ESAS alone. Projected climate change will further destabilize Arctic subsea permafrost and gas hydrate reservoirs and might thus enhance further methane emissions (Piechura and Walczowski, 1995; Westbrook et al., 2009; Reagan and Moridis, 2009; Biastoch et al., 2011; Hunter et al., 2013; Drake et al., 2015; Ruppel and Kessler, 2017). However, a number of recent studies have questioned the

significance of subsea permafrost thaw and hydrate destabilization for methane efflux from Arctic sediment (Thornton et al., 2016; Ruppel and Kessler, 2017), for methane concentrations in Arctic Ocean waters (Overduin et al., 2015; Sapart et al., 2017) and, ultimately, for methane emissions from the Arctic waters (Ruppel and Kessler, 2017; Sparrow et al., 2018). Thus, the contribution of subsea permafrost thaw and gas hydrate destabilization to methane emissions from the warming Arctic shelf and, ultimately, methane-climate feedbacks remains poorly quantified (James et al., 2016; Saunois et al., 2016). As a

consequence, it has not received much attention in the recent IPCC special report (Masson-Delmotte et al., 2018). At present, a major unknown is the strength of methane sinks in Arctic sediments and waters and their influence on methane emissions (Ruppel and Kessler, 2017). Therefore, improved assessments of the present and future climate impact of permafrost thaw and hydrate destabilization require not only a better knowledge Arctic subsea permafrost and hydrates distribution, reservoir size and characteristics, but also a better quantitative understanding of Arctic methane sinks.

In marine sediments, upward migrating methane is generally efficiently consumed by the anaerobic oxidation of methane (AOM) and, to a lesser extend, the aerobic oxidation of methane (AeOM) (Hinrichs and Boetius, 2002; Reeburgh, 2007; Knittel and Boetius, 2009). Although the exact AOM process has not been fully understood yet (James et al., 2016; McGlynn et al., 2015; Milucka et al., 2012; Wegener et al., 2015; Dean et al., 2018), it is thought that AOM is mediated by methane oxidizing archea that use water (or bicarbonate) as electron acceptor (Hinrichs and Boetius, 2002; Dale et al., 2006):

$$CH_4 + 3H_2O \rightarrow 4H_2 + HCO_3^- + H^+ \tag{1}$$

The electrons are then shuttled (Krüger et al., 2003; Hinrichs and Boetius, 2002), via H$_2$, to sulfate reducing bacteria (eq. (2))

$$SO_4^{2-} + 4H_2 + H^+ \rightarrow HS^- + 4H_2O \tag{2}$$

the overall reaction being

$$CH_4 + SO_4^{2-} \rightarrow HCO_3^- + HS^- + H_2O. \tag{3}$$

The first catabolic step is thermodynamically favourable only under a limited range of environmental conditions, while the second step is subject to weaker thermodynamic constraints (LaRowe et al., 2008). A recent assessment indicates that, in global sediments, around 45-61 TgCH$_4$ yr$^{-1}$ (Egger et al., 2018) are consumed by AOM, thus significantly reducing previously published estimates of 320-360 PgCH$_4$ yr$^{-1}$ (Hinrichs and Boetius, 2002; Reeburgh, 2007).

    AOM generally acts as a particularly efficient biofilter for upward migrating methane and oxidizes up to 100% of the

methane flux coming from below (*e.g.* Regnier et al. (2011)). However, a number of environmental conditions can reduce the efficiency of this AOM biofilter, allowing methane to escape from the sediment (Iversen and Jorgensen, 1985; Piker et al., 1998; Jørgensen et al., 2001; Treude et al., 2005; Knab et al., 2008; Dale et al., 2008c; Thang et al., 2013; Egger et al., 2016).





It has been shown that, in particular, high sedimentation rates (Egger et al., 2016), slow microbial growth (Dale et al., 2006, 2008c) or the accumulation of free gas can promote methane efflux from the sediment. These findings are particularly relevant for potential methane escape from Arctic shelf sediments. The Siberian shelf is the largest sedimentary basin in the world (Gramberg et al., 1983) and shelf areas close to the large Arctic rivers reveal sedimentation rates than can be up to 5 times

faster than rates that are typically observed in the ocean (Leifer et al., 2017). In addition, the Arctic shelf is subject to large seasonal, as well as climate-induced longterm, changes in environmental conditions that may influence the efficiency of the AOM biofilter through their effect on microbial biomass dynamics. Finally, observations from the ESAS also indicate that methane gas accumulates in the sediments. When free gas pockets grow enough, methane tends to migrate upwards along pathways with higher permeability or where fractures occur (Yakushev, 1989; Boudreau et al., 2005; Wright et al., 2008;

Shakhova et al., 2014, 2015, 2017; Leifer et al., 2017) and might even crack the sediments themselves (O'Connor et al., 2010; Overduin et al., 2016). However, despite a wealth of AOM-related research, a holistic, quantitative evaluation of the most important environmental controls on the efficiency of the AOM biofilter and its impact on methane escape from marine sediments is currently lacking. Thus our ability to understand and quantify AOM sink in ESAS sediments and thus the climate impact of subsea permafrost thaw and gas hydrate destabilization is seriously compromised.

Therefore, we here use a reaction-transport model approach to understand and quantify the efficiency of the AOM biofilter and its influence on the potential release of methane from ESAS sediments that bear thawing permafrost and/or dissociating methane gas hydrates. The developed model accounts for the most pertinent primary and secondary redox processes, as well as mineral precipitation, methane gas formation and fast equilibrium reactions that affect biogeochemical dynamics in both passive, as well as active sediments influenced by a deep methane source. We limit our model analysis to non-turbulent methane

efflux, because methane in gaseous form is not directly accessible for the AOM community. As a consequence, free gas bubbles are less prone to be consumed by AOM and methane gas either sits in the sediments or rapidly migrates upcore through cracks, faults or fractures (Boudreau, 2012), bypassing the AOM biofilter.

The model is forced with boundary conditions that are broadly representative of conditions encountered on ESAS. It is applied to conduct a comprehensive one-at-a-time, steady-state sensitivity study over the entire plausible range of 1) sedi-

mentation rates, 2) active fluid flow velocities, 3) AOM rate constants, 4) organic matter reactivity and 5) non-local transport activity encountered on the ESAS. In addition, we also evaluated the influence of environmental change induced by 1) seasonal variability and 2) idealized climate change on the efficiency of the AOM-biofilter and non-turbulent methane escape at the seafloor under transient conditions. For this purpose, the model is extended by adopting an explicit description of AOM biomass dynamics and a bioenergetic rate law for AOM (Dale et al., 2006, 2008c, b).

The specific aims of this work are to 1) identify and quantitatively understand the most important environmental controls on the efficiency of the AOM biofilter as well as 2) its significance for non-turbulent methane escape from marine sediments on the ESAS under present-day environmental conditions and in response to idealized environmental variability. Model results will then be used to 3) quantitatively assess the potential for non-turbulent $CH_4$ escape from ESAS sediments and set some constraints on the Arctic methane budget.





## 2 Methods

### 2.1 BRNS: Reaction-transport model

The Biogeochemical Reaction Network Simulator (BRNS) (Regnier et al., 2002; Aguilera et al., 2005; Centler et al., 2010) - an adaptive simulation environment suitable for simulating large, mixed kinetic-equilibrium reaction networks in porous media (*e.g.* Jourabchi et al. (2005); Thullner et al. (2005); Dale et al. (2009)) - is used to quantitatively explore the fluxes and transformations of methane in ESAS sediments. For this purpose, we set-up a reaction network (table S1, S2), model parameters (table S5), as well as boundary conditions (table S6) that are broadly representative for conditions encountered on the present-day Siberian shelf.

In the BRNS, the general mass conservation for each solid and dissolved species is described by a a set of coupled advection-diffusion-reaction equations in porous media which are solved simultaneously (*e.g.* Berner (1980); Boudreau (1997); note that dependencies on $z$ and $t$ have been omitted for simplicity):

$$\frac{\partial \xi C_i}{\partial t} = \frac{\partial}{\partial z}\left[(D_i + D_{b,i})\xi\frac{\partial C_i}{\partial z}\right] - \frac{\partial}{\partial z}(v\xi C_i) + \alpha_i \xi(C_i(0) - C_i) + \mathscr{S}_i. \tag{4}$$

$C_i$ is the concentration of the species $i$ (mass per porewater volume for dissolved species or mass per solid matrix volume for a solid species); $\xi$ *i.e.* the porosity $\xi = \varphi$ for dissolved species and $\xi = \varphi_s = 1-\varphi$ for solid species. $D_i$ is the effective diffusion coefficient for species $i$ and is affected by salinity, temperature and tortuosity (see Table S5). $D_b$ denotes the bioturbation coefficient and $v$ is the advective velocity. For solid species $v = \omega$ with $\omega$ being the burial rate, while the advective velocity for dissolved species is given by the sum of the burial rate and an advective flow velocity, $v_{up}$, *i.e.* $v = \omega + v_{up}$. A site where $v_{up} \neq 0$ is defined as an *active site*, while a site with no advective upward water flow is defined as *passive*. $\alpha_i$ is the bioirrigation coefficient ($\alpha_i = 0$ for solid species) and $C_i(0,t)$ is the concentration of the species $i$ at the Sediment-Water Interface (SWI). The reaction term $\mathscr{S}_i$ is written as:

$$\mathscr{S}_i = \sum_j \lambda_{ij} R_j \tag{5}$$

where $\lambda_{ij}$ are the stoichiometric coefficients of all reaction rates $R_j$ that affect species $i$.

### 2.1.1 Transport

The effective diffusion coefficients $D_i$ are determined by correcting the diffusion coefficients in free solution $D_i^0$ (Boudreau, 1997) for tortuosity $\theta$ and temperature. Tortuosity is calculated by means of porosity $\varphi$ according to a modified Weissberg relation (Boudreau, 1997): $\theta = 1 - \ln(\varphi^2)$. Note that the effective diffusion coefficients used in the model neglect pressure effects. Following Dale et al. (2008a), migration of methane gas is simply parameterized via a pseudo-diffusive term, with an apparent gas diffusion coefficient, $D_{CH_4}(g)$. Bioturbation in the upper decimeters of the sediment is simulated using a diffusive term (*e.g.*, Boudreau (1986)), with a constant bioturbation coefficient, $D_b^0$. The model assumes that bioturbation ceases at the





bioturbation depth, $z_{bio}$ (Boudreau, 1997). Bioirrigation is included in the mass conservation equation as a source or a sink function analogous to a kinetic rate. It is calculated as the product of the irrigation intensity, $\alpha$ ($\alpha = 0$ for all solids), and the difference in concentration of species $i$ relative to the concentration at the SWI, $C_i(0)$. The bioirrigation rate $\alpha$, is evaluated from the bioirrigation coefficient at the sediment surface ($\alpha_0$) and the bioirrigation attenuation depth ($z_{irr}$) and is given by eq.

S9. Porosity is assumed to decrease with depth according to an exponential decay (Athy, 1930):

$$\varphi(z) = \varphi_0 e^{-c_0 z} \tag{6}$$

with $\varphi_0$: porosity at the Sediment-Water Interface (SWI) and $c_0$: typical length scale for compaction. Table S5 provides a detailed overview of the transport parameter values applied in the model.

### 2.1.2   Biogeochemical network

The reaction network implemented here (33 species, 37 reactions) encompasses the most pertinent primary and secondary redox reactions, equilibrium reactions and mineral precipitation and adsorption reactions. A summary of the reactions, their stoichiometry and their rate formulations can be found in Table S2 and Table S3. The following section provides a short description of the implemented reaction network, as well as a more detailed description of the reactions that affect the production/consumption of methane. A complete description can be found in the supplementary information.

The BRNS model accounts for the degradation of organic matter by aerobic degradation, denitrification, manganese oxide reduction, iron reduction, sulfate reduction and methanogenesis (Table S2). Organic matter degradation is described by means of the reactive continuum model (RCM) (Aris, 1968; Ho and Aris, 1987; Boudreau and Ruddick, 1991) that describes compound-specific reactivities and, thus, captures the widely observed decrease in apparent organic matter reactivity with degradation state. The relative importance of each metabolic pathway is simulated through a series of kinetic limitation terms,

reflecting their sequential utilization in the order of their decreasing Gibbs energy yields (Table S1). After all terminal electron acceptors (TEAs) are consumed, the remaining organic matter may be degraded by methanogenesis. The rates of secondary redox reactions (Table S3), are described by bimolecular rate laws (*e.g.* Wang and Van Cappellen (1996)). Adsorption reactions are considered as fast equilibrium processes (Table S3, R28-R30). Mineral precipitation rates are simulated according to kinetic-thermodynamic rate laws (Table S3, R16-R24).

As described above, methane is produced during organic matter degradation by methanogens in deeper sediment layers, once all TEAs are depleted (Table S2, R6). If the concentration of dissolved methane exceeds the saturation concentration $[CH_4]^*$ methane gas forms. The transfer rate of methane between the dissolved and gaseous phase is linearly controlled by the departure of the simulated dissolved methane concentration from the saturation concentration (Haeckel et al., 2004; Hensen and Wallmann, 2005; Tishchenko et al., 2005; Mogollón et al., 2009; Graves et al., 2017). $[CH_4]^*$ is calculated according to

Dale et al. (2008a), derived from the formulation proposed by Duan et al. (1992) for which $[CH_4]^*$ depends on *in situ* salinity, pressure and temperature. Here, we assume that the formed methane gas is inaccessible to microbial activity and hence bypasses anaerobic and/or aerobic oxidation zones. In contrast, dissolved methane can be consumed by anaerobic (AOM) or aerobic oxidation of methane (AeOM). Free gas can re-dissolve into porewater once porewater methane concentration fall





below the saturation level and may then become available to methanotrophs. AeOM rate is simply described by a bimolecular rate law (Table S3, R14). The description of AOM depends on the model scenario. For steady state simulations, we apply a simple bimolecular rate:

$$\text{rate}_{AOM} = k_{AOM}[\text{CH}_4][\text{SO}_4^{2-}]. \tag{7}$$

It is the simplest and most commonly used formulation of the AOM rate in reaction-transport models (*e.g.* Regnier et al. (2011)). It accounts for kinetic controls and assumes that, under steady state conditions, bioenergetic controls are negligible (Dale et al., 2006; Regnier et al., 2011).

   For transient model simulations, we apply a bioenergetic rate law in combination with an explicit description of the AOM-performing biomass (Dale et al., 2006, 2008c). It has been shown that the rates of redox reactions, whose energy yield is

used by micro-organisms to grow, can be coupled to biomass growth rates via a kinetic Monod term and a thermodynamic Boltzmann term (*e.g.* Rittmann and VanBriesen (2019)). Hence, the time derivative of AOM-performing biomass ($B$) can be written as:

$$\frac{dB}{dt} = \mu_g B \cdot F_K \cdot F_T - \mu_d B^2 \tag{8}$$

where $\mu_g$ is the growth rate and $\mu_d$ is the decay rate. $F_K$ is the kinetic constraint given by:

$$F_K = \frac{[\text{CH}_4]}{K_m^{\text{CH}_4} + [\text{CH}_4]} \cdot \frac{[\text{SO}_4^{2-}]}{K_m^{\text{SO}_4^{2-}} + [\text{SO}_4^{2-}]} \tag{9}$$

with $K_m^{\text{SO}_4^{2-}}$ half saturation constant of $\text{SO}_4^{2-}$ and $K_m^{\text{CH}_4}$ half saturation constant of $\text{CH}_4$, according to a typical Michaelis-Menten for enzymatically-catalyzed reactions. $F_T$ represent the thermodynamic limitation and is given by

$$\begin{cases} 1 - \exp\left(\frac{\Delta G_r + \Delta G_{BQ}}{\chi RT}\right), & \text{if } \frac{\Delta G_r + \Delta G_{BQ}}{\chi RT} < 0 \\ 0, & \text{if } \frac{\Delta G_r + \Delta G_{BQ}}{\chi RT} > 0 \end{cases} \tag{10}$$

where R is the gas constant, T is the absolute temperature, $\chi$ is the average number of electrons transferred per reaction per

mole of ATP produced (Jin and Bethke, 2005), $\Delta G_r$ is the Gibbs free energy of the reaction and $\Delta G_{BQ} = 20\,\text{kJ}\,(\text{mol e}^-)^{-1}$ is the minimum energy needed to support synthesis of $\sim \frac{1}{3} - \frac{1}{4}$ mol ATP (Dale et al., 2008c). In order to be thermodynamically favorable the total energy $\Delta G_r + \Delta G_{BQ}$ has to be negative, meaning the that Gibbs free energy provided by the catabolic reaction is sufficient to sustain the microbial biomass growth. $\Delta G_r$ is given by

$$\Delta G_r = \Delta G_r^0 + RT \ln\left(\gamma \frac{[\text{HS}^-] \cdot [\text{HCO}_3^-]}{[\text{CH}_4] \cdot [\text{SO}_4^{2-}]}\right) \tag{11}$$

with $\Delta G_r^0$: standard free energy of the reaction, the second term: deviations from standard conditions (temperature and reaction quotient) on Gibbs free energy and $\gamma$: a parameter representing departure from ideal beahviour.

   The link between substrate consumption and microbial growth (anabolism) is given by Dale et al. (2006):





$$13.8\mathrm{SD} \cdot \mathrm{SO}_4^{2-} + 14.3\mathrm{SD} \cdot \mathrm{CH}_4 + 0.2\mathrm{SD} \cdot \mathrm{NH}_4^+ + 0.3\mathrm{SD} \cdot \mathrm{H}^+ \rightarrow 0.2\mathrm{B} + 13.3\mathrm{SD} \cdot \mathrm{HCO}_3^- + 13.8\mathrm{SD} \cdot \mathrm{HS}^- \quad (12)$$

Assuming that the cellular composition of the biomass $B$ is equal to $\mathrm{C}_5\mathrm{H}_7\mathrm{O}_2\mathrm{N}$ (Bruce and Perry, 2001; Dale et al., 2006, 2008c; Rittmann and McCarty, 2012). $SD = (1 - \varphi)/\varphi$ is the conversion factor between dissolved and solid species, here represented by microorganisms (which are assumed to be attached to the solid matrix). Catabolism is linked to biomass growth

(anabolism) through the growth yield. We apply a yield of 0.0713 (Dale et al., 2006), which falls at the upper end of reported AOM growth yields, *i.e.* $0.05 - 0.07$ (Dale et al., 2006; Nauhaus et al., 2007).

### 2.1.3 Boundary conditions

Boundary conditions place the model in its environmental context. For dissolved species, constant bottom water concentrations (Dirichlet boundary conditions) are applied at the sediment-water interface, while a known flux condition (Neumann bound-

ary condition) are applied for solid species. At the lower boundary, a zero gradient flux boundary condition ($\partial C/\partial z = 0$) is considered for all species except methane, for which a Dirichlet condition is specified to account for methane supplied from thawing permafrost and/or dissociating gas hydrates below.

## 2.2 Model evaluation

In order to evaluate the performance of the BRNS in capturing the main diagenetic patterns observed in Arctic marine sediments

we perform one steady state model case studies for an Arctic sites: a cold seep site off Vesterålen, Norway (68.9179°N, 14.2858°E, 222 m water depth; Sauer et al. (2015, 2016))

Even though it is not located on the ESAS, the core offshore Norway (GC-51) is chosen because it was retrieved in the Hola trough, on the continental shelf of Vesterålen, and is thus representative for the type of shelf sediments considered in our study. In addition, porewater data reveals a well-developed Sulfate-Methane Transition Zone (SMTZ). The site has already

been subject of a modeling analysis by Sauer et al 2016, hence offering a benchmark for our simulation results. The Vesterålen site shows no sign of active water flow and, thus represents a passive setting ($v_{up} = 0$ cm yr$^{-1}$). Upper boundary conditions and model parameters are constrained on the basis of observations reported by Sauer et al. (2016) (Table S4). In addition, we impose the TOC depth-profile reported in Sauer et al. (2015) and evaluate the age of the organic matter using the sedimentology reported in Sauer et al. (2016).

When evaluating model performance, particular attention is given to sulfate, methane and ammonium ($\mathrm{NH}_4^+$) depth profiles. While the former two species are of main interest for evaluating simulated AOM dynamics, $\mathrm{NH}_4^+$ is a good indicator for OM degradation since it is produced by the degradation of organic matter (see Table S2) and is only affected by nitrification (R7) and adsorption (R28). The latter, although important, acts homogeneously throughout the sediments (considering the slight variation in sediment porosity, LaRowe et al. (2017)). It can thus only cause uniform shifts in $[\mathrm{NH}_4^+]$ profile, but does not

affect the overall shape of the $\mathrm{NH}_4^+$ depth profile. OM reactivity parameters are varied to find the best fit between observed and simulated OM, $[\mathrm{NH}_4^+]$ and $[\mathrm{O}_2]$ profiles.





## 2.3 Modeling strategy

### 2.3.1 Steady state sensitivity analysis:

To evaluate the main physical and biogeochemical controls on the efficiency of the AOM biofilter and non-turbulent methane emission from ESAS sediments, we conduct a comprehensive steady state sensitivity study. For this purpose, we design a set

of two baseline scenarios:

1. a passive case, *i.e.* $v_{up} = 0$ cm yr$^{-1}$;

2. an active case, *i.e.* with $v_{up} = 1$ cm yr$^{-1}$, a value which falls within the range of fluid flow velocities $v_{up} = 0.005 - 30$ cm yr$^{-1}$ observed across a wide range of different active environments (Regnier et al., 2011).

For both baseline scenarios, we assume a water depth of 30 m, which is similar to the average water depth of the ESAS $\sim$45

m (James et al., 2016). Temperature is set equal to 0°C, and thus similar to the yearly average of $-0.79$°C observed in the Laptev Sea at a depth of about 30 m (Dmitrenko et al., 2011). The bioturbation coefficients $D_b^0$ and bioirrigation coefficients $\alpha_0$ (Thullner et al., 2009) are then derived from global empirical relationships according to Middelburg et al. (1997) and Thullner et al. (2009), respectively. The methane saturation concentration $[CH_4]^*$ is calculated on the basis of the relationship proposed by Dale et al. (2008a) assuming a soil matrix density of 2.8 g cm$^{-3}$. Values of $\varphi_0$ and $c_0$ (see eq. 6) are determined based

on LaRowe et al. (2017). Boundary conditions are reported in Table S6 and informed by observations. They are chosen to be broadly representative of the wider Siberian shelf environment. Each sensitivity study run is forced with a range of different dissolved $[CH_4]$ concentrations at the lower model boundary, mimicking different methane fluxes from thawing permafrost at depth. The applied set of methane concentrations at the lower boundary range from zero to the methane gas saturation concentration $[CH_4]_- = 0 - 20 - 100 - 330 - 1169 - 5455$ $\mu$M and also include the highest methane concentration in ESAS

cores observed by Overduin et al. (2015) ($[CH_4]_- = 1.169$ mM).

Table 1 and Table S5 summarize the parameters applied in the baseline simulation and Table S6 provides an overview of the applied upper boundary conditions.

**Table 1.** Model parameters changed in the "one-at-time" sensitivity studies. Reported values are for the baseline simulations.

| Quantity | Meaning | Value | Units | Reference |
|---|---|---|---|---|
| $\omega$ | Sedimentation rate | 0.123 | cm yr$^{-1}$ | Burwicz et al. (2011) |
| **a** | Average lifetime of reactive OM | 10 | yr | This study |
| $\mathbf{v_{up}}$ | Upward water velocity | 0, 1 | cm yr$^{-1}$ | This study |
| $\boldsymbol{\alpha_0}$ | Bioirrigation coefficient | 99.5 | yr$^{-1}$ | Thullner et al. (2009) |
| $\mathbf{k_{AOM}}$ | AOM rate constant | $5.0 \cdot 10^3$ | M$^{-1}$ yr$^{-1}$ | Regnier et al. (2011) |
| $[CH_4]_-$ | CH$_4$ lower boundary condition | $0 - 5.455$ | mM | This study |



A set of five "one-at-time" parameter variation experiments, encompassing the most important controls on benthic methane cycling (Regnier et al., 2011; Meister et al., 2013; Egger et al., 2018) is performed for both the passive as well as active baseline scenario:

1. Sedimentation rate $\omega$. The sedimentation rate is varied over two orders of magnitude ($0.03 - 0.123 - 0.17 - 1.5$ cm yr$^{-1}$). Maximum values are comparable to terrestrial sediment accumulation rates in the Lena river delta (Bolshiyanov et al., 2015), fast marine sedimentation rates during the early Holocene sea transgression (Bauch et al., 2001) and marine accumulation on subsea permafrost deposit in Buor Khaya Bay ($\sim 1.1$ cm yr$^{-1}$, inferred from Overduin et al. (2015)), while minimum values are representative of sedimentation rates found in the East Siberian Arctic Sea (Stein et al. (2001) in Levitan and Lavrushin (2009)). The baseline value of $\omega$ is calculated based on the empirical global relationship proposed by Burwicz et al. (2011).

2. Active fluid flow $v_{up}$. Buoyancy-induced motion (Baker and Osterkamp, 1988), water streams channeled through fault lines or groundwater discharge (Charkin et al., 2017) can cause active fluid flow in Arctic shelf sediments underlain by subsea permafrost or gas hydrates (Judd and Hovland, 2009; Semenov et al., 2019). Therefore, $v_{up}$ is varied from $0 - 0.3 - 0.5 - 1 - 3 - 7 - 10$ cm yr$^{-1}$. This interval falls in the range of reported upward advective water velocities in marine sediments $0.005 - 30$ cm yr$^{-1}$ (Regnier et al., 2011).

3. AOM constant $k_{AOM}$. Rate constants implicitly account for factors that are not explicitly described in the model and thus tend to show a strong variability between sites. A comprehensive compilation of published model AOM rate constants (Regnier et al., 2011) reveals a variability of over 6 order of magnitudes ($10 - 10^7$ M$^{-1}$ yr$^{-1}$). The AOM rate constant $k_{AOM}$ (eq. 7) is thus varied over the range $k_{AOM} = 5 \cdot 10^2 - 5 \cdot 10^3 - 5 \cdot 10^4 - 5 \cdot 10^5 - 5 \cdot 10^6 - 5 \cdot 10^7$ M$^{-1}$ yr$^{-1}$.

4. Organic matter reactivity (i.e. RCM parameter). Although the apparent OM reactivity is controlled by a combination of two parameters ($a$ and $\nu$), previous studies indicate a less pronounced variability in $\nu$ (Arndt et al., 2013; Sales de Freitas, 2018), as well as a strong control of $a$ on the SMTZ depth (Regnier et al., 2011; Meister et al., 2013). Thus, $\nu$ was kept constant, while $a$ was varied over the entire range of previously published values $a = 0.1 - 1 - 10 - 100 - 500 - 1000$ yr (Arndt et al., 2013).

5. Bioirrigation coefficient $\alpha_0$. Bioirrigation activity remains largely unconstrained on the Siberian shelf due to the scarcity of observational data (Teal et al., 2008). However, environmental stressors, such as ice scouring (*e.g.* Shakhova et al. (2017) and references therein) and trawling are detrimental to the local fauna, thus suggesting a low bioirrigation intensity. Yet, observations from other polar sites indicate that although biological diversity and activity is often low, it might be locally enhanced (Clough et al., 1997). In addition, ice scouring might also enhance non-local transport seasonally. We therefore, varied $\alpha_0$ over the entire range of plausible values : $0 - 33 - 66 - 99.5 - 120 - 240$ yr$^{-1}$ (Thullner et al., 2009).





### 2.3.2 Transient Sensitivity Study

Dale et al. (2008c) showed that temporally varying environmental conditions may reduce the efficiency of the benthic AOM filter and facilitate methane escape due to the delayed response of the microbial community to changing conditions. Therefore, we also perform a series of transient simulations with a bioenergetic rate law for AOM (eq. 8) and an explicit description of
AOM biomass to explore the impacts of seasonal and climate change driven environmental activity on methane escape from the ESAS. Simulation results from the passive steady state baseline run with $[CH_4]_- = 0$ mM are used as initial conditions for the transient experiments. Four different transient environmental perturbation scenarios that reflect seasonal (1, 2), as well as idealized future (3, 4) environmental variability on the ESAS are run with three different values of $v_{up}=0-1-5$ cm yr$^{-1}$ over a period of 200 years:

1. *Seasonal* $CH_4$: seasonal change of methane supply from permafrost thaw and/or hydrate destabilization. $CH_4$ concentration at the bottom of the sediment column: null for 6 months, then increasing up to a peak of $[CH_4]_-$ ($20-100-330-1169-5455$ $\mu$M) for the remaining 6 months of the year and again back to null concentration.

2. *Seasonal* $CH_4 + SO_4^{2-}$: seasonal freshening of waters due to riverine discharge and sea ice melt. During winter, higher bottom salinity (Dmitrenko et al., 2011) results in higher sulfate concentration (Dickson and Goyet, 1994), while lower
salinities and thus sulphate concentrations characterise the melt season. The bottom boundary condition for methane $[CH_4]_-$ follows an opposite trend: it is set to zero during the winter months and increases in Arctic summer.

3. *Linear* $CH_4$: slow increase in methane supply from permafrost thaw and/or hydrate destabilization. A linear increase of the bottom boundary methane concentration $[CH_4]_-$ (from 0 up to the peak) over 200 years is applied.

4. *Sudden* $CH_4$: abrupt increase of methane supply from permafrost thaw and/or hydrate destabilization. An instantaneous
change of bottom boundary methane concentration - from 0 to one of the peak value $[CH_4]_-$ - is applied.

### 2.3.3 Analyzed output

For each simulation we evaluate the effect of the respective parameter change on:

1. the non-turbulent (i.e. not-ebullition driven) flux of methane from the sediments into the water column;

2. the depth of the SMTZ;

3. the efficiency ($\eta$) of the AOM biofilter (see Appendix A for the exact definition of AOM applied here).

In addition, fluxes of $SO_4^{2-}$ and $CH_4$ at the SMTZ, the maximum and integrated AOM rate and the Damköhler number ($D_a$) for AOM and methanogenesis are also calculated. Damköhler number is defined as eq. B4 (see Appendix B) and sets the ratio between the typical transport time-scale and the typical reaction time-scale. If $D_a < 1$, the reaction time-scale is longer than transport time-scale (i.e. the reaction is slower) and the process is reaction-limited. If $D_a > 1$ the process is transport-limited.
Finally, for transient simulations, the integrated AOM-perfoming biomass ($\Sigma B$) was also analyzed.





# 3 Results and discussion

## 3.1 Case studies

### 3.1.1 Case study: Cold seep off Vesterålen, Norway

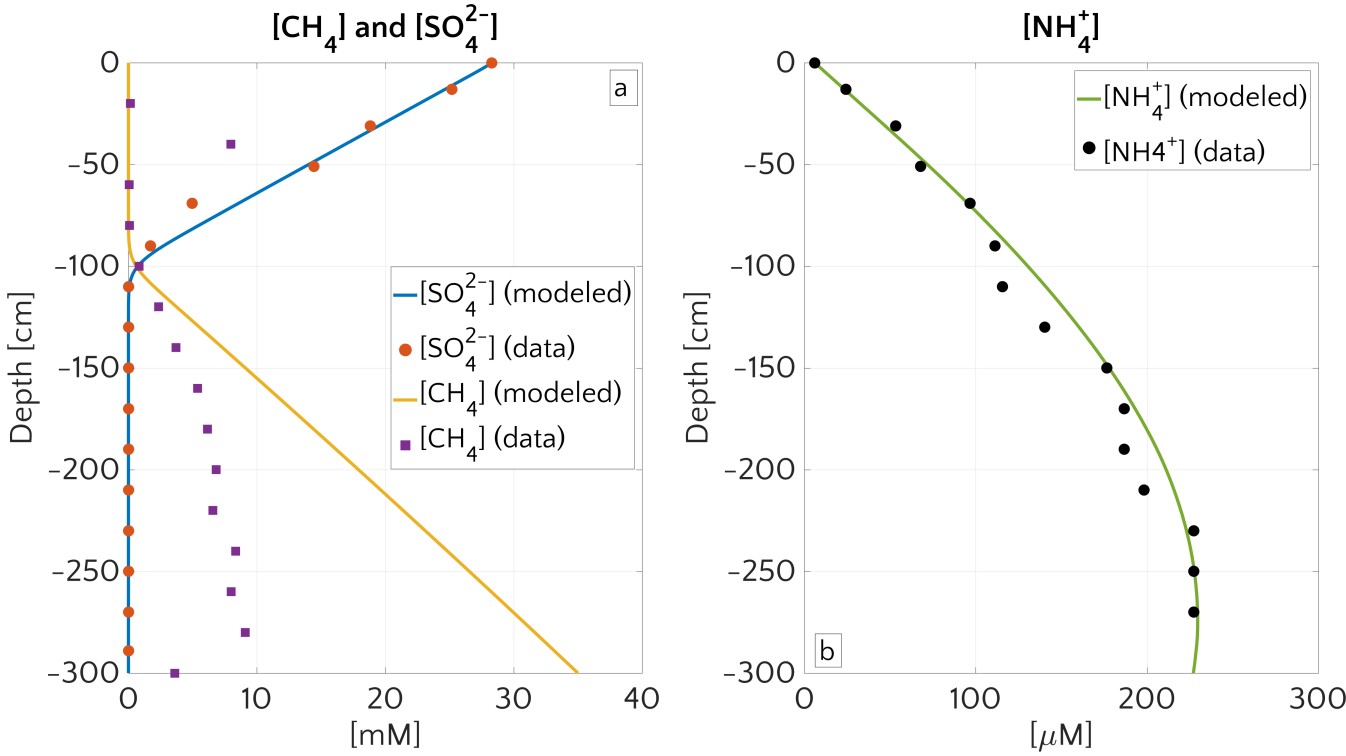

**Figure 1.** Pore water concentration profiles for $CH_4$, $SO_4^{2-}$ (*a*) and $NH_4^+$ (*b*) at site GC-51 of Hola trough. Dots represents the measurements and continuous lines the simulated results. The boundary conditions employed in the model are reported in table S4.

Fig. 1 compares simulated and observed depth profiles for site GC-51. Simulation results show an overall satisfactory
5   agreement with measurements. The general shape of the downward diffusing $SO_4^{2-}$ and upward diffusing $CH_4$ depth profiles
is similar to the profiles that are typically observed in passive sediments. In addition, the model reproduces the observed SMTZ
depth, located at about 100 cm. Above the SMTZ, the simulated $CH_4$ concentrations closely agree with measurements, but
simulated and observed depth profiles diverge significantly below the SMTZ. Such a discrepancy is common (*e.g.* Dale et al.
(2008a); Sauer et al. (2016)) and likely results from degassing during core extraction and recovery (Dickens et al., 2003). Yet,
10   the simulated $CH_4$ concentration close to the lower model boundary (35 mM) is consistent with the values reported in Sauer
et al. (2016) (30 mM) and is lower than the in-situ methane saturation concentration at that depth (39 mM).



Furthermore, the observed $NH_4^+$ profile is also well reproduced, suggesting that the model captures OM degradation dynamics well. Model derived organic matter degradation rate parameters of $a = 1100$ yr, $\nu = 0.100$ indicate a generally low reactivity of OM depositing at this site, which is in agreement with observations and low $NH_4^+$ concentrations.

## 3.2  Main physical and biogeochemical controls on potential non-turbulent methane flux from ESAS sediments

### 3.2.1  General patterns of methane and sulfate cycling on the ESAS

The comprehensive ensemble of all sensitivity experiments allows exploring the general patterns of methane and sulfate cycling under a range of environmental conditions that is broadly representative for conditions encountered on the ESAS (Fig. 2). Model results confirm that AOM is an efficient sink for the diffusive $CH_4$ supply from below. For most of the investigated environmental conditions (95% of the runs), 95-99.9% of the upward diffusing $CH_4$ is consumed within the SMTZ, resulting in very small or negligible methane effluxes ($\leq 10^{-2}$ $\mu molCH_4$ cm$^{-2}$ yr$^{-1}$) from the sediment. If upscaled to the total area of the ESAS ($\sim 1.485 \cdot 10^6$ km$^2$, Wegener et al. (2015)), for which methane outgassing estimates have been published, the smallest simulated non-turbulent methane flux (*i.e.* $1.4 \cdot 10^{-13}$ $\mu mol$ cm$^{-2}$ yr$^{-1}$, Fig. 2.*b*) would sum up to a total flux of 2.1 mmolCH$_4$ yr$^{-1}$, resulting in a negligible role of non-turbulent, benthic methane fluxes to the Arctic methane budget.

Yet, model results also show that, under a specific set of environmental conditions that lower the efficiency of the AOM biofilter (see detailed discussion below), non-turbulent $CH_4$ escape from ESAS sediments can reach values of up to 27 $\mu molCH_4$ cm$^{-2}$ yr$^{-1}$. Simulation results show that these high effluxes and, thus, low AOM biofilter efficiencies are generally simulated for environmental conditions that cause a shallow location of the SMTZ ($< 18$ cm) and that they are very sensitive to changes in environmental conditions that would cause a deepening of the SMTZ. For instance, a deepening of the SMTZ from 18 to 26 cm results in a rapid increase in AOM efficiency from 1% to 98% (Fig. 2.*a*). Furthermore, results indicate that, for SMTZ depths larger than 26 cm, AOM remains an efficient barrier across the full spectrum of investigated environmental conditions (Fig. 2). The observed link between AOM filter efficiency and SMTZ is reflected in the strong (semilog) linear relationship between methane flux at the SWI and the SMTZ depth (Fig.2.*b*). Such a relationship reveals the pivotal connections between these two quantities and mirrors the empirically found linear log-log relationship between measured $CH_4$ fluxes at the SMTZ and the SMTZ depths (Fig. S4) by Egger et al. (2018). Maximum simulated $CH_4$ effluxes are thus comparable in magnitude to fluxes reported from mud-volcanos, *e.g.* in the Gulf of Cadiz 2.1-40.7 $\mu molCH_4$ cm$^{-2}$ yr$^{-1}$ (Niemann et al., 2006a) or Mosby mud-volcano in the Barents Sea 0.03 $\mu molCH_4$ cm$^{-2}$ yr$^{-1}$ (Niemann et al., 2006b); other coastal settings (Dutch coastal reservoir 20-80 $\mu molCH_4$ cm$^{-2}$ yr$^{-1}$, Egger et al. (2016)) or tidal flats (4-800 $\mu molCH_4$ cm$^{-2}$ yr$^{-1}$,Dale et al. (2008b)). Upscaling the highest simulated non-turbulent flux to the ESAS results in a total efflux of $0.408$ TmolCH$_4$ yr$^{-1} = 6.52$ TgCH$_4$ yr$^{-1}$ a value that equals $\sim 10\%$ of global marine seepage at seabed level (Saunois et al., 2016) and similar in magnitude to the global methane efflux that has been estimated for upper continental slope sediments on a centennial timescale (4.73 TgCH$_4$ yr$^{-1}$, Kretschmer et al. (2015)).

Further insights into the general drivers that control methane dynamics in ESAS sediments are provided by Damköhler numbers. Damköhler numbers for simulated methanogenesis ($D_{a_{MG}}$) and AOM ($D_{a_{AOM}}$) are reported in Fig. S2. $D_{a_{MG}}$





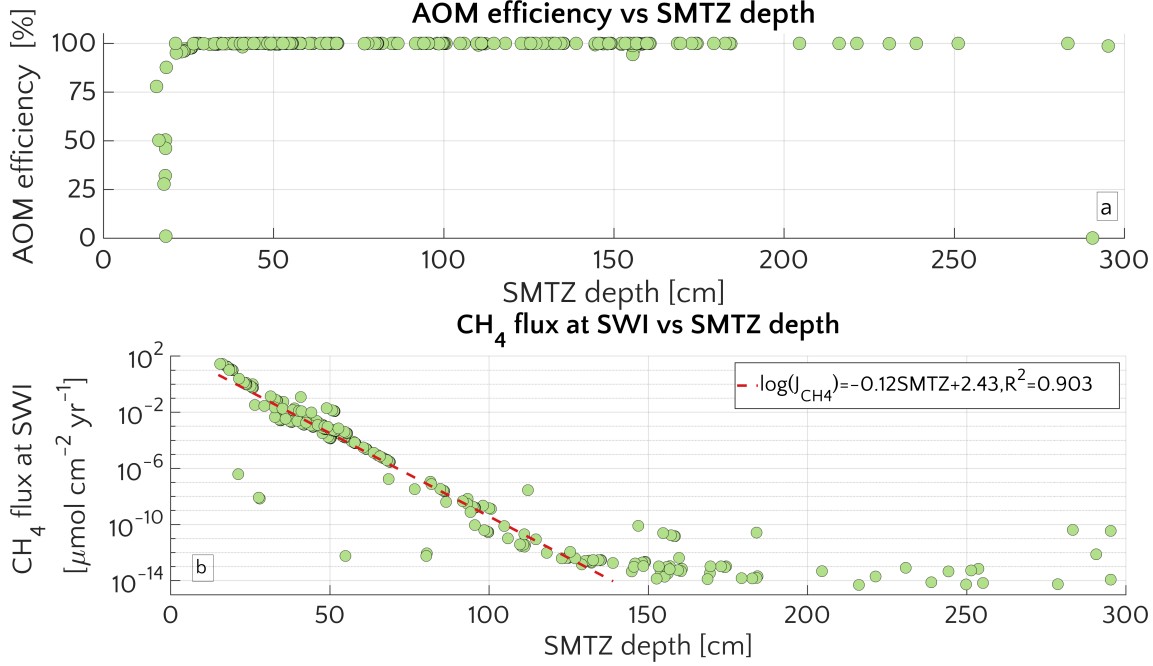

**Figure 2.** Aggregation of all the simulation performed for the "one-at-time" sensitivity study. *a*. AOM efficiency versus the depth of the SMTZ. *b*. Scatter plot and semi-log fit of the methane flux ($J_{CH4}$) at the SWI versus SMTZ depth.

(purple circles) are $< 1$ , span a range of $\sim 0.0021 - 0.43$ and are thus comparable to previously reported $D_{a_{MG}}$ of 0.22 for methane gas hydrate bearing sites, such as Hydrate Ridge and Kithley Canyon (Chatterjee et al., 2011). They reveal that methanogenesis is always slower than methane transport and that $CH_4$ dynamics driven by methanogenesis are thus reaction-limited. This result is consistent with the fact that methanogenesis rates are merely supported by the slow influx and transport

5 of OM by burial and bioturbation.

In contrast, high $D_{a_{AOM}}$ values ($D_{a_{AOM}}$ =32-2.78 $\cdot 10^5$ - Fig. S2, orange circles), show that AOM is transport-limited, suggesting a sensitive role of transport parameters in determining AOM efficiency and in controlling methane flux across the SMTZ and subsequently the SWI.

### 3.2.2 Environmental controls and mechanisms of methane escape from ESAS sediments

10 The simulated general patterns of methane and sulfate cycling on the ESAS thus broadly corroborate previous findings regarding the dominant environmental controls on AOM biofilter efficiency and SMTZ depth (Regnier et al., 2011; Egger et al., 2018; Meister et al., 2013; Winkel et al., 2018). Yet, they also challenge traditional views on the factors that favour high $CH_4$ escape through the SWI. In particular, they highlight the essential link between AOM efficiency and SMTZ depth, and as a consequence the central importance of environmental conditions that control the depth of the SMTZ. In addition, they suggest

15 that transport processes play a dominant role for non-turbulent methane effluxes from ESAS sediments. The following sections



explore the role of each of the investigated environmental conditions on methane efflux in more detail. They also shed light on the mechanisms behind non-turbulent methane escape from ESAS sediments.

### 3.2.3 Role of advective transport

Fig. 3.a illustrates the effects of sedimentation rate $\omega$ on the flux of methane across the SWI. For both active ($v_{up} = 1$ cm

yr$^{-1}$) and passive ($v_{up} = 0$ cm yr$^{-1}$) settings, simulated CH$_4$ effluxes increase exponentially with sedimentation rate (log-log linear, see fig. 3.c) from $5.5 \cdot 10^{-15}$ $\mu$molCH$_4$ cm$^{-2}$ yr$^{-1}$ for low sedimentation rates ($\omega = 0.03$ cm yr$^{-1}$) to values as high as 27.5 $\mu$molCH$_4$ cm$^{-2}$ yr$^{-1}$ for high sedimentation rates ($\omega = 1.5$ cm yr$^{-1}$). Accordingly AOM acts as an efficient filter for upward diffusing methane (with $\eta \sim 100\%$, see Fig. S3), in slowly accumulating sediments. Integrated AOM rates ($\Sigma$AOM), for both active and passive settings, are in agreement with these findings. They range from $0.04 - 3.7$ mol m$^{-2}$

yr$^{-1}$ and are, thus, comparable to values that are typically observed in sediments characterised by an efficient AOM biofilter (*e.g.* Albert et al. (1998); Martens et al. (1998); Regnier et al. (2011)). In contrast, the efficiency of the AOM biofilter drops to $50 - 0\%$ for high sedimentation rates. The main driver behind the simulated high CH$_4$ fluxes and low AOM efficiencies in these rapidly accumulating sediments, are enhanced methanogenesis rates. High sedimentation rates facilitate not only the supply of organic matter to the methanogenic zone of the sediment, but also reduce residence times in the upper sediment layer,

resulting in a lower OM age (see eq. S13, S15)/degradation state (see eq. S11) within the methanogenic zone. The enhanced supply of reactive OM to anoxic sediment layers supports higher methanogenesis rates, resulting in higher methane porewater concentrations and an upward shift of the SMTZ.

In addition, the presence of active fluid flow further enhances methane efflux. The CH$_4$ fluxes from below adds complexity to the overall methane dynamics and this effect is investigated further by contrasting Damköhler numbers for passive and

active margins. Table 2 shows that for low to intermediate sedimentation rates, $Da_{AOM}$ values significantly decrease with $v_{up}$, indicating that less and less methane consumption occurs within the typical transport time scale $\tau_T$, thus, leading to a reduction in AOM biofilter efficiency. For instance, for $\omega = 0.123$ cm yr$^{-1}$, $\tau_T$ is about three orders of magnitude slower than $\tau_R$ without the presence of active fluid flow, while for $v_{up} = 10$ cm yr$^{-1}$ $\tau_T$ accelerates and is only one order of magnitude slower than $\tau_R$, resulting in a reduced consumption within the SMTZ. Accordingly, the decrease in $Da_{AOM}$ coincides with an increase in

CH$_4$ effluxes (Fig. 3. The trend in $Da_{AOM}$ is reversed for high sedimentation rates ($\omega > 1.5$ cm yr$^{-1}$, *i.e.* $Da_{AOM}$ increases with increasing $v_{up}$, while CH$_4$ efflux remains constant. This increase in $Da_{AOM}$ can be explained with a simple increase in AOM rates due to the build-up of methane gas in deeper sediment layers and its partial re-dissolution with in the AOM zone where porewater methane concentrations decrease (also see Fig. 4 below).

Maximum simulated flux differences between active and passive settings can reach up to 10 orders of magnitude. Yet, flux

differences quickly decrease with increasing sedimentation rates. Rapidly accumulating sediments show almost no difference in efflux between active and passive sites (Fig. 3.a). In contrast to sedimentation rates, the mechanism behind the control of $v_{up}$ on non-turbulent methane efflux is straightforward and self-evident. Active flow enhances the upcore transport of CH$_4$, shifting the SMTZ upcore and, thus, increasing CH$_4$ concentrations at shallow sediment depths (see Fig. 3.d). The apparent paradox of the CH$_4$ efflux insensitive to fluid flow in fast accumulating sediments can be resolved by examining the


**Figure 3.** *a*. Barplot of the methane flux at the SWI versus $\omega$ for passive case (plain style) and active case (pattern style) and the $[CH_4]-$ reported in the text. The squared value of $\omega$ is the reference value. *b*. Semilog plot of methane flux at SWI versus $v_{up}$ for the different $[CH_4]-$ reported in the text. *c*. Log-log plot of methane efflux at SWI versus $\omega$ for passive case (diamonds) and active case (circle). The log-log fit is also displayed. *d*. Log-log plot of SMTZ depth versus $\omega$ for passive case (diamonds) and active case (circle) with log-log fit. The red line is the trend found by Egger et al, 2018 (the term $\log(100)$ is to take into account unit conversion).



**Table 2.** AOM Damköhler number for $\omega = 0.123$ cm yr$^{-1}$ and $\omega = 1.5$ cm yr$^{-1}$. The two values are for the maximum and minimum values among the simulations with different bottom methane concentration. Missing values are because simulations were not run with the corresponding pair of parameters.

| | | $v_{up}$ [cm yr$^{-1}$] | | | | | | |
|---|---|---|---|---|---|---|---|---|
| | | 0 | 0.3 | 0.5 | 1 | 3 | 7 | 10 |
| $\omega$ | 0.123 | 1206 | 1124 | 683 | 327 | 120 | 52 | 32 |
| [cm yr$^{-1}$] | | 1521 | 1473 | 772 | 409 | 139 | 57 | 42 |
| | 1.5 | 470 | - | - | 1408 | - | - | - |
| | | 518 | | | 1630 | | | |

dissolved CH$_4$ depth profiles (Fig. 4). Simulated depth profiles are nearly identical and reveal CH$_4$ concentrations at or near the saturation concentration. In fast accumulating sediments, high methanogenesis rates result in an over-saturation of porewaters directly below the generally shallow SMTZ. High methanogenesis rates thus support the build up of methane gas. Methane gas formation also explains why, in for these cases, integrated methanogenesis exceed no-turbulent CH$_4$ fluxes by up to 6 times.

In rapidly accumulating, active and passive sediments, non-turbulent CH$_4$ fluxes are thus essentially identical. However, active settings will be characterised by the additional build-up of gaseous CH$_4$ and its potential escape through the sediment-water interface- a process not simulated in the present study.

Model results thus show that the dominant mechanism behind the observed transport-control on non-turbulent CH$_4$ efflux is an overall increase in CH$_4$ concentration and an upcore shift of the SMTZ rather than an increasing relative contribution

of advective transport processes to the total efflux. In fact, a comparison of the different methane transport processes across the SWI (Fig. 5) shows that the relative contribution of both the advection and molecular diffusion flux to the total flux is small and further decreases with increasing $v_{up}$. High non-turbulent methane effluxes in rapidely accumulating and/or active settings are thus largely driven by the non-local irrigation flux (see section 3.2.5 for more details on the role of irrigation). With increasing $\omega$ or $v_{up}$, the SMTZ shifts upcore, resulting in higher methane concentrations at shallow sediment depths

and thereby reinforcing the relative contribution of non-local transport for CH$_4$ fluxes, as well as lowering the efficiency of the AOM barrier from $\eta \sim 100\%$ to $\eta \sim 78\%$. The important role of the SMTZ location as a key control on CH$_4$ efflux is further confirmed by the observed exponential relationship between the location of the SMTZ and $\omega$ (Fig. 3.$d$). This result is qualitatively in agreement with the global compilation of empirical data by Egger et al. (2018), which reveals the same log-log decreasing trend between SMTZ and sedimentation rate. Our results are also consistent with observations from brackish

sediments that show that sedimentation rates $> 10$ cm yr$^{-1}$ give rise to high non-turbulent CH$_4$ fluxes ($20 - 80$ $\mu$molCH$_4$ cm$^{-2}$ yr$^{-1}$) and a high OM burial efficiency ($\sim 78\%$, Egger et al. (2016)). Egger and co-workers explained these findings by the slow growth of AOM microorganisms and the resulting inability of the microbial community to consume all of the CH$_4$ produced. Yet, our results show that the same pattern can be observed without having to invoke a low AOM efficiency. Our

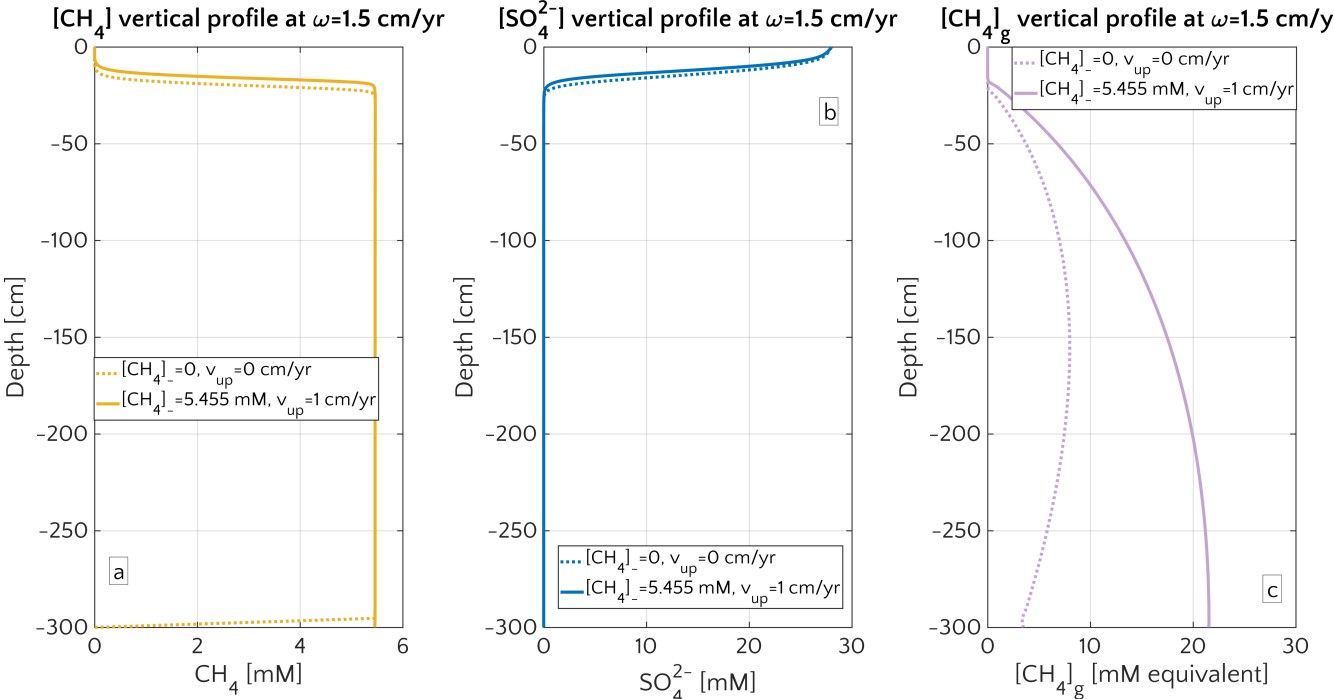

**Figure 4.** Porewater profiles in case of $\omega = 1.5$ cm yr$^{-1}$ for CH$_4$ (*a*), SO$_4^{2-}$ (*b*) and gaseous CH$_4$ (*c*). Dashed lines are simulation in passive scenario with [CH$_4$]$_-$ = 0 mM, while continuous lines simulations display active scenario with [CH$_4$]$_-$ = 5.455 mM, corresponding to the saturation concentration in the environmental conditions considered for the representative profile.

simulations thus indicate that the rapid burial of reactive organic matter to deeper sediment layers in rapidly accumulating sediments is sufficient to explain high CH$_4$ effluxes.

### 3.2.4   Role of organic matter quality

The quality of organic matter deposited onto the sediment exerts an additional control on CH$_4$ efflux. Fig. 6 illustrates the
5   influence of organic matter quality (as a function of OM degradation model parameter $a$, see eq. S11) and sedimentation rate $\omega$ on non-turbulent methane efflux for both active and passive settings, as well as different methane fluxes from below. Results corroborate the dominant influence of sedimentation rates on methane efflux, while organic matter quality exerts a secondary control. Maximum fluxes are generally simulated for rapidly accumulating sediments $\omega > 0.5$ cm yr$^{-1}$ that receive organic matter of intermediate quality ($a = 10 - 100$ yr).
10   These findings are in agreement with previously published studies (Regnier et al., 2011; Meister et al., 2013) and can be explained with the fact that high methanogenesis rates require a supply of reactive OM to the methanogenic zone. If organic matter quality is high ($a < 10$ yr), methanogenesis becomes substrate limited due to the rapid degradation of organic matter through energetically more favourable degradation pathways in the shallow sediments. In turn, if organic matter quality is


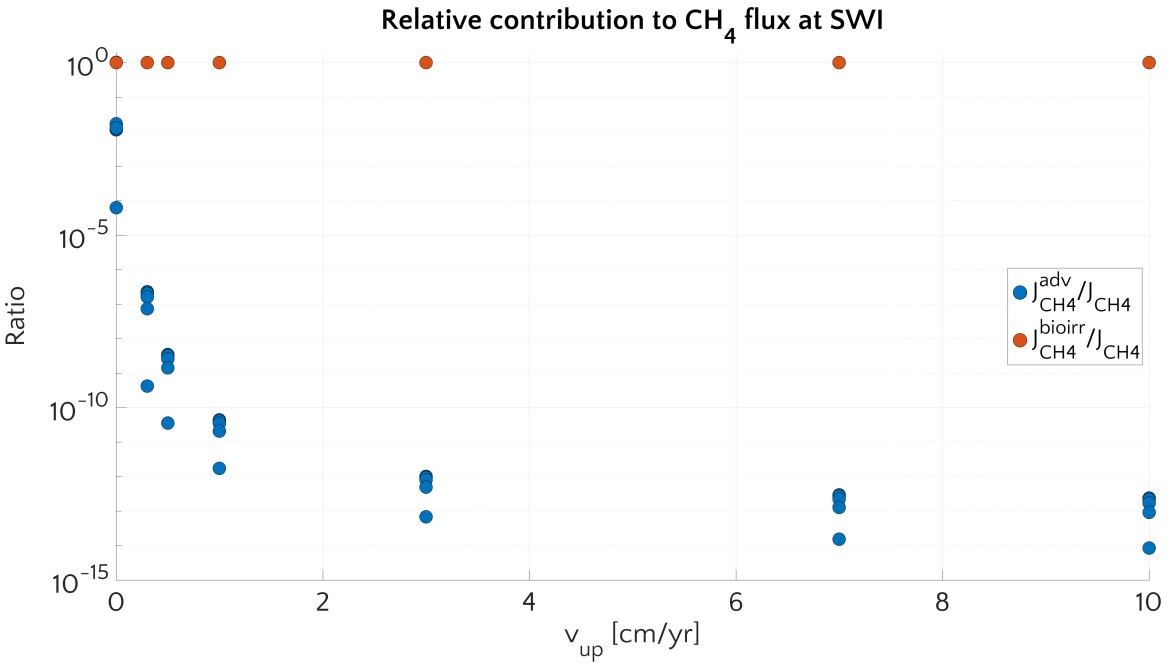

**Figure 5.** Relative contribution of transport process to the methane flux at the SWI: the advective component (blue) and the bioirrigation component (red). $\omega$ is set to the baseline value of 0.123 cm yr$^{-1}$. For each value of $v_{up}$ and a specific flux component each dot corresponds to a simulation with a different value of bottom CH$_4$ concentration. Diffusive component of the flux is always $< 10^{-10}$.

low ($a > 100$ yr), methanogenesis becomes reactivity limited. The ideal combinations of organic matter reactivity and sedimentation rate that result in maximum methane effluxes correspond to conditions characterised by OM that is i) sufficiently reactive to support enhanced methanogenesis rates and thus an accumulation of CH$_4$ at depth, but ii) sufficiently unreactive (in comparison to the burial rate) to escape the complete degradation in non-methanogenic sediments. Model results show that the

5    onset of active fluid flow and an enhanced methane supply from below (i.e. higher CH$_4$ concentration at the lower boundary) increase the CH$_4$ efflux through the SWI without altering the overall patterns.

### 3.2.5   Role of non-local transport

Fig. 7 further investigates the influence of bioirrigation on non-turbulent CH$_4$ efflux from the ESAS. It enhances methane efflux in sediments that are characterised by a shallow SMTZ, for instance, due to high sedimentation rates, active fluid flow and//or

10    methane flux from below. Yet, bioirrigation exerts a limited effect under a range of environmental conditions that favour a deep or shallow SMTZ location respectively.

In passive settings, changes in bioirrigation coefficient, $\alpha_0$, exert a limited influence on CH$_4$ effluxes. For most model scenarios, the SMTZ is located well below the sediment layer affected by bioirrigation ($z_{irr} = 3.5$ cm, hence bioirrigation is strongly suppressed below 15 cm) and, thus, changes in $\alpha_0$ have no effect on methane efflux. Changes in bioirrigation intensity



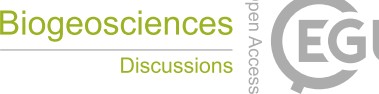

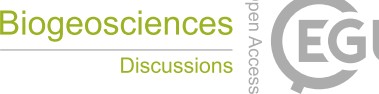

**Figure 6.** Flux of methane at the SWI as dependent on $a$ and $\omega$. For $[CH_4]_-$=0 mM (*left*) and $[CH_4]$=5.455 mM (*right*), and passive (*top*) and active (*bottom*) case. The circle with pattern corresponds to the baseline simulation.

only exert a noticeable effect on methane efflux when methane concentrations at the lower boundary exceed $[CH_4]_- = 5.455$ mM. Under these conditions, a decrease in methane efflux is observed with increasing $\alpha_0$, because the increasing bioirrigation activity supports an enhanced downcore transport of $SO_4^{2-}$, leading to a deepening of the SMTZ and a reduction in methane efflux. Model results thus partly support previously published findings by Cordes et al. (2005) and Niemann et al. (2006a), who





argued that bioirrigation increases methane consumption due to the enhanced downcore electron acceptors transport. However, model results also show that this effect is only observed under environmental conditions that result in a shallow SMTZ and that methane consumption and efflux remain largely unaffected by changes in bioirrigation intensity if the SMTZ is located deeper in the sediment.

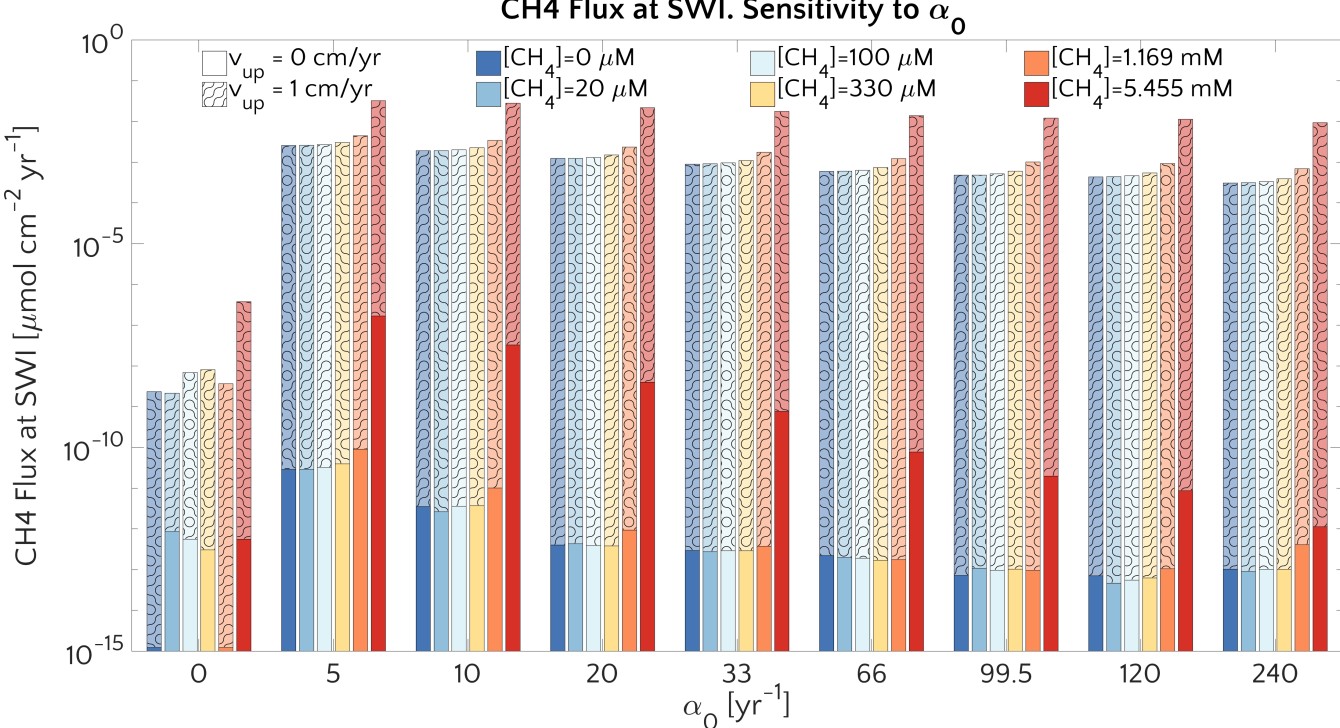

**Figure 7.** Barplot of the methane flux at the SWI versus $\alpha_0$ for passive case (plain style) and active case (pattern style) and the $[CH_4]$_ reported in the text.

5     In contrast to passive settings, active settings reveal a rapid increase in methane efflux with the onset of bioirrigation activity. Methane effluxes first increase by up to 5 orders of magnitude from $\alpha_0$=0 yr$^{-1}$ to $\alpha_0$=5 yr$^{-1}$, reaching maximum effluxes of $\sim 0.02$ $\mu$molCH$_4$ cm$^{-2}$ yr$^{-1}$, before remaining almost constant with a further increase in bioirrigation coefficients (up to 240 yr$^{-1}$). The simulated increase in methane efflux is a direct effect of the transport process itself, which enhances the upcore transport of methane accumulating in the upper sediment layers, including layers below the generally shallow SMTZ.

10   The subsequently simulated constant methane effluxes with increasing bioirrigation intensity in combination with the fact that bioirrigation represents the largest flux term at SWI (Fig.8) suggest that concentration differences close the the sediment-water interface remain broadly similar for all $\alpha_0 > 5$ yr$^{-1}$.

    These results are corroborated by the concomitant analysis of CH$_4$ dynamics over the 3-dimensional transport coefficient $\omega$, $v_{up}$ and $\alpha_0$ space shown in Fig.8.





**Figure 8.** Efflux of methane at the SWI as dependent on $v_{up}$ and $\omega$ for $\alpha_0 = 0$ yr$^{-1}$ (*a*), $\alpha_0 = 5$ yr$^{-1}$ (*b*), $\alpha_0 = 10$ yr$^{-1}$ (*c*) and $\alpha_0 = 33$ yr$^{-1}$ (*d*). Circles represent simulations outcomes. Results for $\alpha_0 \neq 0$ yr$^{-1}$ are almost the same. The lower boundary condition for methane is $[CH_4]_- = 1.169$ mM.



A comparison between simulations with $\alpha_0 = 0$ yr$^{-1}$ and $\alpha_0 \neq 0$ yr$^{-1}$ ($\alpha_0 = 5$ yr$^{-1}$, $\alpha_0 = 10$ yr$^{-1}$ and $\alpha_0 = 33$ yr$^{-1}$) shows that irrigation increases the CH$_4$ efflux at low to intermediate sedimentation rates and/or high $v_{up}$ (lower-left corner of the phase space in both plots). Yet, maximum methane effluxes that are simulated for high sedimentation rates or $v_{up}$ are almost identical between bioirrigated and non-irrigated sites despite the differences in dominant transport mechanism (diffusion when

$\alpha_0 = 0$ yr$^{-1}$; irrigation when $\alpha_0 \neq 0$ yr$^{-1}$). Under these conditions (i.e. high $v_{up}$ and/or high $\omega$), the SMTZ is located close to the SWI. Under these conditions, non-local transport becomes the dominant transport process in bioirrigated sediments (see section 3.2.3) because it weakens concentration gradients near the SWI and, thus, contributes to a substantial reduction in the gradient-driven, diffusive transport terms. As a consequence, simulated CH$_4$ efflux at the SWI are are broadly similar for all of the investigated $\alpha_0 \neq 0$ yr$^{-1}$ (Fig. 8.*b,c,d*). It is worth noticing that, independently on the $\alpha_0$, CH$_4$ efflux for $\omega = 0.03$ cm

yr$^{-1}$ and $v_{up} = 10$ cm yr$^{-1}$ is $\sim 1$ $\mu$molCH$_4$ cm$^{-2}$ yr$^{-1}$- a value almost identical to the one reported in Luff and Wallmann (2003) - 1.4 $\mu$molCH$_4$ cm$^{-2}$ yr$^{-1}$ - for a sediments characterised by $v_{up} = 10$ cm yr$^{-1}$ and $\omega = 0.0275$ cm yr$^{-1}$.

### 3.2.6 AOM rate constant

Given its crucial role in AOM biogeochemistry, one would expect a pronounced influence of the kinetic rates constant, $k_{AOM}$, on non-turbulent methane effluxes. However, simulation results reveal that $k_{AOM}$ only plays a minor role for non-turbulent

methane fluxes across the SWI (see Fig. S11, S12). An increase in $k_{AOM}$ can reduce methane effluxes from passive shelf sediments by up to 5 order of magnitude. Still, its effect remains small compared, for instance, to the response to variations in sedimentation rate, which can change methane efflux by up to 14 orders of magnitude. The most important effect of increasing $k_{AOM}$ is the increasing linearity of the [CH$_4$] and [SO$_4^{2-}$] profiles around the SMTZ and the concurrent narrowing and down-core movement of the SMTZ, which can result in a reduction in methane efflux. Model results thus show that the AOM biofilter

and, as a consequence, non-turbulent methane effluxes from sediments are not affected by the exact value of the kinetic rate constant, at least in the range we analyzed. This is in disagreement with results by Dale et al. (2008c), which show that, in dynamic settings subject to large methane fluxes, an increase of 3 orders of magnitude in $k_{AOM}$ (from $10^2$ M$^{-1}$ yr$^{-1}$ to $10^5$ M$^{-1}$ yr$^{-1}$) leads to a reduction in steady state methane fluxes below $10^{-2}$ $\mu$molCH$_4$ cm$^{-2}$ yr$^{-1}$. However this discrepancy might be ascribable to the high water flow velocity employed in their simulation ($v_{up} = 10$ cm yr$^{-1}$), ten times higher than

the one we considered in our active simulations. Finally, on the ESAS, dissolved methane concentrations are limited by the comparably low gas saturation concentration, resulting in lower methane fluxes (fig. S11) limiting the influence of $k_{AOM}$. In addition, Luff and Wallmann (2003) already showed that, as long as not null, the actual value of $k_{AOM}$ plays only a secondary role for the precipitation of authigenic carbonate. Since this authigenic carbonate precipitation is largely driven by alkalinity produced during AOM, the observed independence precipitation rates from $k_{AOM}$ supports our findings.

### 3.2.7 Summary of steady state experiments

Succinctly, the results of the steady state sensitivity study indicate that, under environmental conditions that are broadly representative for the ESAS, low AOM efficiencies and thus high non-turbulent CH$_4$ effluxes (larger than 4 $\mu$molCH$_4$ cm$^{-2}$ yr$^{-1}$) are promoted by intense advective transport (sedimentation rate $\omega > 1$ cm yr$^{-1}$, active fluid flow $v_{up} > 7$ cm yr$^{-1}$). Under





these conditions, $CH_4$ efflux can be further enhanced by moderate OM reactivity ($a = 10 - 10^2$ yr) and intense non-local transport processes, such as bioirrigation (irrigation constant $\alpha_0 > 0$ yr$^{-1}$). Overall, non-turbulent $CH_4$ fluxes appear to be mainly controlled by the concurrent effects of $\omega$, $v_{up}$ and $\alpha_0$. In contrast, maximum AOM rates, $k_{AOM}$, exert no influence on the AOM filter efficiency.

### 3.2.8 Geographic pattern and potential for non-turbulent methane emissions from Laptev Sea sediments

The results of the model sensitivity study provide a quantitative framework in which first-order estimates of potential non-turbulent methane escape from ESAS sediments can be derived. For instance, the functional relationship between sedimentation rate and methane flux across the SWI reported in Fig. 3.*c* allows estimating a potential non-turbulent methane efflux for a given sedimentation rate. Thus, if the spatial distributions of these environmental controls on methane efflux are known, a first-order
geographical distribution of potential non-turbulent methane escape from the Siberian Shelf can be derived. However, the availability of observational data from the Siberian Shelf is extremely scarce. Therefore, we here focus on the Laptev Sea - a comparable well studied part of the Siberian Shelf. The Laptev Sea is well-known for its subsea permafrost and gas hydrate content and subject to large riverine inputs from the Lena river. To derive a map of sedimentation rates for Laptev Sea shelf sediments, we use published linear sedimentation rates (Table S7) and extrapolate these values to the entire region by applying
a simple 3D kriging method (see Fig. 9.*a*), using the International Bathymetric Chart of Arctic Ocean (IBCAO) (Jakobsson et al., 2012) and employing longitude, latitude and water depth as predictors for $\omega$.

Observations indicate that sedimentation rates are highest ($\omega$=0.45 cm yr$^{-1}$) close to the mouth of the Lena river and Moustakh Island in the Buor-Khaya Gulf. As a consequence, the vicinity of the river mouth, as well as the area along the shallow bathymetric profile towards the NE of the Lena delta are characterized by comparably high sedimentation rates ($\omega = 0.27 - 0.42$ cm yr$^{-1}$). The relatively shallow areas ($\sim 10$ m deep) around the New Siberian islands reveal intermediate values
($\omega = 0.06 - 0.12$), while minimum sedimentation rates ($\sim 0.002 - 0.03$ cm yr$^{-1}$) roughly follow the 55 m isobath down to the continental slope at 100m. Deeper shelf areas are characterized by a more homogeneous distribution of sedimentation rates with values around $0.03 - 0.06$ cm yr$^{-1}$.

**Table 3.** Estimated flux of $CH_4$ at SWI in mol yr$^{-1}$ for different depth regions of Laptev Sea in a passive ($v_{up} = 0$ cm yr$^{-1}$) and active ($v_{up} = 1$ cm yr$^{-1}$) case.

|  | $v_{up}$ | |
| --- | --- | --- |
| Region (water depth, area) | 0 | 1 |
| $0 - 10$ m, $7.7 \cdot 10^4$ km$^2$ | 6.5 | $8.9 \cdot 10^5$ |
| $10 - 80$ m, $4.5 \cdot 10^5$ km$^2$ | 296.2 | $8.5 \cdot 10^6$ |

Estimated non-turbulent methane effluxes corresponding to the highest measured sedimentation rates close to the Lena
mouth do not exceed $1.57 \cdot 10^{-1}$ $\mu$mol$CH_4$ cm$^{-2}$ yr$^{-1}$ assuming the presence of active fluid flow and $2.25 \cdot 10^{-5}$ $\mu$mol$CH_4$ cm$^{-2}$ yr$^{-1}$ for passive settings. These findings are not surprising as steady state sensitivity results indicate that high $CH_4$ efflux


Sedimentation rate across Laptev Sea

Flux of CH$_4$ across Laptev Sea (v$_{up}$=0 cm/yr)

Flux of CH$_4$ across Laptev Sea (v$_{up}$=1 cm/yr)

**Figure 9.** *a.* values of the sedimentation rate extrapolated for the whole Laptev Sea via a simple kriging method. The reference values (circles) are the ones reported in Table S7. *Bottom* (Log) Values of the potential methane emissions at the SWI considering the relationship presented in Fig. 3.*c* for passive (*b*) and active (*c*) cases.

requires sedimentation rates of $\omega > 1$ cm yr$^{-1}$. The regional non-turbulent CH$_4$ efflux budget for different depth sections of the Laptev Sea assuming the absence of active fluid flow in Laptev Sea shelf sediments (see Table 3) thus indicates that non-turbulent CH$_4$ efflux is negligible. Even if we assume the omnipresence of an active fluid flow of $v_{up} = 1$ cm yr$^{-1}$, the



estimated non-turbulent methane efflux merely sums up to $9.39 \cdot 10^6$ molCH$_4$/yr ($\sim 0.1$ GgCH$_4$/yr) over the entire Laptev Sea area of $527.4 \cdot 10^3$ km$^2$. Such small effluxes would most likely be subject to further oxidation in the water column, thus limiting any potential impact on atmospheric methane concentrations and climate.

Higher advective fluid flow velocities, intermediate organic matter reactivity and/or a more intense macrobenthic biological
activity could increase these estimates of non-turbulent methane escape from the Laptev Sea shelf. Higher advective fluid flow velocities (i.e. $v_{up} > 1$ cm yr$^{-1}$), possibly in connection with active seepages, groundwater discharges and fault lines (the latter follow parallel pattern in Laptev Sea Drachev et al. (1998) on the direction SW-NE from the west of Lena delta up to the little Lyakhovsky and Kotelny island), could result in methane effluxes of up to $10 - 10^{1.3}$ $\mu$molCH$_4$ cm$^{-2}$ yr$^{-1}$ (see Fig. 6 and Fig. 8). However, such high fluid flow velocities would be only found locally and would thus merely give rise to a number
of methane emission hot spots that would not change the overall non-turbulent methane flux budget. In addition, intermediate organic matter reactivity, in particular in the fast accumulating sediments close to the coastline and the Lena River Delta that receive more reactive organic matter from thawing terrestrial permafrost (Wild, 2019) could result in a higher estimated non-turbulent methane escape . However, our sensitivity study results show that OM reactivity merely plays a secondary role, suggesting that changes in OM reactivity would only change efflux by less than an order of magnitude assuming both $a = 100$
yr or $a = 1$ yr. Changes in bioirrigation intensity would exert merely a limited effect on efflux estimates, as bioirrigation has already been included in the estimate calculations. Additional physical reworking such as ice scouring or dredging, or the absence of bioirrigation, which is known to be patchy in Arctic sediments could even further reduce estimated methane efflux.

Model results thus show that, under present-day, steady state environmental conditions, AOM acts as an efficient biofilter for potential non-turbulent methane fluxes in Laptev Sea sediments. The estimated non-turbulent methane escape from Laptev
Sea shelf sediments cannot support previously estimated methane outgassing fluxes of few teragrams of CH$_4$ yr$^{-1}$ (Berchet et al., 2016) or even tens of teragrams of CH$_4$ yr$^{-1}$ (Shakhova et al., 2014). If such outgassing were to be supported by methane efflux from Laptev Sea sediments, it would require the build-up of CH$_4$ gas reservoirs in Laptev Sea sediments of at least similar or larger size than the evaded amount, as well as the preferential and rapid transport of this CH$_4$ gas to the atmosphere. Nevertheless, model results also suggest that projected trends of terrestrial permafrost thawing and coastal
permafrost degradation (Vonk et al., 2012) might increase the importance of non-turbulent methane escape for the Artic's methane budget by potentially increasing sedimentation rates through coastal erosion and increased riverine inputs (Guo et al., 2007); active fluid flow through permafrost and methane gas hydrate degradation (James et al., 2016; Ruppel and Kessler, 2017); organic matter reactivity through an enhanced delivery of more reactive permafrost organic matter (Wild et al., 2019) and/or an enhanced macrobenthic activity through warming and Atlantification. However, the magnitude of these projected
environmental changes and thus there effect on non-turbulent methane escape from ESAS sediments is difficult to assess.

### 3.3 Methane efflux dynamics in response to seasonal and long term environmental variability

Although steady state sensitivity results revealed that AOM represents an efficient biofilter for upward migrating methane, transient dynamics induced by, for instance, seasonal variability, or climate change, may weaken the efficiency of the AOM biofilter. Therefore, we also explore the potential for non-turbulent methane escape from ESAS sediments under transient





conditions. Table 4 summarizes maximum simulated, non-turbulent methane escape for two seasonal environmental change scenarios, as well as two longterm environmental change scenarios.

**Table 4.** Maximum of methane fluxes (in $\mu$mol cm$^{-2}$ yr$^{-1}$) at SWI for transient simulations. Values in round parenthesis indicate the year after the beginning of simulation corresponding to the reported maximum.

| | | [*1.Seasonal* CH$_4$] | | | [*2.Seasonal* CH$_4$ + SO$_4^{2-}$] | | |
| | | $v_{up}$ (cm yr$^{-1}$) | | | $v_{up}$ (cm yr$^{-1}$) | | |
| | | 0 | 1 | 5 | 0 | 1 | 5 |
|---|---|---|---|---|---|---|---|
| CH$_4$ ($\mu$M) | 20 | 0.030 (200) | 0.550 (50) | 12.7 (17.5) | 0.059 (200) | 0.772 (51) | 13.7 (18) |
| | 100 | 0.029 (200) | 0.550 (50) | 12.7 (17.5) | 0.058 (200) | 0.753 (51) | 13.7 (18) |
| | 330 | 0.030 (200) | 0.552 (49.5) | 12.8 (18) | 0.058 (200) | 0.775 (51) | 13.8 (18) |
| | 1169 | 0.031 (200) | 0.558 (49.5) | 12.9 (18) | 0.059 (200) | 0.783 (51) | 14.0 (18) |
| | 5455 | 0.034 (200) | 0.577 (49) | 14.0 (19) | 0.062 (200) | 0.832 (50) | 15.2 (19) |
| | | [*3.Linear* CH$_4$] | | | [*4.Sudden* CH$_4$] | | |
| | | $v_{up}$ (cm yr$^{-1}$) | | | $v_{up}$ (cm yr$^{-1}$) | | |
| | | 0 | 1 | 5 | 0 | 1 | 5 |
| CH$_4$ ($\mu$M) | 20 | 0.029 (200) | 0.550 (50) | 11.7 (20) | 0.029 (200) | 0.550 (50) | 12.7 (18) |
| | 100 | 0.030 (200) | 0.550 (50) | 11.7 (20) | 0.030 (200) | 0.552 (50) | 12.7 (18) |
| | 330 | 0.030 (200) | 0.550 (50) | 11.7 (20) | 0.031 (200) | 0.557 (50) | 12.9 (18) |
| | 1169 | 0.032 (200) | 0.550 (50) | 11.7 (20) | 0.033 (200) | 0.565 (49.5) | 13.4 (18) |
| | 5455 | 0.036 (200) | 0.560 (50) | 11.8 (20) | 0.040 (200) | 0.639 (47) | 18.8 (23) |

Interestingly, model results reveal that the temporal dynamics of simulated, non-turbulent methane fluxes does not depend on the specific environmental scenario (*i.e.* fluxes respond in a similar way to all methane forcing scenarios), but is rather

5  controlled by the presence/absence of active fluid flow (Table 4). In passive settings, methane efflux monotonously increases to low, maximum fluxes of 0.03-0.05 $\mu$molCH$_4$ cm$^{-2}$ yr$^{-1}$. At the same time, the SMTZ merely migrates $11.5 - 29$ cm upcore (Fig. S15). Over a period of 200 years, the non-turbulent methane escape from passive settings merely reaches 3-4 $\mu$molCH$_4$ cm$^{-2}$. Even under transient environmental conditions on both seasonal and longterm scales, passive settings thus generally allow for little methane escape (Fig. S14). Active settings, in turn, are characterised by an initial increase in CH$_4$ fluxes to

10  maxima of 0.55-0.83 $\mu$molCH$_4$ cm$^{-2}$ yr$^{-1}$ ca. 50 years into the simulation (assuming a flow velocity of $v_{up} = 1$ cm yr$^{-1}$). During this initial time period, the SMTZ rapidly shifts upcore by 100 cm. While the SMTZ subsequently remains located in shallow sediment layers for the remaining simulation period, methane escape temporarily decreases by 17-20% until year 70-75, followed by a monotonous increase until the end of the simulation at year 200. For $v_{up} = 1$ cm yr$^{-1}$, the temporally





integrated methane efflux (over 200 years) falls within the range 66-121 $\mu$molCH$_4$ cm$^{-2}$. For $v_{up} = 5$ cm yr$^{-1}$, the integrated efflux is 10- to 14-fold larger (i.e $\sim 0.95 - 1.154$ mmolCH$_4$ cm$^{-2}$). For $v_{up} = 1$ cm yr$^{-1}$, almost 30% of these emissions occurs in the first century after the perturbation. This fraction increases to 48-87% for $v_{up} = 5$ cm yr$^{-1}$.

The similarity of the CH$_4$ efflux dynamics in response to different environmental scenarios (*i.e.* seasonal CH$_4$, seasonal CH$_4$+SO$_4^{2-}$, linear CH$_4$ and sudden CH$_4$) as well as the smooth, continuous upcore movement of the SMTZ thus indicates that the response time of the biogeochemical process network that controls CH$_4$ dynamics and efflux (*i.e.* biomass growth, AOM rate, methanogenesis) is slower than the characteristic timescales of the investigated environmental variability. In addition, results show that notable methane escape from sediments in response to environmental variability on both seasonal as well as long timescales requires environmental conditions that allow for the creation of a "window of opportunity" during which the efficiency of the AOM biofilter is temporarily weakened. The following sections explore the factors that control the creation of such a window of opportunity and discusses the mechanisms behind the simulated methane escape.

### 3.3.1 Window of opportunity

Given the broadly similar behaviour of methane fluxes to the range of environmental scenarios, we will focus the following discussion on scenario 4 (*i.e.* step-like CH$_4$ forcing) with $v_{up} = 1$ cm yr$^{-1}$ and a specific bottom concentration, *e.g.* [CH$_4$]$_- = 1.169$ mM. In contrast to the other scenarios that are characterized by transient CH$_4$ supply from below, scenario 4 allows for a straightforward definition of initial and final state, which allows attributing a typical system response timescale.

Fig. 10 illustrates the temporal evolution of the simulated filter efficiency and AOM rate (a), CH$_4$ flux (b), SMTZ depth (c) and AOM biomass (d) for scenario 4 ($v_{up} = 1$ cm yr$^{-1}$, [CH$_4$]$_- = 1.169$ mM). During the initial 23 years, AOM biomass is constant and thus, AOM rate and, filter efficiency are zero. In addition, aerobic methane oxidation represents a weak barrier as oxygen is merely present in the upper few centimetres and aerobic methane oxidation competes with aerobic organic matter degradation, as well as a number of additional secondary redox reactions (see Table S3). As a consequence, *in situ* produced, as well as externally supplied methane diffuses upward and mostly escapes, leading to an increase in CH$_4$ fluxes. A large fraction of this methane efflux is produced *in-situ* since the average advective velocity of methane in the sediment ($\bar{v} = v_{up} - \omega = 0.877$ cm yr$^{-1}$) merely covers 20.2 cm in 24 years. It is hence too slow to allow methane from deep sources to reach the sediment-water interface.

Transient model results thus reveal that the temporary perturbation of AOM and, thus, the creation of a "window of opportunity" for methane escape from sediments requires a significant shift of the SMTZ, which has to be rapid enough to prevent the establishment of an AOM community within the SMTZ. In the passive settings, all investigated environmental scenarios trigger a limited and comparably slow movement of the SMTZ (Fig. S15) thus allowing for the establishment of an AOM community and preventing the creation of such a window of opportunity. In contrast, active settings show a rapid and significant shift of the SMTZ in response to methane supply from below, which creates a window of opportunity for methane escape, whose onset and duration is controlled by advective velocity $v_{up}$ of the active fluid flow and the AOM biomass growth. Assuming typical values of $v_{up}$ reported for active marine sediments (0.5-5 cm yr$^{-1}$), we show that methane from deep sources (ca. 3 m) reaches the sediment water interface within 7 to 20 years. Maximum CH$_4$ effluxes are typically simulated 2-3 decades after the onset





of methane supply. Furthermore, simulation results reveal that the maximum magnitude of methane effluxes increases with $v_{up}$ from 0.5-0.6 $\mu$molCH$_4$ cm$^{-2}$ yr$^{-1}$ for $v_{up} = 1$ cm yr$^{-1}$ to $11 - 15$ $\mu$molCH$_4$ cm$^{-2}$ yr$^{-1}$ for $v_{up} = 5$ cm yr$^{-1}$. In parallel, the duration of the window of opportunity for methane escape in turn decreases with increasing $v_{up}$. Values of methane fluxes for the maximum and for the new steady state fall in the range of other previous model results (Sommer et al., 2006; Dale et al.,

2008c) but do not reach the high values measured in other settings (Linke et al., 2005; Regnier et al., 2011).

An insight into the mechanism that drive the creation of this window of opportunity and control non-turbulent methane efflux under these conditions can be inferred by evaluating methane migration time scales within the sediments. After the first 23 years, AOM begins to efficiently consume upward migrating methane (see Fig. 10.$a$) and reduces the methane flux by 40%. Because consumption occurs at SMTZ, it does not immediately affect the methane efflux through the SWI. The effect of this

consumption on methane concentration first has to propagate upwards through the sediments till it reaches the SWI, resulting in a delayed efflux response to the onset of AOM. The velocity of this propagation is given by $\bar{v} = v_{SMTZ} + v_{up} - \omega$, where $v_{SMTZ}$ denotes the velocity at which the SMTZ (where consumption happens) moves upward. From Fig. 10.$c$ and fig. S15 we can infer that, initially, the SMTZ moves with a fairly constant velocity of about 2.46 cm yr$^{-1}$ and, hence, $\bar{v} = 2.46 + 1 - 0.123 = 3.337$ cm yr$^{-1}$. At the onset of an efficient AOM barrier (i.e. after 23 years), the SMTZ is located at a sediment depth of 100.4 cm

and the time required for the consumption signal to propagate to the SWI thus amounts to $\frac{100.4\,\mathrm{cm}}{3.337\,\mathrm{cm}\,\mathrm{yr}^{-1}} = 30.1$ yr. After this initial period of 53.1 years, methane consumption at the SMTZ starts to reduce non-turbulent methane efflux (Fig. 10.$b$).

Simulations show that $v_{SMTZ}$ is solely controlled by $v_{up}$ and does not depend on additional environmental conditions, as revealed by the constant velocity with which the SMTZ moves upwards ($\sim 11.4$ cm yr$^{-1}$). This indicates that the methane efflux is initially controlled by the velocity of the SMTZ movement, which is in turn is determined by the upward velocity

$v_{up}$. The reduction in methane efflux after the onset an efficient AOM barrier lasts until the upward movement of the SMTZ slows down. At this point, the AOM filter efficiency reaches a quasi-stationary level of $\sim 85\%$ (as Fig. 10.$a$). Meanwhile, in-situ methane production continues to produce methane, which is not entirely consumed by the AOM community that already reached its full capacity. As a consequence, methane fluxes at SWI increase again and until a new steady state is reached. This simulated pattern arises even more clearly in simulations with $v_{up} = 5$ cm yr$^{-1}$ (see Fig. S16). Here, the onset of a new

steady state occurs earlier and AOM suppresses the non-turbulent methane efflux to the value of about 7 $\mu$molCH$_4$ cm$^{-2}$ yr$^{-1}$ (Fig. S16). The comparison of such value with simulated steady state efflux under identical environmental conditions (i.e. $v_{up} = 5$ cm yr$^{-1}$, $\omega = 0.123$ cm yr$^{-1}$ and $[\mathrm{CH}_4]_- = 1.169$ mM; inferred from Fig. 8) indicates that the final steady state flux observed in transient simulations (bioenergetic AOM formualtion) is roughly two order of magnitude larger than the flux of $\sim 0.1$ $\mu$molCH$_4$ cm$^{-2}$ yr$^{-1}$ simulated with a bimolecular rate law. These findings are in agreement with Dale et al. (2008c),

who reported a new steady state efflux of similar magnitude (3 $\mu$molCH$_4$ cm$^{-2}$ yr$^{-1}$) for $v_{up} = 10$ cm yr$^{-1}$ and $[\mathrm{CH}_4]_- = 70$ mM. They also show that CH$_4$ efflux simulated with a bimolecular AOM rate law can vary from being higher to much smaller than the one estimated in the bioenergetic approach, depending on the value of $k_{AOM}$.

Such a conclusion might sound in disagreement with what we showed in section 3.2.6, where we deflated the role of $k_{AOM}$, but it has to be put into perspective. Firstly it is indeed expected that, also with the bimolecular AOM implementation, CH$_4$

flux increases if the $k_{AOM}$ were further reduced down, and it could not be otherwise, considered that it controls the AOM





**Figure 10.** Time evolution over 200 years for the case of an active setup with $v_{up} = 1$ cm yr$^{-1}$ and a step-like methane forcing from below from 0 to $[CH_4]_- = 1.169$ mM. *a.* AOM vertically integrated rate (blue) and AOM efficiency (red). *b.* $CH_4$ flux at SWI. *c.* SMTZ depth. *d.* Vetically integrated biomass (number of cells).

rate. But the employment of values for $k_{AOM}$ smaller than the ones we explored is not supported by any other previous study and would have then therefore rather arbitrary. Finally some light should be shed on why the bimolecular and the bioenergetic




AOM formulation give such different methane effluxes, under the same conditions. For this purpose we assessed an apparent

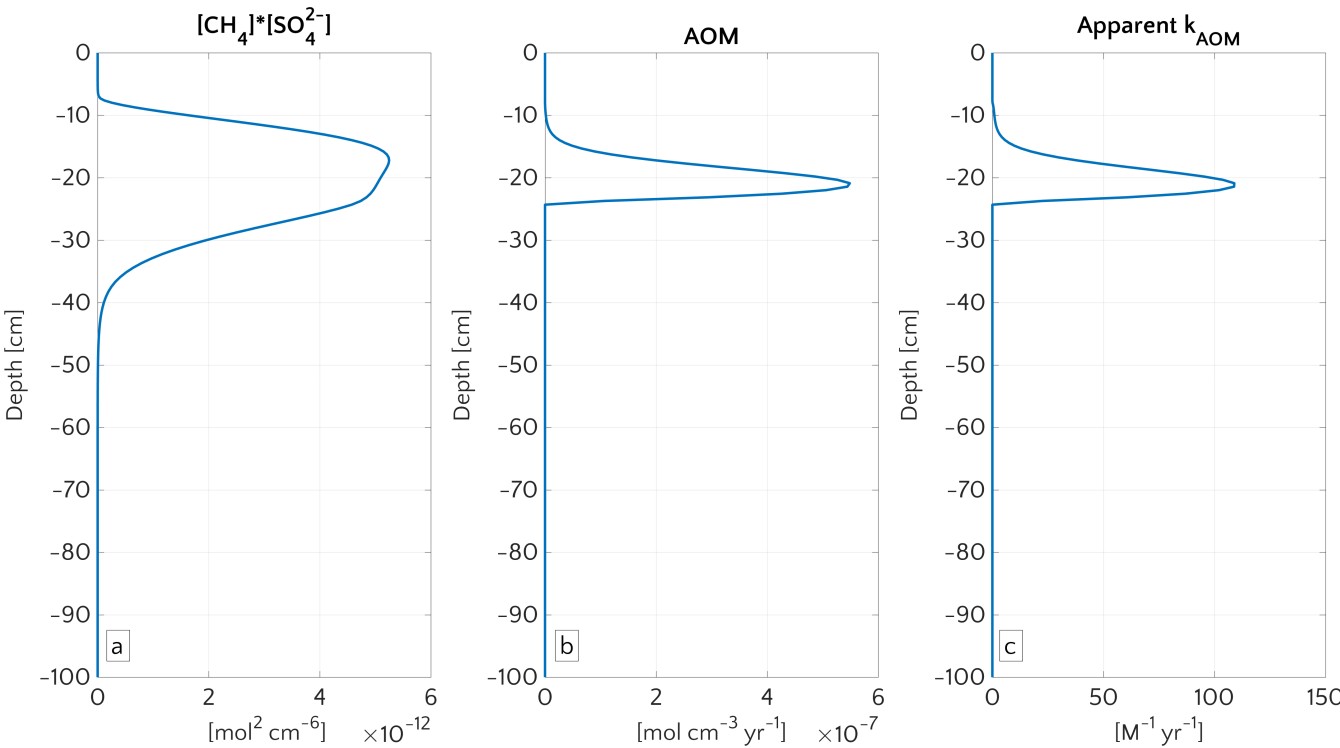

**Figure 11.** Vertical profiles at the end of transient simulation (after 200 years) with bioenergetic AOM fomulation for the case $[CH_4]_- = 1.169$ mM and $v_{up} = 5$ cm yr$^{-1}$. *a.* Bimolecular product $[CH_4] \cdot SO_4^{2-}$. *b.* AOM rate. *c.* Apparent $k_{AOM}$, estimated from eq. 7.

$k_{AOM}$, *i.e.* what $k_{AOM}$ would look like if we wanted to describe AOM rate we find at the end of the transient simulation by means of the bimolecular description of eq. 7. Results are shown in Fig. 11. Panel *a* shows the the product $[CH_4] \cdot [SO_4^{2-}]$ is broader than the AOM profile (panel *b*), which results then being is strongly limited by the thermodynamic constraint. Fig. 11.*c* also shows that the apparent $k_{AOM}$ is not constant and never exceeds $109$ M$^{-1}$ yr$^{-1}$, being for the most of the depths well below $100$ M$^{-1}$ yr$^{-1}$. It confirms that AOM resulting from bioenergetic formulation cannot be trivially described by a simple bimolecular expression of the rate with a constant value. This, combined to the low values of apparent $k_{AOM}$, gives reason of the difference in steady-state $CH_4$ effluxes.

The onset of an efficient AOM biofilter requires the establishment of an AOM community that is sufficiently large to consume upward migrating methane. Therefore, the onset of AOM and, consequently, the efficiency of the AOM filter are controlled by AOM biomass dynamics, which in turn are determined by kinetic and thermodynamic constraints. Fig. 12 illustrates the depth profiles of the thermodynamical and kinetic terms in the bioenergetic AOM formulation (eq. 8), as well as their evolution in response to the onset of a sudden methane flux from below. Initially, although kinetically possible (i.e. $F_K \neq 0$, eq. 9), AOM is inhibited by thermodynamic constraints (i.e. $F_T = 0$, eq. 10). These thermodynamic constraints ease when the SMTZ





becomes stationary after the initial decades. At that point, favourable conditions are encountered over a depth of about 20 cm (for methane scenario 4 and $v_{up} = 5$ cm yr$^{-1}$) and the increasing AOM filter efficiency reduces methane efflux (Fig. 12.$b$). After 200 years (Fig. 12.$c$), a more uniform sulfide concentration in lower sediments together with the upward movement of the SMTZ pushes the maximum of $F_T$ upwards, thus limiting the zone where AOM is thermodynamically favourable ($\sim 13$

5 cm deep). $F_T$ remains the main constraint on AOM throughout the simulation.

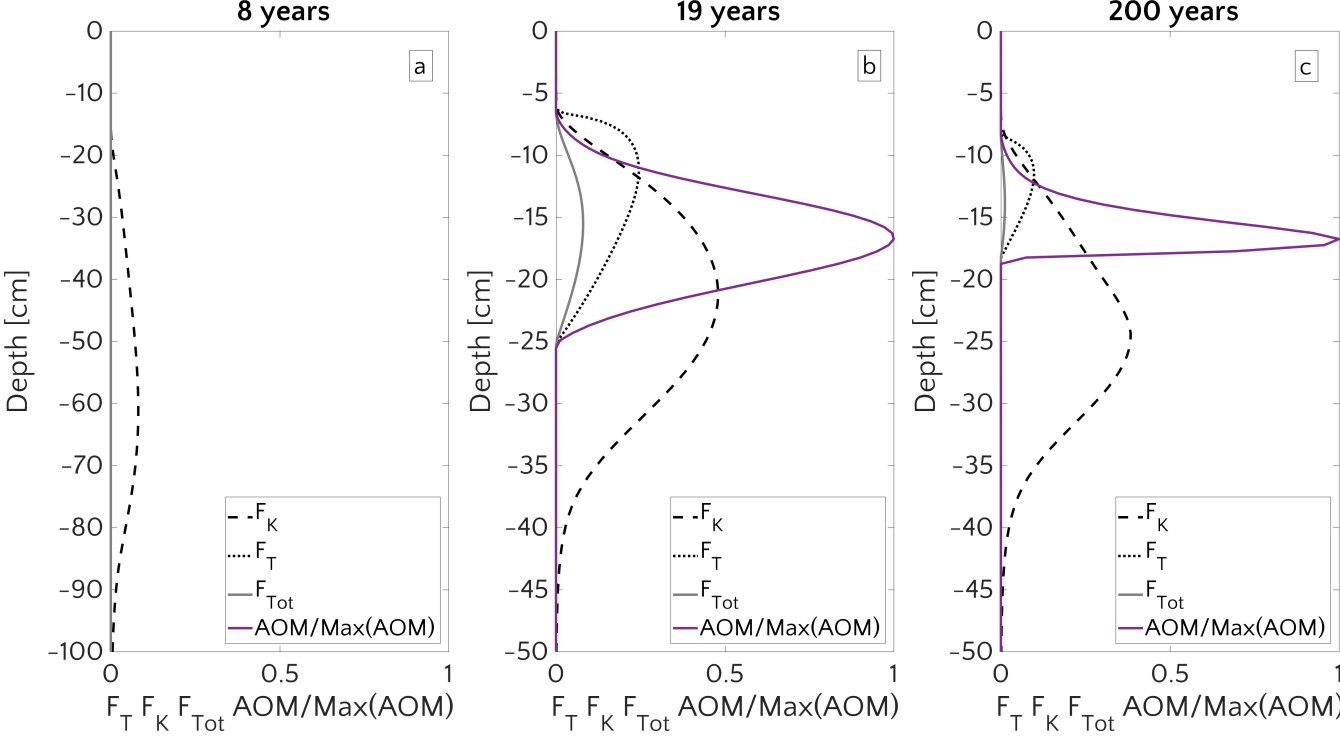

**Figure 12.** Vertical profile of $F_T$, $F_K$, $F_{Tot} = F_K \cdot F_T$ and the AOM (scaled to the maximum) for three instant in times. 8 years ($a$), 19 years ($b$)and 200 years (($c$) of simulation, for the case $[CH_4]_- = 1.169$ mM and $v_{up} = 5$ cm yr$^{-1}$.

Integrated biomass $\Sigma B$ ranges from $\sim 1.2 \cdot 10^{10}$ to $3.5 \cdot 10^{11}$ cells cm$^{-2}$ (except for simulation with $v_{up} = 5$ cm yr$^{-1}$ and $[CH_4]_- = 5.455$ mM, whose $\Sigma B = 1.2 \cdot 10^{12}$). These values are comparable with AOM biomass reported in Treude et al. (2003) $(1.5 - 1.8 \cdot 10^{10}$ cells cm$^{-2})$ or with values simulated in Dale et al. (2008c) $(3.7 \cdot 10^{11}$ cells cm$^{-2}$ for $v_{up} = 5$ cm yr$^{-1}$). In addition, the maximum simulated biomass for active settings of $0.5 - 2.5 \cdot 10^{10}$ cells cm$^{-3}$ agrees well with previously reported

10 values, ranging from 0.27 to $7.4 \cdot 10^{10}$ cells cm$^{-3}$ (Dale et al., 2008c). However, integrated AOM rates, $\Sigma$AOM, are instead smaller then previously published rates for shallow, active sites (Boetius et al., 2000; Haese et al., 2003; Luff and Wallmann, 2003; Linke et al., 2005; Wallmann et al., 2006b; Dale et al., 2008c), but comparable to those observed in active sites below the shelf break (Aloisi et al., 2004; Wallmann et al., 2006a; Maher et al., 2006) or in passive settings (Borowski et al., 1996; Martens et al., 1998; Fossing et al., 2000; Jørgensen et al., 2001; Dale et al., 2008c). The discrepancy is likely due to different

15 environmental conditions encountered at these sites. For instance, Dale et al. (2008c) applied an advective velocity of $v_{up} = 10$




cm yr$^{-1}$ and [CH$_4$]$_-$ = 60 mM). While differences in $v_u p$ affect the $\Sigma$AOM, its effect on $\Sigma$B is negligible since an efficient AOM microbial filter is known to account for at least $> 10^{10}$ cells cm$^{-3}$ (Lösekann et al., 2007; Knittel and Boetius, 2009).

Simulation results finally show that AOM biomass and, thus, AOM rate increase with an increase in methane supply from below (Fig. S17). The ratio between the flux of methane at the SWI and the advective methane flux at the bottom of the

sediment column reflects this behaviour. It decrease from values $> 1$ to values $< 1$ with an increase in methane from below (Fig. S18.$b$). This does not only mean that *in situ* methanogenesis rather than methane supply from below drives methane efflux for low methane supply scenarios, but also that the influence of methanogenesis on efflux decreases with an increase in methane supply. This shift is accompanied by a shift in the a diffusion-driven to a advection-driven influx (not shown). In addition, although absolute methane efflux is higher for higher [CH$_4$]$_-$, the smaller values of the efflux/influx ratio show that

the system becomes much more efficient in removing methane when it is forced with a higher methane flux. Simulation results show that the AOM biofilter efficiency increases by 17% (49% in passive settings) over the increase of [CH$_4$]$_-$ from 20 $\mu$M to 5.455 mM in agreement with observations (Treude et al., 2003; de Beer et al., 2006; Niemann et al., 2006a). However, although AOM becomes more efficient, it cannot keep up with increasing fluxes, indicating that the inability of the AOM biomass to completely consume higher CH$_4$ flux does not exclusively depend on the presence of methane bubbles, as previously stated

(James et al., 2016).

## 4 Conclusions

In this study, we evaluate the potential for non-turbulent methane escape from both passive as well as active East Siberian Arctic Shelf (ESAS) sediments that are affected by deep methane supply from, for instance, thawing subsea permafrost or methane gas hydrate dissociation. We identify the most important biogeochemical and physical controls on non-turbulent

methane escape from those sediments under steady state conditions, as well as in response to environmental variability on seasonal and centennial timescales. Finally, we derive a first regional estimate of (not-turbulent) methane benthic-pelagic flux and of potential methane consumption in the Laptev Sea.

Model results reveal that AOM is an efficient sink for upward migrating dissolved methane in ESAS sediments. Simulated non-turbulent methane effluxes are negligible for a broad range of environmental conditions under both steady state and tran-

sient conditions. On the ESAS, AOM is a transport-limited process and transport parameters thus exert an important control on the efficiency of the AOM biofilter and, thus, on methane efflux. Both steady state and transient model results confirm the key role of advective transport (sedimentation and active fluid flow) in supporting methane escape from Arctic shelf sediments. Under steady state conditions, high methane effluxes (up to 27.5 $\mu$mol cm$^{-2}$ yr$^{-1}$) are generally found for sediments that are characterized by high sedimentation rates and/or active fluid flow (sedimentation rate $\omega > 0.7$ cm yr$^{-1}$, active fluid flow

$v_{up} > 6$ cm yr$^{-1}$). Under these conditions, methane efflux can be further enhanced by intermediate organic matter reactivity (RCM model parameter $a = 10 - 10^2$ yr) and intense local transport processes, such as bioirrigation (irrigation constant $\alpha_0 > 1$ yr$^{-1}$). Our results indicate therefore that present methane efflux from ESAS sediments can be supported by methane gas escape and non-turbulent CH$_4$ efflux from rapidly accumulating and/or active sediments (*e.g.* coastal settings, portions close to river





mouths or submarine slumps). In particular, active sites sediments may release methane in response to the onset or increase of permafrost thawing or $CH_4$ gas hydrate destabilization.

High methane escape (up to 11-19 $\mu molCH_4$ cm$^{-2}$ yr$^{-1}$ corresponding to 2.6-4.5 $TgCH_4$ yr$^{-1}$ if upscaled to the ESAS) can occur during a transient period following the onset of methane flux from the deep sediments. Under these conditions,

substantial methane escape from sediments requires the presence of active fluid flow that supports a significant and rapid upward migration of the SMTZ in response to the onset of $CH_4$ flux from below. Such rapid and pronounced movements create a window of opportunity for non-turbulent methane escape by inhibiting the accumulation of AOM-performing biomass within the SMTZ - mainly through thermodynamic constraints - thereby perturbing the efficiency of the AOM biofilter. The magnitude of methane effluxes, as well as the duration of this window of opportunity, is largely controlled by the active flow

velocity. In addition, results of transient scenario runs indicated that the characteristic response time of the AOM biofilter is of the order of few decades (20-30 years), thus exceeding seasonal-interannual variability. Consequently, seasonal variation of bottom methane and sea water sulfates exert a negligible effect on methane escape through the sediment-water interface.

AOM generally acts as an efficient biofilter for upward migrating $CH_4$ under environmental conditions that are representative for the present-day ESAS with potentially important, yet unquantified implications for the Arctic ocean's alkalinity budget and,

thus, $CO_2$ fluxes. Our results thus suggest that previously published fluxes estimated from ESAS waters to atmosphere cannot be supported by non-turbulent methane efflux alone.

A regional upscaling of non-turbulent methane efflux for the Laptev Sea Shelf using a model-derived transfer function that relates sedimentation rate and methane efflux merely sums up to $\sim 0.1$ $GgCH_4$ yr$^{-1}$. Nevertheless, it also suggests that the evaluation of methane efflux from Siberian Shelf sediments should pay particular attention to the dynamic and rapidly

changing Arctic coastal areas close to big river mouths, as well as areas that may favor preferential methane gas release (*e.g.* rapidly eroding coastlines, fault lines or shallow sea floors, *i.e* <30 m). In addition, our findings call for more data concerning sedimentation and active fluid flow rates, as well as the reactivity of depositing organic matter and bioirrigation rates in Arctic shelf sediments.

In conclusion, we argue that the evaluation of projected subsea permafrost thaw and/or hydrate destabilization impacts on the

Arctic environment requires models that include an explicit description of 1) methane gas, 2) AOM biomass, as well as 3) the entire network of the most pertinent biogeochemical reactions. Such approaches, valid globally for all the shelves underlain by methane reservoirs (*e.g.* continental slopes), are even more recommended in order to enable a robust quantification of methane escape from the Arctic shelf to the Arctic ocean, settings even more sensible to the rapidly changing environmental conditions. Finally such refined modeling will also help evaluate the impact of subsea permafrost thaw and methane destabilization on

Arctic alkalinity and biogeochemical cycling.

*Code and data availability.* Primary data needed to reproduce the analyses presented in this study are archived by the MaxPlanck Institute for Meteorology are available upon request (publications@mpimet.mpg.de)





## Appendix A:  AOM efficiency $\eta$

If we identify the SMTZ region as the portion of the sediment column where the rate of AOM is 1% of the maximum, we can
define the efficiency of the AOM filter $\eta$ as

$$\eta(\%) = \left(1 - \frac{J^+_{CH4}}{J^-_{CH4}}\right) \cdot 100 \qquad (A1)$$

5   where $J^+_{CH4}$ is the methane flux at the shallowest point where the AOM rate is 1% of the maximum (upper dashed line in Fig.
A1), and $J^-_{CH4}$ is methane flux at the deepest point where the AOM rate is 1% of the maximum (lower dashed line in Fig. A1).

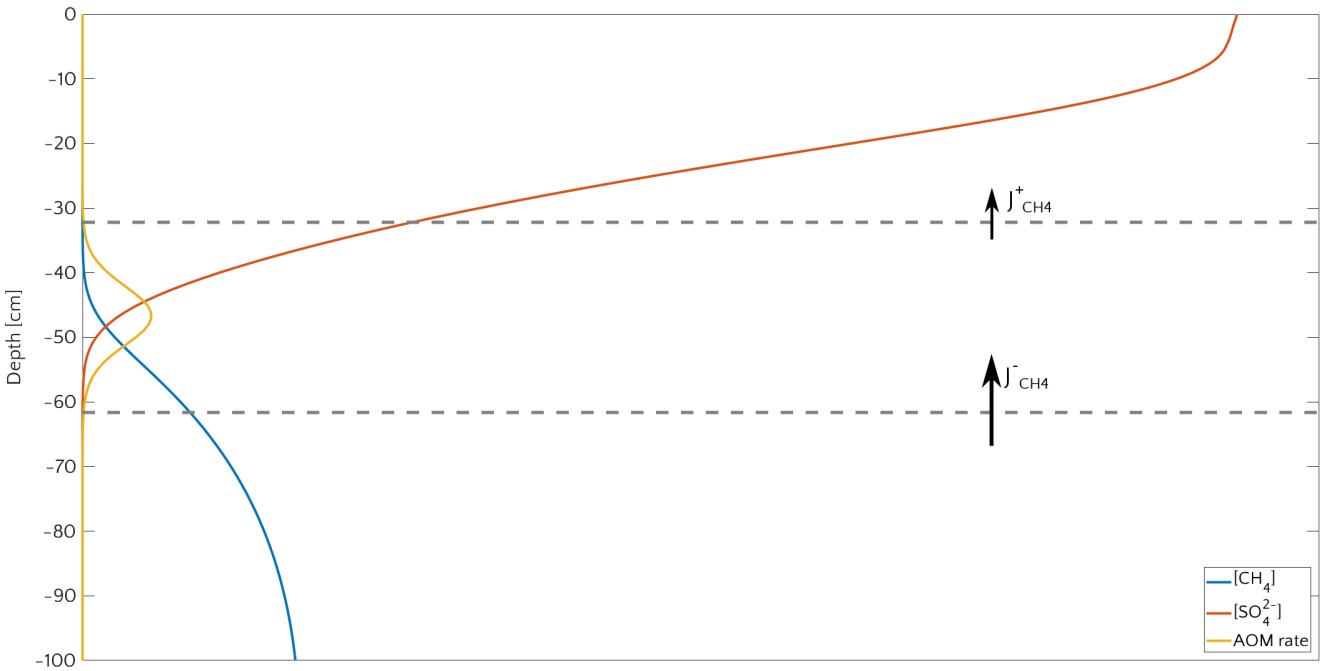

**Figure A1.** Typical sediment profile of $[SO_4^{2-}]$, $[CH_4]$ and AOM rate. Units are mM for concentration and mM yr$^{-1}$ for rate. The region
between the two dashed lines represents the zone where AOM rate is larger than 1% of it its maximum and defines the Sulfate Methane
Transition Zone (SMTZ). The fluxes $J^-_{CH4}$ and $J^+_{CH4}$ are the fluxes used in the definition of $\eta$ of eq. (A1).

## Appendix B:  Damköhler number

The Damköhler number $D_a$ is a dimensionless quantity which relates time scales typical of transport processes to time scales
typical of chemical reactions. It compares the consumption/production rate with the advective transport and is defined as

10   $D_a = \tau_T / \tau_R$ \hfill (B1)





where $\tau_T$ is the advective timescale and $\tau_R$ is the reaction timescale. $\tau_R$ is defined as $1/K_R$ where $K_R$ is the reaction rate of AOM or methanogenesis. If we call $R$ the reaction rate then $K_R$ reads:

$$K_R = \frac{1}{\mathcal{L}} \int_{\mathcal{L}} \frac{R}{[\mathrm{CH_4}]} dz \tag{B2}$$

where $\mathcal{L}$ is the width where the reaction rate is larger than 1% of the maximum rate. $\tau_T$ is instead defined as

$$5 \quad \tau_T = \frac{\mathcal{L}}{|v_{up} - \omega|} \tag{B3}$$

where $v_{up} - \omega$ is the effective advective velocity. $D_a$ can be the expressed by:

$$D_a = \frac{\tau_T}{\tau_R} = \frac{1}{|v_{up} - \omega|} \int_{\mathcal{L}} \frac{R}{[\mathrm{CH_4}]} dz. \tag{B4}$$

*Competing interests.* All contributing authors declare that no competing interests are present.

*Acknowledgements.* The research leading to these results has received funding from the European Union's Horizon 2020 research and
10   innovation programme under the Marie Skłodowska-Curie grant agreement No 643052 (C-CASCADES project).





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
