# Peer review of "Assessing the potential for non-turbulent methane escape from the East Siberian Arctic Shelf"

_Biogeosciences, 2019_

## Referee Comment (RC1) · Anonymous Referee #1 · 7 Sep 2019

Dear Editor, I read the manuscript by Puglini et al. you asked me to review for Biogeosciences. Hopefully, there are other reviewers that know more about modeling than I do, because I feel not 100% confident about judging part of this manuscript. Despite this (partly) mismatch, I found the manuscript well-outlined, clear and an overall pleasure to read. The manuscript focuses on non-turbulent methane escape from the East Siberian Arctic Shelf and evaluates the main physical and biogeochemical controls (for example organic matter quality, sedimentation rate, AOM rate...) on the efficiency of the biofilter that regulate non-turbulent methane emission. The authors conducted a comprehensive steady state sensitivity study through a of two baseline scenarios, active and passive case. Since the manuscript does not comprise the methane escape by bubble ebullition, a conclusion for overall gas escape on the East Siberian Arctic

Shelf cannot be ruled out but still, this is a very interesting contribution for the scientific community interested in this area and in methane related processes.

The manuscript address relevant scientific questions within the scope of Biogeosciences, and it present an interesting modelling approach to quantity the non-turbulent methane escape from the East Siberian Arctic Shelf. The conclusions are substantial, and the scientific methods and assumptions valid although a bit more technical while describing the modelling parameters. The results clearly support the the interpretations and conclusions of the manuscript. The manuscript cites the relevant paper then the number and quality of references seem to be appropriate. I have only few comments and I therefore suggest to accept this manuscript after moderate revisions.

General comments: The abstract is very long and contains too many information. Suggest to re-write it in a more concise way. The same comment is valid for the chapter 3.3.1 Window of opportunities, here there are interesting observations, but sometimes slightly verbose. The authors indicate that the active sediments are influenced by Âńa deep methane sourceÂż. then at the end of the paper they define that the deep methane source is ca 3 m below the seafloor, which is not exactly very deep. Would, it be possible to find another term instead of "deep"? In any case, this has to be better defined at the beginning of the manuscript. Specific comments: Page 2 Lines 17-18: "Under these conditions, permafrost aggraded on the shelf and was subsequently submersed when rising sea level flooded the shelf during the Holocene sea transgression (12 and 5 kyr BP)". Reference is needed.

Page 2 Line 19: explain what is "gas hydrate"

Page 2 Lines 29-30: "The increasing influx of warmer Atlantic water into the Arctic Ocean - the so-called Atlantification …". This term need to be explained and relevant papers need to be cited. In both "Zhang et al., 1998; Biastoch et al., 2011" the term Atlantification is not mentioned. Page 2 Line 2: what destabilize gas hydrate? Pressure changes or temperature increase? Or what? Page 4 Line 6: which are the "changes

in environmental conditions" mentioned here? Page 4 Line 12: for methane emissions and fractures, it might be useful to read a recently published paper in Biogeosciences "Yao et al., 2019". Biogeosciences, 16, 2221–2232, 2019

Rage 4 Line 19: What are the "passive and active sediments"? although there is some explanation later in the manuscript, these concepts need to be explained here, as soon as they are mentioned in the text. Page 6 Line 15: what about the anaerobic oxidation of methane? Page 9 Line 10: why the authors have assumed both baseline scenarios a water depth of 30 m when the average water depth of the ESAS is âĹij45 m (data from James et al., 2016)? Page 9 Line 28: is the trawling in the area affecting gas hydrate stability also? Is the gas hydrate close to the seafloor? Where is the real sediment depth? Which is the thickness of the sediments that is affected by trawling? Few cm or maybe 1 meter? Page 17 Line 13: "rapidely". To be corrected Page 23 lines 26-29: Would it be possible to better explain this concept here? I found very difficult to follow the reasoning here and related gas saturation concentration with precipitation of authigenic carbonate. Page 24 Line 28: Lena river and Moustakh Island in the Buor-Khaya Gulf need to be included in Figures and captions. As a general rule, all the locations that are mentioned in the main text need to be reported in location maps and relative captions. Page 26 Lines 16-17: The authors indicate that Additional physical reworking such as ice scouring or dredging, or the absence of bioirrigation, which is known to be patchy in Arctic sediments could even further reduce estimated methane effluxÂż. I would assume that these processes might enhance the methane fluxes instead since they remobilize sediments. More elaboration is needed here. Page 26 Line 25 ′: "Artic's". To be corrected Page 26 Line 26: How does it happen that "increasing sedimentation rates occur through coastal erosion"? please clarify. Page 28 Lines 33-34: "we show that methane from deep sources (ca. 3 m) reaches the sediment water interface within 7 to 20 years." A comment on the fact that 3 meters is considered deep has been previously reported. Page 29 Line 29: wording Âńwhich is in turn is determined". Chapter 3.3.1 this chapter is not very well organized and it is difficult to follow. Page 33 Lines 25-26: ÂńOn the ESAS, AOM is a transport-limited process

and transport parameters thus exert an important control on the efficiency of the AOM biofilter and, thus, on methane efflux". Please rewrite in a more clear way. Page 33 line27: what does "sedimentation and active fluid flow" in brackets mean respect the advective transport?

―――――――――――――――――

---

## Referee Comment (RC2) · Volker Brüchert (Referee) · 20 Sep 2019

Puglini et al comments I have a lot of respect for the sophisticated details of the diagenetic reaction-transport model BRNS described in the manuscript by Puglini et al. It is a sophisticated, well-established model framework and has been used in many important publications, not the least already in the sensitivity analysis of anaerobic oxidation of methane in many different marine settings. This study takes advantage of the long developmental work that has been done previously with respect to AOM with this model. Here it is used to simulate sediment methane cycling for one of the big hotspots for potential future marine methane emissions – the East Siberian shelf sea, with its potential for thawing submarine permafrost and the potential presence of gas hydrates (although the presence of both is often contested in the literature for good reasons).

[Figure]

The model uses the conventional setup of a network of biogeochemical reactions directly or indirectly coupled to the degradation of organic matter deposited at the sea floor. The paper is mostly not about the Siberian shelf, but is a very thorough assessment of AOM dynamics with explicit treatment of upward flow, bioenergetics controls of AOM, and a complex reaction network of biogeochemical redox reactions as they may occur in Siberian shelf sediment. The manuscript is well written up section 3.3.1., after which it deteriorates conspicuously. In principle, there were two objectives: 1. Broadscale simulation of AOM dynamics: It does a very good job at simulating a range of broadly set environmental conditions with direct impact on the filter efficiency of anaerobic methane-oxidizing microbial consortia that use methane and sulfate. The range of the environmental conditions is set broad enough to encompass conditions that may be encountered on the East Siberian shelf. However, this part is not very novel and AOM dynamics and filter efficiency have been reviewed by Regnier et al. (2011) previously. Therefore all sections of the manuscript that relate to the simulation tests should be significantly shortened. 2. Regional application: The second part of the manuscript is the application of the model to the East Siberian shelf. I found this part the more relevant one, given the title, but unfortunately also less well constrained due to the paucity of data used to constrain their model in face of the diversity and size of the targeted marine region. For reference, my guess is that the authors would certainly not model the whole of the North Sea or the Baltic Sea with this model, two marginal seas of similar size or even smaller than the Laptev Sea. My specific critique relates to the following points, which to my opinion are important in controlling the biogeochemical rates and flux output of the model, but that are not or too poorly constrained in the model to substantially further our understanding of how efficient anaerobic methane oxidation is and will be in the Siberian shelf sediments. • Even with the reduction of the investigated area to the Laptev Sea only, the depositional environments and geological settings are so much more variable that a simple sedimentation rate/bathymetry-based prediction of present-day organic carbon accumulation gives a starting condition for the model that is too simplifying to be acceptable. For example, the authors rely on a selected

handful of Pb-210 data (there are more available in the literature for better coverage (see Bröder et al., 201; Strobl et al., 1988) for sedimentation rates. The model doesn't consider the regionally diverse sediment types, permeabilities and rates in the Siberian Shelf Sea (see for example Dudarev et al., 2006 Oceanology; Rekant et al., 2015). The model doesn't consider known clay/sand/sand grain size variation and their influence of carbon concentration, permeability, transport, and resulting biogeochemical rates. • The model assumes Barents Sea depositional conditions as a good analog, however, these are unlike those of the Siberian shelf, since the Barents Sea is much deeper, has higher marine productivity, less ice cover, and much less input of terrestrial organic matter. In addition, it does not have terrestrial permafrost underneath the recent Holocene sediments. It is therefore not a particularly good analog. If the authors are interested I can provide porewater methane, sulfate and ammonium data from this region. • The reactive continuum approach employed here probably overestimates the reactive organic carbon amount that is available to organic carbon degradation at depth. In reality, the reactivity of the organic matter below the oxic horizons is one to two orders of magnitude lower than commonly observed in marine shelf sediments (see Figure 9, Brüchert et al., 2018). • The model doesn't consider Holocene sealevel change to elaborate on the mass of sediment available for methane generation since the last glacial maximum, which is the time since reactive sedimentary organic carbon accumulation began. Given the very low reactivity of carbon in these sediments (See Brüchert al., 2018; Bröder et al., 2916; Tesi et al., 2014), sulfate is likely never exhausted and methanogenesis and AOM may not even take place in these sediments at all. I am therefore not surprised at all that the authors arrive at such low regional dissolved benthic methane fluxes, seemingly at odds with the broadly published claims of extensive methane emission from the Siberian shelf. In fact, these fluxes confirm my own direct measurements of porewater methane concentrations and methane fluxes from a range of stations investigated in the summer of 2014 during the SWERUS expedition with the Swedish icebreaker Oden. If the authors are interested, I am willing to share these data with them to better constrain their model. • The model design relies on a sequence

of thermodynamically regulated terminal electron acceptor reactions driven by fresh carbon accumulation at the top of the model domain. In reality, non-biogenic or old Pre-Holocene-produced methane transport from below (of thermogenic or Pleistocene age, i.e., terrestrial) is the key unique characteristic of the Siberian shelf with respect to methane cycling. This carbon is old and uncoupled to recent carbon accumulation. In addition, carbon accumulation varied greatly through time on the Siberian shelf. The model appears to assume continuity of recent depositional conditions back in time and space, which is most certainly incorrect. Only the section with the transient model scenarios therefore applies to the Siberian shelf and only scenarios with an explicit upward flux of methane are relevant for investigating AOM dynamics in these sediments. However, because of the difficulties in constraining the regional distribution of seeps, flux rates cannot be reliably extrapolated and one should refrain from a regional flux estimate. My objections to the present manuscript are therefore not whether the model's capabilities are useful to the scientific community in general, which it certainly is, but a critique of the attempt to mimic biogeochemical as well as recent and past depositional conditions on the Siberian shelf to better predict sediment methane emissions from this region. I am fully aware of the infected discussion of the relevance of the Siberian shelf sea's role as a potentially huge methane source to the atmosphere put forward by Shakhova and co-authors. The outcome of the model simulations presented here, even in their most generous state (high advective upward flow and moderately to high sedimentation rates), would imply that the emissions proposed by Shakhova and co-authors are very hard to achieve without invoking massive gas emissions (which are not seen regionally in atmospheric measurements). However, the inability of this 1D model to encapsulate environmental conditions that are found in the Laptev and East Siberian Sea make it impossible to use its scaled model output to the current system or to use the model to make reliable assessments of how the shelf environment may change methane fluxes in the future. Particularly the latter requirement is key to the use of a reaction transport model such as this one in climate science. The authors may therefore consider a new title for their manuscript for the first section and resubmit

it under this new title without much reference to dissolved methane emissions on the East Siberian shelf, since this is not what they can model reasonably with the data they have available. The study and conclusions give the false impression that this particular model is capable, with certainty, to predict the non-gaseous methane flux emanating from this 1.5 million square kilometer large region, if one only knows the sedimentation rate and water depth. Alternatively, the model simulations can be tested with actual data from the Siberian shelf, which I am willing to share. In this case, I would suggest to reduce the first part of the manuscript and focus on the application of the BRNS to the Siberian shelf sea rather than a broad treatment of the model's performance.

Specific comments: See attached summary comment file.

Please also note the supplement to this comment:
https://www.biogeosciences-discuss.net/bg-2019-264/bg-2019-264-RC2-supplement.pdf

—————————————————

[Figure]

**Supplement:**

Author: VolkerB     Subject: Comment on Text     Date: 8/31/2019 2:56:19 PM

This is a crude overgeneralization. The authors must provide more references on the physical oceanography of the Laptev Sea and its sediment distribution and bathymetry to justify this comparison. The Norwegian setting has much higher primary productivity, is up to 8 times deeper and has substantially less ice cover over the year. If anything, the Vesterålen site shares very few similarities with the Laptev Sea or the East Siberian Shelf Sea.

Author: VolkerB     Subject: Comment on Text     Date: 8/31/2019 3:22:22 PM
please correct, not for methane
* * *
T Author: VolkerB      Subject: Comment on Text      Date: 8/31/2019 3:35:13 PM

It is not correct to make reference to the ESAS, since the range of the environmental conditions applied here is sufficiently broad to be applied to a wide range of shelf and slope margin settings with possible AOM. One condition worthwhile exploring and not done here is whether at low OM reactivities, the consumption of sulfate may not be completed for the time span of Holocene sediment accumulation on the ESAS (i.e., since ca 7000 years ago).
* * *
T Author: VolkerB      Subject: Comment on Text      Date: 8/31/2019 9:05:19 PM

Please correct to : 'to the SWI'

The model does not provide any constraint on the SWI flux, i.e., the benthic flux itself, because here other processes play an important that are modelled here.
* * *
T Author: VolkerB      Subject: Comment on Text      Date: 8/31/2019 9:42:52 PM

Referencing this study to other studies that show a range of 5 orders of magnitude in methane fluxes to justify its applicability seems odd. Please clarify how exactly each of the referenced studies supports the model findings in your simulations.
* * *
T Author: VolkerB      Subject: Comment on Text      Date: 8/31/2019 9:23:47 PM

which value was that? Not clear from the text. Apart from that, I deeply object to the use of one value to the whole of the ESAS. What is the purpose of this upscaled value? The original model value desn't gain any more legitimacy from upscaling and the fact that the upscaled value may be in the range of expected values neither. Please delete this section.

Author: VolkerB      Subject: Comment on Text      Date: 8/31/2019 9:28:45 PM

This is an interesting conclusion. How can one reconcile the observation that methane concentrations in the methanogenic zone  generally tend to increase with depth, i.e., their transport away from the zone of formation is too slow relative to the methanogenesis rate?

Author: VolkerB      Subject: Comment on Text      Date: 8/31/2019 8:49:59 PM

This is a curious assertion for the Siberian shelf system. It is wellknown that the sediments of the Siberian shelf are not reactive enough to yield significant methane. It is instead supposed that externally introduced methane from the thawing permafrost that serves as the methane source. The current model does not take external sources into account and this is the major flaw of this paper. It is actually not suited in the current version to model the processes on the Siberian shelf.

Author: VolkerB      Subject: Comment on Text      Date: 8/31/2019 9:56:14 PM

This introduction paragraph is rather wordy and doesn't say much. Can it be shortened?

Author: VolkerB      Subject: Comment on Text      Date: 8/31/2019 9:02:52 PM

Please provide a reference to the 'traditional views'. The view proposed here is not new.

Author: VolkerB     Subject: Comment on Text     Date: 8/31/2019 10:13:04 PM
what is meant by 'margin'?

T Author: VolkerB     Subject: Comment on Text     Date: 9/18/2019 12:23:03 PM
The authors should avoid trivial sentences such as this one.

Author: VolkerB    Subject: Comment on Text    Date: 9/18/2019 12:28:24 PM
I wonder whether the reactivity of organic matter in large parts of the SIberierian Shelf isn't even lower than 100 years. More 1000 years.

Author: VolkerB    Subject: Comment on Text    Date: 9/18/2019 5:09:58 PM
The authors are conflating to independent processes into one.

T Author: VolkerB     Subject: Comment on Text     Date: 9/18/2019 5:19:26 PM
These calculated actuve and passive fluxes are so low that they are empirically not verifiable with currently available measurement techniques.

Author: VolkerB     Subject: Comment on Text     Date: 9/18/2019 5:26:11 PM

The question is more, whether biogenic methane ever forms in these sediments, as the authors likeöly overestimate the reactivity of the organic matter. Altogether I think that the authors arrive at the right conclusion for the wrong reasons.

Author: VolkerB    Subject: Comment on Text    Date: 9/19/2019 9:50:28 AM

From this section on the manuscript becomes distinctly less well written, more typographic errors and less succinct writing. At the same time, the discussion of transient conditions is most relevant to the Siberian shelf system. This section needs to be carefully revised and improved in its writing.

Author: VolkerB    Subject: Comment on Text    Date: 9/19/2019 10:31:44 AM
A better way of explaining the discrepancy between the two methane fluxes at steady state and the transient condition would be to show the AOM rate for the two rate laws.

Author: VolkerB     Subject: Comment on Text     Date: 9/19/2019 10:34:45 AM

This is hard to understand. It should be possible to extract the instantaneous apparent kAOM value throughout the simulation. Ultimately of relevance is not what the kAOM is at the end of the simulation, but its time-integrated AOM rate throughout the modelled transient run.

Author: VolkerB     Subject: Comment on Text     Date: 9/19/2019 10:09:29 AM

Improve English. What do you mean here?

Author: VolkerB     Subject: Comment on Text     Date: 9/19/2019 10:11:39 AM

Poor English makes this paragraph hard to understand, most importantly it is not clear how the authors arrive at their conclusion with this argument.

Author: VolkerB     Subject: Comment on Text     Date: 9/19/2019 11:11:37 AM

thermodynamic

Author: VolkerB    Subject: Comment on Text    Date: 9/19/2019 11:14:08 AM

19 years

Author: VolkerB    Subject: Comment on Text    Date: 9/19/2019 11:16:29 AM

The role of sulfide was not mentioned previously. Is sulfide generally an important player for thermodynamic calculations done here?
* * *
T Author: VolkerB     Subject: Comment on Text     Date: 9/19/2019 11:20:50 AM

The wording should be reversed. An AOM biomass accounts for an AOM filter, not the other way round.
* * *
T Author: VolkerB     Subject: Comment on Text     Date: 9/19/2019 11:48:34 AM

Overall, this is irrelevant. The supply from below is what counts for the Siberian shelf, not the in-situ production, which is negligible in almost all settings except for the Eastern East Siberian Sea and the Chukchi Sea. In addition, the statement is also irrelevant in a general sense. As the supply from below is increased, so must the proportional contribution of in-situ produced methane decrease. This is not worth mentioning.
* * *
T Author: VolkerB     Subject: Comment on Text     Date: 9/19/2019 11:57:04 AM

typo here: from ... to..
* * *
T Author: VolkerB     Subject: Comment on Text     Date: 9/19/2019 11:34:59 AM

I am getting lost with the abbreviations
* * *
T Author: VolkerB     Subject: Comment on Text     Date: 9/19/2019 12:03:07 PM

As stated this is not true and must be corrected. Never did you investigate ESAS shelf sediments in this study. Modelling scenarios were investigated, of which some conditions may apply to selected environmental setting on the ESAS. The passive/active terminology strictly applies to theoretical scenarios of system behavior.
* * *
T Author: VolkerB     Subject: Comment on Text     Date: 9/19/2019 12:04:09 PM

Seriously, the authors have not investigated these sediments directly at all and should not make a claim to have investigate them.
* * *
T Author: VolkerB     Subject: Comment on Text     Date: 9/19/2019 12:04:50 PM

first or first-order?

---

## Author Comment (AC1) · 25 Nov 2019

**Response to Review n.1**

November 25, 2019

**General comments**

General comments: The abstract is very long and contains too many information. Suggest to re-write it in a more concise way. The same comment is valid for the chapter 3.3.1 Window of opportunities, here there are interesting observations, but sometimes slightly verbose. The authors indicate that the active sediments are influenced by "deep methane source", then at the end of the paper they define that the deep methane source is ca 3 m below the seafloor, which is not exactly very deep. Would it be possible to find another term instead of "deep"? In any case, this has to be better defined at the beginning of the manuscript

**Response**

We would like to thank the reviewer for the overall positive comment and suggestions. We will revise the abstract and the section 3.3.1 *Window of opportunity* for the final version of the paper.

In addition, we will also clarify the term "deep". We used the term "deep" to refer to methane sources below the simulated sediment column (*i.e.* > 3 m) not investigating the precise origin of this methane (permafrost/hydrates/thermogenic sources/in situ production) at the base of the sediment column (which could also come from even deeper depths). But we do agree that we must refer more clearly to the base of the sediment column.

**Specific comments**

1. Page 2 Lines 17-18: "Under these conditions, permafrost aggraded on the shelf and was subsequently submersed when rising sea level flooded the shelf during the Holocene sea transgression (12 and 5 kyr BP)". Reference is needed
   **Response: We added a reference to Romanovskii and Hubberten, 2001; Romanovskii, Hubberten, et al., 2005, for the thickness after submersion and Bauch et al., 2001 for the sea transgression.**

2. Page 2 Line 19: explain what is "gas hydrate"
   **Response: a state of matter in which a low molecular weight gas (like CH$_4$) is trapped in a "cage" of water molecules and whose structure is thermodynamically stable under specific temperature-pressure-salinity conditions that are found either in oceanic depths or beneath the permafrost (Sloan Jr et al., 2007). We will integrate a definition in the revised version of the manuscript.**.

3. Page 2 Lines 29-30: "The increasing influx of warmer Atlantic water into the Arctic Ocean - the so-called Atlantification". This term need to be explained and relevant papers need to be cited. In both "Zhang et al., 1998; Biastoch et al., 2011" the term Atlantification is not mentioned.
   **Response: the influence of warmer and saltier waters of Atlantic origins has been identified and brought up to the attention of the scientific community already in Biastoch et al., 2011; Carmack et al., 1995; Zhang et al., 1998, but the term "Atlantification" appears only in Polyakov et al., 2017 and Barton et al., 2018. These reference will be added in the revised version of the manuscript.**

4. "Page 2 Line 2: what destabilize gas hydrate? Pressure changes or temperature increase? Or what?"
   **Response: both pressure and temperature change are responsible of gas hydrates destabilization as reported in paragraph 3.3 of Shakhova, Semiletov, and Chuvilin, 2019. It has been suggested that in the case of subsea permafrost associated gas hydrates, temperature plays a more important role gas hydrate destabilization (Chuvilin et al., 2018; Makogon et al., 2007).**

5. Page 4 Line 6: which are the "changes in environmental condition" mentioned here?
   **Response: The transient change in lower CH$_4$ boundary conditions and, in case of the seasonal scenario n.2, also the change in the upper boundary conditions of SO$_4{}^{2-}$. We will clarify this point in the revised version of the manuscript.**

6. Page 4 Line 12: for methane emissions and fractures, it might be useful to read a recently published paper in Biogeosciences "Yao et al., 2019". Biogeosciences, 16, 2221-2232, 2019.
   **Response: Thanks for the suggestion. The recommend paper indeed supports our understanding of methane transport and biogeochemistry in fracture-affected sediments and we will add a reference to the revised version of the manuscript.**

7. Rage 4 Line 19: What are the "passive and active sediment"? Although there is some explanation later in the manuscript, these concepts need to be explained here, as soon as they are mentioned in the text.

Response: "Passive sediments" are sediments characterized by the absence of an advective water flow. In contrast, "active sediments" are subject to a non-zero water flow pointing upwards towards the sediment-water interface. The definition in the paper is reported at page 5, line 18-19. We will define these terms earlier in the revised version of the manuscript.

8. Page 6 Line 15: what about the anaerobic oxidation of methane?
Response: The aerobic and the anaerobic oxidation of methane have been regarded as secondary redox reaction, as they are not directly involved in the degradation of the organic matter. They are described in detail later on (page 6, line 32 and page 7).

9. Page 9 Line 10: why the authors have assumed both baseline scenarios a water depth of 30 m when the average water depth of the ESAS is ∼45 m (data from James et al., 2016)?
Response: mainly for two reasons:

   • We do not expect a large difference in the results between $30$ or $45$ meters, as well as if we had used $60$ m. The mechanisms we identify and the sensitivity we explore is expected to be largely unaffected by such small changes in the water depth. Results indicate that one of the main controls on non-turbulent methane escape is the sedimentation rate $\omega$. Applying the formulation of Burwicz et al., 2011, $\omega$ has basically the same value for $30$ m and $45$ m water depth. The only factor which is sensitive to water depth is the saturation value of methane ($[CH_4]^*$). At a water depth of $30$ m, $[CH_4]^* = 5.45$ $\mu$M as opposed to $\sim 10$ $\mu$M at $45$ m. This last value might increase even more the efficiency of the biofilter, leading in case simply to a reduction of the maximum $CH_4$ we identified.

   • The observed increase in summer temperature (Dmitrenko et al., 2011) occurs at shallower depths ($\sim 10$ m). We wanted to investigate even shallower shelves, as they are the ones expected to be more delicate and active from the biogeochemical point of view. For this reason we set a depth halfway between the average value of $45$ m (which takes into account also deeper depths, not really important for methane emissions) and shallower shelves closer to the coast.

10. Page 10 Line 28: is the trawling in the area affecting gas hydrate stability also? Is the gas hydrate close to the seafloor? Where is the real sediment depth? Which is the thickness of the sediments that is affected by trawling? Few cm or maybe 1 meter?
Response: On the Siberian shelf, gas hydrates are often associated with subsea permafrost (the so called subsea permafrost associated gas hydrates, Ruppel et al., 2017) and are located below the subsea

permafrost. Trawling can affect sediments: from centimeters to me­ters to a few meters (Shakhova, Semiletov, Gustafsson, et al., 2017) and, thus, is not expected to exert a significant effect on hydrate sta­bility. In any case, we do not simulate subsea permafrost thawing or hydrate destabilization explicitly, but rather explore the fate of plausi­ble methane fluxes from such deep sources and therefore do not make assumptions about release mechanisms and drivers.

11. Page 17 Line 13:"rapidely".
**Response: Thanks. Typo corrected**

12. Page 23 lines 26-29: Would it be possible to better explain this concept here? I found very difficult to follow the reasoning here and related gas saturation concentration with precipitation of authigenic carbonates.
**Response: Thanks. We will revise this section to clarify these aspects.**

13. Page 24 Line 28: Lena river and Moustakh Island in the Buor-Khaya Gulf need to be included in Figures and captions. As a general rule, all the locations that are mentioned in the main text need to be reported in location maps and relative captions.
**Response: The revised version of the manuscript will include a map reporting the mentioned locations.**

14. Page 26 Lines 16-17: The authors indicate that Additional physical reworking such as ice scouring or dredging, or the absence of bioirrigation, which is known to be patchy in Arctic sediments could even further reduce estimated methane efflux. I would assume that these processes might enhance the methane fluxes instead since they remobilize sediments. More elaboration is needed here.
**Response: The effects of non-local mixing processes are complex. They can indeed increase fluxes by enhancing transport through the sedi­ment. However, they can also reduce fluxes of methane (and other reduced species) by increasing the flux of oxygen and sulfate into the sediment. We will revise this section to clarify this point.**

15. Page 26 Line 25: "Artic's".
**Response: Thanks. Typo corrected**

16. How does it happen that "increasing sedimentation rates occur through coastal erosion"? please clarify.
**Response: Coastal erosion and the erosion of coastal ice complex pro­vide an input of debris and sediments which are sink rapidly to the sea floor (Vonk et al., 2014). Areas close to the coast are affected by coastal erosion and will thus receive a higher input of terrigeneous material.**

17. Page 28 Lines 33-34: "we show that methane from deep sources (ca. 3 m) reaches the sediment water interface within 7 to 20 years." A comment on the fact that

3 meters is considered deep has been previously reported.
**Response: see comment above**

18. Page 29 Line 29: wording "which is in turn is determined".
**Response: Thanks. Corrected.**

19. Chapter 3.3.1 this chapter is not very well organized and it is difficult to follow.
**Response: We will carefully revise this section.**

20. Page 33 Lines 25-26: "On the ESAS, AOM is a transport-limited process and transport parameters thus exert an important control on the efficiency of the AOM biofilter and, thus, on methane efflux". Please rewrite in a more clear way.
**Response: Since AOM is a transport-limited process, transport processes and parameters exert a dominant control on the efficiency of the AOM biofilter and, ultimately, on the methane efflux at the SWI. We will revise the section accordingly.**

21. Page 33 line27: what does "sedimentation and active fluid flow" in brackets mean respect the advective transport?
**Response: We simply list the two possible types of advective transport considered.**

**References**

Barton, Benjamin I, Yueng-Djern Lenn, and Camille Lique (2018). "Observed Atlantification of the Barents Sea causes the Polar Front to limit the expansion of winter sea ice". In: *Journal of Physical Oceanography* 48.8, pp. 1849–1866.

Bauch, Henning A, Thomas Mueller-Lupp, Ekaterina Taldenkova, Robert F Spielhagen, Heidemarie Kassens, Peter M Grootes, Jörn Thiede, J Heinemeier, and VV Petryashov (2001). "Chronology of the Holocene transgression at the North Siberian margin". In: *Global and Planetary Change* 31.1-4, pp. 125–139.

Biastoch, Arne, Tina Treude, Lars H Rüpke, Ulf Riebesell, Christina Roth, Ewa B Burwicz, Wonsun Park, Mojib Latif, Claus W Böning, Gurvan Madec, et al. (2011). "Rising Arctic Ocean temperatures cause gas hydrate destabilization and ocean acidification". In: *Geophysical Research Letters* 38.8.

Burwicz, Ewa B, LH Rüpke, and Klaus Wallmann (2011). "Estimation of the global amount of submarine gas hydrates formed via microbial methane formation based on numerical reaction-transport modeling and a novel parameterization of Holocene sedimentation". In: *Geochimica et Cosmochimica Acta* 75.16, pp. 4562–4576.

Carmack, Eddy C, Robie W Macdonald, Ronald G Perkin, Fiona A McLaughlin, and Richard J Pearson (1995). "Evidence for warming of Atlantic water in the southern Canadian Basin of the Arctic Ocean: Results from the Larsen-93 expedition". In: *Geophysical Research Letters* 22.9, pp. 1061–1064.

Chuvilin, Evgeny, Boris Bukhanov, Dinara Davletshina, Sergey Grebenkin, and Vladimir Istomin (2018). "Dissociation and self-preservation of gas hydrates in permafrost". In: *Geosciences (Switzerland)* 8.12. ISSN: 20763263. DOI: 10.3390/geosciences8120431. URL: https://www.mdpi.com/2076-3263/8/12/431.

Dmitrenko, Igor A, Sergey A Kirillov, L Bruno Tremblay, Heidemarie Kassens, Oleg A Anisimov, Sergey A Lavrov, Sergey O Razumov, and Mikhail N Grigoriev (2011). "Recent changes in shelf hydrography in the Siberian Arctic: Potential for subsea permafrost instability". In: *Journal of Geophysical Research: Oceans* 116.C10.

Makogon, Y. F., S. A. Holditch, and T. Y. Makogon (2007). "Natural gas-hydrates - A potential energy source for the 21st Century". In: *Journal of Petroleum Science and Engineering* 56.1-3, pp. 14–31. ISSN: 09204105. DOI: 10.1016/j.petrol.2005.10.009. URL: https://www.sciencedirect.com/science/article/pii/S0920410506001859.

Polyakov, Igor V, Andrey V Pnyushkov, Matthew B Alkire, Igor M Ashik, Till M Baumann, Eddy C Carmack, Ilona Goszczko, John Guthrie, Vladimir V Ivanov, Torsten Kanzow, et al. (2017). "Greater role for Atlantic inflows on sea-ice loss in the Eurasian Basin of the Arctic Ocean". In: *Science* 356.6335, pp. 285–291.

Romanovskii, Nikolai N and H-W Hubberten (2001). "Results of permafrost modelling of the lowlands and shelf of the Laptev Sea region, Russia". In: *Permafrost and periglacial processes* 12.2, pp. 191–202.

Romanovskii, Nikolai N, H-W Hubberten, AV Gavrilov, AA Eliseeva, and GS Tipenko (2005). "Offshore permafrost and gas hydrate stability zone on the shelf of East Siberian Seas". In: *Geo-marine letters* 25.2-3, pp. 167–182.

Ruppel, Carolyn D and John D Kessler (2017). "The interaction of climate change and methane hydrates". In: *Reviews of Geophysics* 55.1, pp. 126–168.

Shakhova, Natalia, Igor Semiletov, and Evgeny Chuvilin (2019). "Understanding the Permafrost–Hydrate System and Associated Methane Releases in the East Siberian Arctic Shelf". In: *Geosciences* 9.6, p. 251.

Shakhova, Natalia, Igor Semiletov, Orjan Gustafsson, Valentin Sergienko, Leopold Lobkovsky, Oleg Dudarev, Vladimir Tumskoy, Michael Grigoriev, Alexey Mazurov, Anatoly Salyuk, et al. (2017). "Current rates and mechanisms of subsea permafrost degradation in the East Siberian Arctic Shelf". In: *Nature communications* 8, p. 15872.

Sloan Jr, E Dendy and Carolyn Koh (2007). *Clathrate hydrates of natural gases.* CRC press.

Vonk, Jorien E, Igor P Semiletov, Oleg V Dudarev, Timothy I Eglinton, August Andersson, Natalia Shakhova, Alexander Charkin, Birgit Heim, and Örjan Gustafsson (2014). "Preferential burial of permafrost-derived organic carbon in S iberian-A rctic shelf waters". In: *Journal of Geophysical Research: Oceans* 119.12, pp. 8410–8421.

Zhang, Jinlun, D Andrew Rothrock, and Michael Steele (1998). "Warming of the Arctic Ocean by a strengthened Atlantic inflow: Model results". In: *Geophysical Research Letters* 25.10, pp. 1745–1748.

---

## Author Comment (AC2) · 25 Nov 2019

**Response to Review n.2: Volker Brüchert**

November 25, 2019

**General comment**

"I have a lot of respect for the sophisticated details of the diagenetic reaction-transport model BRNS described in the manuscript by Puglini et al. It is a sophisticated, well-established model framework and has been used in many important publications, not the least already in the sensitivity analysis of anaerobic oxidation of methane in many different marine settings. This study takes advantage of the long developmental work that has been done previously with respect to AOM with this model. Here it is used to simulate sediment methane cycling for one of the big hotspots for potential future marine methane emissions - the East Siberian shelf sea, with its potential for thawing submarine permafrost and the potential presence of gas hydrates (although the presence of both is often contested in the literature for good reasons)."

**Response: We would like to thank the reviewer for his appreciative, extremely constructive and insightful comment that not only sheds light on some critical aspects of our manuscript and helps to improve the quality of the manuscript, but also provides an opportunity to provide important clarifications and/or further detail.**

**Here we would like to stress that we included in the model a methane source from below (assuming different methane concentration spanning the range from $0$ to the saturation concentration) which is supposed to resemble any kind underlying source. Our focus is in the upper $3$ m of the sediments and we do not investigate and/or specify any explicit origin of the methane coming from below nor the model is, in such a version, sensitive to this origin. Since the area of interest is the ESAS, we hypothesize that subsea permafrost or gas hydrates may be the origin of such methane, but no results rely on this specific assumption. In fact we just wanted to stress the potential character of the non-turbulent methane emissions we found.**

"The model uses the conventional setup of a network of biogeochemical reactions directly or indirectly coupled to the degradation of organic matter deposited at the sea floor. The paper is mostly not about the Siberian shelf, but is a very thorough assessment of AOM dynamics with explicit treatment of upward flow, bioenergetics controls of AOM, and a complex reaction network of biogeochemical redox reactions

as they may occur in Siberian shelf sediment"

**Response: While the reviewer is absolutely right in pointing out that the results of the comprehensive sensitivity study described in the manuscript are universally valid, we would like to stress that the model setup and the sensitivity study have been specifically designed with the aim of assessing the fate of dissolved methane released from a deep source (*e.g.* dissociating hydrates or thawing subsea permafrost) in warming Siberian Shelf sediments. More specifically:**

- **The model is forced with a variable flux of dissolved methane potentially originating from dissociating methane hydrates and/or thawing permafrost in the deeper sediment. The methane flux is constrained by assuming lower model boundary methane concentrations ranging from $0$ to a maximum concentration that is constrained by the saturation of dissolved $CH_4$ under pressure, temperature and salinity conditions encountered on the Siberian shelf.**

- **All model boundary conditions, forcings and parameters (Tables S5 and S6) are chosen to be representative of environmental conditions encountered on the Siberian shelf.**

- **The range of boundary conditions and parameters tested in the steady state sensitivity study are constrained based on data compiled for the Siberian shelf.**

**As a consequence, the study presented here does not cover the entire range of possible conditions (*e.g.* methane fluxes, active fluid flow, organic carbon concentrations etc.) encountered at the global ocean seafloor, but is representative for conditions (likely) encountered on the present and future Siberian Shelf.**

"The manuscript is well written up section 3.3.1., after which it deteriorates conspicuously"

**Response: We agree that the logical structure of section 3.3.1 could be improved and have carefully revised this part.**

"In principle, there were two objectives: 1. Broadscale simulation of AOM dynamics: It does a very good job at simulating a range of broadly set environmental conditions with direct impact on the filter efficiency of anaerobic methane-oxidizing microbial consortia that use methane and sulfate. The range of the environmental conditions is set broad enough to encompass conditions that may be encountered on the East Siberian shelf. However, this part is not very novel and AOM dynamics and filter efficiency have been reviewed by Regnier et al. (2011) previously. Therefore all sections of the manuscript that relate to the simulation tests should be significantly shortened."

**Response: We strongly disagree with this comment. Regnier et al., 2011 present a comprehensive review of previously developed models that have**

been applied to investigate a large employed to simulate a large set of diverse depositional environments affected by intense methane cycling, ranging from mud volcanoes and active seeps to passive sediments experiencing groundwater discharge or high organic matter inputs. The review explicitly explores how different model implementations/formulations (with increasing complexity of the biogeochemical network) perform in simulating methane-affected sediments, as well as explore simulated AOM efficiency in response to a discrete, non-specific set of environmental conditions considered in these models.

However, the analysis of AOM filter efficiency and $CH_4$ effluxes presented has a completely different focus and goes well beyond the analysis presented in Regnier et al., 2011. As pointed out above, the main aim of this model study is to specifically investigate the potential escape of dissolved methane released from a deep source (*e.g.* dissociating hydrates or thawing subsea permafrost) from warming Siberian Shelf sediments. It thus assesses the efficiency of the microbial AOM filter in attenuating potential dissolved permafrost/hydrate methane fluxes under a continuous and specifically chosen range of environmental conditions/scenarios (likely) encountered on the present and (idealized) future Siberian shelf using an identical model set-up and thus offering not only more robust theoretical consistency and comparability. The main focus of the presented sensitivity analysis lies on identifying environmental conditions (and thus potential areas on the Siberian Shelf) that favor non-turbulent dissolved methane fluxes across the sediment-water interface.

We further emphasized this point in the manuscript by modifying the introduction and abstract accordingly.

"2. Regional application: The second part of the manuscript is the application of the model to the East Siberian shelf. I found this part the more relevant one, given the title, but unfortunately also less well constrained due to the paucity of data used to constrain their model in face of the diversity and size of the targeted marine region. For reference, my guess is that the authors would certainly not model the whole of the North Sea or the Baltic Sea with this model, two marginal seas of similar size or even smaller than the Laptev Sea"

Response: We also disagree with this statement. One strength of a models is that it can provide the explorative means to assess dynamics at spatial/temporal scales that cannot easily be assessed by observations alone. In particular, transfer functions, simple look-up tables and/or neural networks that are derived from or trained on a large ensemble of individual model simulations over a broad range of plausible boundary conditions have been frequently and successfully used to investigate regional and even global dynamics.

For instance, Gypens et al., 2008, Dale, Nickelsen, et al., 2015, Dale, Graco, et al., 2017, Capet et al., 2016 use simple transfer functions derived from a large ensemble of 1D diagenetic model simulations to predict benthic

nutrient recycling fluxes for the coastal North Sea (Gypens et al., 2008), the Peruvian Upwelling system (Dale, Graco, et al., 2017), the entire global ocean (Bohlen et al., 2012; Dale, Nickelsen, et al., 2015) or the entire Black Sea (Capet et al., 2016). Marquardt et al., 2010 used a transfer function to estimate the global gas hydrate inventory in marine sediments. In addition, Bourgeois et al., 2017 used a generalized additive model to calculate oxygen fluxes through the sediment-water interface for the entire Arctic Ocean and Artificial Neural Networks have been used to estimate sulfate (Bowles et al., 2014) fluxes through the sediment-water interface on a global scale.

These approaches are similar to the regional assessment presented here and illustrate the power of such transfer functions. We now highlight this in the introduction.

"My specific critique relates to the following points, which to my opinion are important in controlling the biogeochemical rates and flux output of the model, but that are not or too poorly constrained in the model to substantially further our understanding of how efficient anaerobic methane oxidation is and will be in the Siberian shelf sediments. Even with the reduction of the investigated area to the Laptev Sea only, the depositional environments and geological settings are so much more variable that a simple sedimentation rate/bathymetry-based prediction of present-day organic carbon accumulation gives a starting condition for the model that is too simplifying to be acceptable."

Response: The results of the extensive sensitivity study presented here clearly indicate the sedimentation rate and active fluid flow exert the dominant control on the escape of methane derived from thawing permafrost and/or disintegrating methane gas hydrates through the Siberian shelf sea floor across a wide range of contrasting environmental conditions encountered in this depositional environment. Results show that additional environmental conditions, such as OM content or AOM efficiency (*i.e.* $k_{AOM}$) play a minor or negligible role. Sedimentation rate can thus be used to predict the non-turbulent of methane escape on the Siberian Shelf.

The extensive sensitivity study presented here, thus also confirms the general approach that underlies the ensemble of studies listed in the previous response: single benthic biogeochemical characteristics, such as seafloor fluxes, redox horizons or inventories are often controlled by a limited set (1-2) of dominant factors that can then be used to robustly predict these characteristics on a regional/global scale.

"For example, the authors rely on a selected handful of Pb-210 data (there are more available in the literature for better coverage (see Bröder et al., 201; Strobl et al., 1988) for sedimentation rates"

Response: We thank the reviewer for the suggestions. Bröder et al., 2016 reports values for two sites in the East Siberian Sea and can thus unfortunately not be used to improve data coverage in the Laptev Sea. However, the reported linear sedimentation rate $(0.14 - 0.15 \text{ cm yr}^{-1})$ is not only similar to the sedimentation rate used in our local model application ($0.12$ cm yr$^{-1}$), but would also not change flux calculations if applied (see sensitivity study). We now include the values reported by Strobl et al., 1998. They show that sedimentation rate in the Laptev sea is of the same order ($0.15$ cm yr$^{-1}$)- a value that falls well in the range we explored.

"The model doesn't consider the regionally diverse sediment types, permeabilities and rates in the Siberian Shelf Sea (see for example Dudarev et al., 2006 Oceanology; Rekant et al., 2015). The model doesn't consider known clay/sand/sand grain size variation and their influence of carbon concentration, permeability, transport, and resulting biogeochemical rates."

Response: We would like to stress again that the presented study does account for the regional variability of sedimentation rate: 1) in the sensitivity study considering a large range spanning almost two orders of magnitude ($0.03 - 1.5$ cm yr$^{-1}$), and 2) in the regional analysis that applies a spatially variable sedimentation rate. In addition, the influence of the amount of degradable OM has also been tested in the sensitivity study and, because it is of secondary importance, is qualitatively discussed in the regional study.

It is however correct that we assume a porosity profile, which is representative for fine-grained shelf sediments. This is in agreement with Dudarev et al., 2006 (although they focus on the East Siberian Sea and not the Laptev Sea). They suggest that: "*The distribution of sediments demonstrates that they sustain fine-grained texture in the major part of the continental shelf regardless of the distance from the shore*". Considering that the overall geomorfological characteristics of the East Siberian Sea and Laptev Sea are similar, we can assume that a $3$ m sediment column with a prescribed porosity (dependent on depth) and a uniform texture and sediment type might be a decent representative for a large setting of the ESAS. We added a comment to the methods section.

"The model assumes Barents Sea depositional conditions as a good analog, however, these are unlike those of the Siberian shelf, since the Barents Sea is much deeper, has higher marine productivity, less ice cover, and much less input of terrestrial organic matter. In addition, it does not have terrestrial permafrost underneath the recent Holocene sediments. It is therefore not a particularly good analog. If the authors are interested, I can provide porewater methane, sulfate and ammonium data from this region."

Response: We would like to thank the reviewer for this offer. We have been in contact with the reviewer for porewater methane, sulfate and ammonium data and now include an additional model test case for this Laptev Sea site. We would however also like to stress that we do not consider the Barents Sea shelf offshore Versterålen as a good analog for the ESAS. Due to the paucity of observational data from the Laptev Sea for model testing, we used this Arctic site to illustrate the performance of our model set-up in simulating biogeochemical dynamics in high-latitude shelf sediments.

"The reactive continuum approach employed here probably overestimates the reactive organic carbon amount that is available to organic carbon degradation at depth. In reality, the reactivity of the organic matter below the oxic horizons is one to two orders of magnitude lower than commonly observed in marine shelf sediments (see Figure 9, Brüchert et al., 2018). Given the very low reactivity of carbon in these sediments (See Brüchert al., 2018; Bröder et al., 2016; Tesi et al., 2014), sulfate is likely never exhausted and methanogenesis and AOM may not even take place in these sediments at all. I am therefore not surprised at all that the authors arrive at such low regional dissolved benthic methane fluxes, seemingly at odds with the broadly published claims of extensive methane emission from the Siberian shelf."

**Response: This is a misunderstanding which we would like to clarify. First of all, we would also like to emphasize again that, according to our findings, the organic matter reactivity only exerts a secondary effect on our conclusions and therefore does not alter the overall picture of our results. In addition, we would like to stress again that the focus of the presented analysis centers on the fate of methane fluxes from thawing permafrost and/or disintegrating methane gas hydrates and not in-situ biogenically produced methane for which OM reactivity may play a more important role. The presence of a deep methane flux from thawing permafrost and/or disintegrating methane gas hydrates also ensures the presence of an AOM and the depletion of sulfates.**

**However, apart from this, we also disagree with the overall comment that the reactive continuum model (RCM) overestimates reactivity in these sediments. In fact, the RCM accounts for the decrease of OM reactivity with sediment depth/degradation state. Here, we test a wide range of RCM parametrizations (*i.e. a*) including those that result in a rapid decrease of OM reactivity by 1-2 orders of magnitude. Moreover the two papers cited actually support the use of a reactive-continuum model.**

1. **Bröder et al., 2016 show that the half-life of the organic matter deposited at two sites in the East Siberian Sea is $19 - 27$ yr. These half-life are represented by our RCM parametrizations in the intermediate range. Assuming $\nu = 0.125$ the corresponding a for the two samples would be $a = 3.4 - 4.8$ yr - values that are well within the range explored in our sensitivity analysis.**

2. **Tesi et al., 2014 in their conclusions clearly state: "*Therefore our results suggest that TerrOC is made of several allocthonous pools each with distinct reactivity toward the oxidation (i.e., reactive continuum)*".**

**We modified the method section to clarify this point and also added the two references.**

"In fact, these fluxes confirm my own direct measurements of porewater methane concentrations and methane fluxes from a range of stations investigated in the summer of 2014 during the SWERUS expedition with the Swedish icebreaker Oden. If the authors are interested, I am willing to share these data with them to better constrain their model."

**Response: We are really thankful for this offer and have been in contact with the reviewer.**

"The model doesn't consider Holocene sealevel change to elaborate on the mass of sediment available for methane generation since the last glacial maximum, which is the time since reactive sedimentary organic carbon accumulation began."

**Response: This is a misunderstanding. Again, the focus of the presented paper is on the fate of methane released from subsea permafrost/gas hydrates on the present-day and future Siberian shelf. We do not intend to simulate the historical evolution of the SSPF and of related historical methane emission, but only a plausible range of current/future ones. Furthermore, our model analysis is based on the simulation of the first $3$ meters of sediment and the Holocene sedimentation rates we explored ($0.03 - 1.5$ cm yr$^{-1}$) indicate that the sediment layer overlying the subsea permafrost always exceeds $3$ m.**

"The model design relies on a sequence of thermodynamically regulated terminal electron acceptor reactions driven by fresh carbon accumulation at the top of the model domain. In reality, non-biogenic or old Pre-Holocene-produced methane transport from below (of thermogenic or Pleistocene age, i.e., terrestrial) is the key unique characteristic of the Siberian shelf with respect to methane cycling. This carbon is old and uncoupled to recent carbon accumulation. In addition, carbon accumulation varied greatly through time on the Siberian shelf. The model appears to assume continuity of recent depositional conditions back in time and space, which is most certainly incorrect."

**Response: This is a misunderstanding. In fact, the model analysis focus on this "non-biogenic or old Pre-Holocene-produced methane transport from below (of thermogenic or Pleistocene age, i.e., terrestrial)" and not on the in-situ produced biogenic methane. Because it is impossible to reconstruct depositional conditions over the Holocene for the entire region, we indeed assume broadly similar depositional conditions during the Holocene. This is an acceptable simplification, in particular because:**

1. **Early diagenetic rates are highest in the shallow, young sediment layers and decrease rapidly with depth. As a consequence, biogeochemical dynamics are mostly affected by recent depositional conditions. This is especially true in the light of the fast decrease in OM reactivity reported by broder 2016; Brüchert et al., 2018; Tesi et al., 2014.**

2. **Our comprehensive sensitivity study indicates that OM degradation and biogenic methane production in the Holocene sediment layer ex-**

**erts a minor control on non-turbulent methane fluxes across the sediment-water interface. Holocene fluctuations in environmental conditions will thus exert a negligible effect on our results.**

**We clarify this throughout the manuscript (see previous replies).**

"Only the section with the transient model scenarios therefore applies to the Siberian shelf and only scenarios with an explicit upward flux of methane are relevant for investigating AOM dynamics in these sediments. However, because of the difficulties in constraining the regional distribution of seeps, flux rates cannot be reliably extrapolated and one should refrain from a regional flux estimate."

**Response: This is a misunderstanding. All steady-state simulations also apply an upward flux of methane (as outlined in the method section for details). They are thus relevant for investigating the fate of permafrost/hydrate derived methane in the Holocene sediment column and its possible escape through the sediment water interface. They also allow to derive the transfer function for possible non-turbulent methane escape that has been used to establish a regional estimate. We clarify this point throughout the manuscript (see previous replies).**

**Because our steady state analysis shows that AOM acts as an efficient biofilter and mostly prevents non-turbulent methane escape from the sediment, we also explored a number of plausible transient scenarios to explore if microbial dynamics could possibly create "windows of opportunity" for methane escape and assess their importance. We further clarify this in the introduction and method section. in the transient analysis we performed we actually refrained from an upscale estimate and we just explained the result of the flux out of simulated sediment column.**

"My objections to the present manuscript are therefore not whether the model's capabilities are useful to the scientific community in general, which it certainly is, but a critique of the attempt to mimic biogeochemical as well as recent and past depositional conditions on the Siberian shelf to better predict sediment methane emissions from this region."

**Response: see responses above.**

"I am fully aware of the infected discussion of the relevance of the Siberian shelf sea's role as a potentially huge methane source to the atmosphere put forward by Shakhova and co-authors. The outcome of the model simulations presented here, even in their most generous state (high advective upward flow and moderately to high sedimentation rates), would imply that the emissions proposed by Shakhova and coauthors are very hard to achieve without invoking massive gas emissions (which are not seen regionally in atmospheric measurements)."

**Response: This is indeed one of the conclusions of our analysis.**

"However, the inability of this 1D model to encapsulate environmental conditions

that are found in the Laptev and East Siberian Sea make it impossible to use its scaled model output to the current system or to use the model to make reliable assessments of how the shelf environment may change methane fluxes in the future. Particularly the latter requirement is key to the use of a reaction transport model such as this one in climate science. [...] The study and conclusions give the false impression that this particular model is capable, with certainty, to predict the non-gaseous methane flux emanating from this 1.5 million square kilometer large region, if one only knows the sedimentation rate and water depth. The authors may therefore consider a new title for their manuscript for the first section and resubmit it under this new title without much reference to dissolved methane emissions on the East Siberian shelf, since this is not what they can model reasonably with the data they have available. [...] Alternatively, the model simulations can be tested with actual data from the Siberian shelf, which I am willing to share. In this case, I would suggest to reduce the first part of the manuscript and focus on the application of the BRNS to the Siberian shelf sea rather than a broad treatment of the model's performance."

**Response: This comment reflects a string of misunderstandings. We do not aim at quantifying, "*with certainty*" the exact evolution of present and future methane emissions from the Siberian shelf. As highlighted in the title, abstract, introduction, the presented study assesses the potential for non-turbulent methane escape (derived from deep sediment sources such as permafrost/gas hydrates) from Siberian shelf sediments. As pointed out in the results and conclusion section, it thus provides a robust, quantitative framework suitable to make first order estimates and draw conclusions with respect to present and potential future emissions, as well as methane gas emissions required to support previous estimates of Arctic Ocean methane emissions to the atmosphere. Given the urgent need to assess this potentially ticking time bomb, but the paucity of observational data, it represents a feasible and robust quantitative first step towards a better assessment of the threat methane emissions from thawing subsea permafrost/ disintegrating methane hydrates pose for our climate.**

**Therefore, we are convinced that the title, as well as the approach of the presented study adequately reflect its scope and do not give a false impression. However, we have adapted the abstract, introduction, method and conclusion sections to further clarify these points. In addition, we have also included a new case study for the Laptev sea site based on the data provided by the reviewer.**

**Specific comments**

Page 8: "This is a crude overgeneralization. The authors must provide more references on the physical oceanography of the Laptev Sea and its sediment distribution and bathymetry to justify this comparison. The Norwegian setting has much higher primary productivity, is up to 8 times deeper and has substantially less ice cover over the year. If anything, the Vesterålen site shares very few similarities with the Laptev

Sea or the East Siberian Shelf Sea."

**Response: This is a misunderstanding. As pointed out in the response to general comments, we used the Hola trough sediments merely to assess the ability of the model to simulated carbon and sulfur dynamics in high latitude shelf sediments porewater profiles in a Northern shelf. No calibration of the BRNS or other following results relies on the simulations performed to reproduce the Vesterålen site, nor do we claim any similarity with the shelf areas of the East Siberian Arctic shelf. However, we do agree that our statement could be misunderstood and have now modified this section accordingly.**

Page 12: "Please correct, not for methane"

**Response: "Simulation results show an overall satisfactory agreement with measurements except for methane."**

Page 13:

- "It is not correct to make reference to the ESAS, since the range of the environmental conditions applied here is sufficiently broad to be applied to a wide range of shelf and slope margin settings with possible AOM. One condition worthwhile exploring and not done here is whether at low OM reactivities, the consumption of sulfate may not be completed for the time span of Holocene sediment accumulation on the ESAS (i.e., since ca 7000 years ago)."

  **This is a misunderstanding. As stated earlier, we investigate the fate of methane from deep sources (permafrost/hydrate) rather than in-situ produced methane (although the model also accounts for biogenic production in the Holocene sediment layer). As a consequence, we apply a range of methane fluxes from below that ensure a consumption of sulfate. With respect to the comment on the environmental conditions, we would like to repeat our response to a similar general comment here.**

  **"*While the reviewer is absolutely right in pointing out that the results of the comprehensive sensitivity study described in the manuscript are universally valid, we would like to stress that the model setup and the sensitivity study have been specifically designed with the aim of assessing the fate of dissolved methane released from a deep source (e.g. dissociating hydrates or thawing subsea permafrost) in warming Siberian Shelf sediments. More specifically:***

  - *The model is forced with a variable flux of dissolved methane potentially originating from dissociating methane hydrates and/or thawing permafrost in the deeper sediment. The methane flux is constrained by assuming lower model boundary methane concentrations ranging from $0$ to a maximum concentration that is constrained by the saturation of dissolved $CH_4$ under pressure,*

> > *temperature and salinity conditions encountered on the Siberian shelf.*

> > – *All model boundary conditions, forcings and parameters (Tables S5 and S6) are chosen to be representative of environmental conditions encountered on the Siberian shelf.*

> > – *The range of boundary conditions and parameters tested in the steady state sensitivity study are constrained based on data compiled for the Siberian shelf.*

> *As a consequence, the study presented here does not cover the entire range of possible conditions (e.g. methane fluxes, active fluid flow, organic carbon concentrations etc.) encountered at the global ocean seafloor, but is representative for conditions (likely) encountered on the present and future Siberian Shelf."*

- "Please correct to : 'to the SWI' The model does not provide any constraint on the SWI flux, i.e., the benthic flux itself, because here other processes play an important that are modelled here."

  **Response: We are not sure which processes the reviewer refers to, but in addition to diffusion and advection, the model explicitly accounts for bioturbation and non-local transport (through bioirrigation or ice scouring). It thus provides a robust representation of transport through the SWI.**

- "Referencing this study to other studies that show a range of 5 orders of magnitude in methane fluxes to justify its applicability seems odd. Please clarify how exactly each of the referenced studies supports the model findings in your simulation."

  **Response: The referenced studies offer a comparison with respect to the fluxes, as well as the flux variability in response to different environmental conditions we simulated.**

- "Which value was that? Not clear from the text. Apart from that, I deeply object to the use of one value to the whole of the ESAS. What is the purpose of this upscaled value? The original model value doesn't gain any more legitimacy from upscaling and the fact that the upscaled value may be in the range of expected values neither. Please delete this section"

  **The maximum value we found was $27.48\ \mu\mathrm{molCH_4\ cm^{-2}\ yr^{-1}}$. We added the exact value to the respective section. As pointed out in the earlier response, model results provide a robust quantitative framework to evaluate the potential for non-turbulent methane escape from the Siberian Shelf. The purpose of upscaling the maximum value to the ESAS is simply to offer an upper limit for this possible non-turbulent methane flux and show that, even if the most favorable conditions for methane escape were to be found over large shelf areas (note, this is**

**different from claiming that they are), non-turbulent methane fluxes would still be negligible and would not be able to support earlier estimates of methane emissions to the atmosphere.**

Page 14:

- "This is an interesting conclusion. How can one reconcile the observation that methane concentrations in the methanogenic zone generally tend to increase with depth, i.e., their transport away from the zone of formation is too slow relative to the methanogenesis rate?"
  **Response: The Damköhler numbers are defined in such a way that the transport process considered occurs in the same region as the reaction, *i.e.* we considered the methane transport within the methanogenic zone for the evaluation of $Da_{MG}$ and the SMTZ for the evaluation of the $Da_{AOM}$ . Simulation results reveal that methane transport is efficient within the methanogenetic zone. However, comparison with $Da_{AOM}$ shows that methane consumption within SMTZ is slower than its transport. In other words, methane can be efficiently transported to SMTZ but it is not quickly consumed there. As a consequence, methane accumulates below the SMTZ because at the SMTZ level it is not consumed and below the SMTZ no AOM occurs.**

- "This is a curious assertion for the Siberian shelf system. It is wellknown that the sediments of the Siberian shelf are not reactive enough to yield significant methane. It is instead supposed that externally introduced methane from the thawing permafrost that serves as the methane source. The current model does not take external sources into account and this is the major flaw of this paper. It is actually not suited in the current version to model the processes on the Siberian shelf."
  **Response: Deep (external) sources of methane are the main focus of the presented study. See response to general comments for details on biogenic methane production, methane fluxes from permafrost/hydrates.**

- "This introduction paragraph is rather wordy and doesn't say much. Can it be shortened?"
  **Response: we will shorten it in the finalized version of the paper, although we value the fact that an introduction might already provide the main message of what is described in detail later.**

- "Please provide a reference to the 'traditional views'. The view proposed here is not new."
  **Response: We replaced "traditional" with "intuitive". Our findings give further evidence of the dominant role of transport processes for non-turbulent methane effluxes also in modeling scenario compatible with ESAS settings.**

Page 15:"What is meant by 'margin'?"
**Response: the continental margin. We could replaced "margin" with "shelf" to avoid confusion.**

Page 17: "The authors should avoid trivial sentences such as this one."
**Response: it is not necessarily trivial, since a high methanogenesis might also be expected to foster a higher oxidation process and therefore accumulation of methane is not necessarily a triviality**

Page 19: "I wonder whether the reactivity of organic matter in large parts of the Siberian Shelf isn't even lower than 100 years. More 1000 years."
**Response: we also explored the $a \geq 100$ yr. As already stated in the reply to the general comment, the reactivity of the organic matter reported in other studies (*e.g.* Bröder et al., 2016) shows that $a$ is $< 5$, not far from the value $a = 10$ yr we used for the baseline simulation. In addition, $a$-values $>1000$ years are characteristic for deep sea sediments underlying extremely oligotrophic gyres, such as the deep South Pacific. Shelf, slope and most deep sea environments are generally characterized by $a < 1000$ years.**

Page 23: "The authors are conflating to independent processes into one."
**It is not clear which processes the reviewer refers to. We guess they are, on one hand, the actual AOM and, on the other hand, the precipitation of authigenic carbonate. We do not claim or mix them up and we are aware that they are two different processes but it is well established that they are not independent, since the alkalinity produced during the AOM can drive precipitation of authigenic carbonates as reported in many site all over the globe (*e.g.* Aloisi et al., 2004; Crémière, Lepland, Chand, Sahy, Condon, et al., 2016; Crémière, Lepland, Chand, Sahy, Kirsimäe, et al., 2016; Karaca et al., 2010; Luff et al., 2005; Meister et al., 2018; Pierre et al., 2012). We are simply hinting at an indirect effect supporting our findings, aware that the two processes are however well distinct and not trivially connected.**

Page 24: "These calculated active and passive fluxes are so low that they are empirically not verifiable with currently available measurement techniques."
**Response: We are aware of this limit and acknowledge it in the study. However, we would also like to point out that the exact quantity of these small fluxes is of minor importance. What is important here is that the potential for non-turbulent methane fluxes from Siberian Shelf sediments, even under the most favorable environmental conditions, is extremely limited and previous estimates of methane emissions to the atmosphere would thus require the build up of large quantities of methane gas.**

Page 26: "The question is more, whether biogenic methane ever forms in these sediments, as the authors likely overestimate the reactivity of the organic matter. Altogether I think that the authors arrive at the right conclusion for the wrong reasons."

**As stated previsouly, we disagree with this comment. Please see reply to general comment for details.**

Page 28: "From this section on the manuscript becomes distinctly less well written, more typographic errors and less succinct writing. At the same time, the discussion of transient conditions is most relevant to the Siberian shelf system. This section needs to be carefully revised and improved in its writing."
**Response: We will carefully revise and improve this section.**

Page 29: "A better way of explaining the discrepancy between the two methane fluxes at steady state and the transient condition would be to show the AOM rate for the two rate laws."
**Thanks for the suggestion. We add the AOM rate profile to fig. 11.b**

Page 31:

- "This is hard to understand. It should be possible to extract the instantaneous apparent kAOM value throughout the simulation. Ultimately of relevance is not what the kAOM is at the end of the simulation, but its time-integrated AOM rate throughout the modelled transient run."
  **It is actually possible to extract the $k_{AOM}$ at each simulated time step. However, here we wanted to explain why the final, new steady-state flux in the bioenergetic formulation is different from the simulation with the bimolecular formulation and that is the reason we focused on the final $k_{AOM}$, its shape and values.**

- "Poor English makes this paragraph hard to understand, most importantly it is not clear how the authors arrive at their conclusion with this argument"
  **Response: We will carefully revise and improve this section.**

- "thermodynamical"
  **Response: Corrected**

Page 32:

- "19 years"
  **Response: Corrected**

- "The role of sulfide was not mentioned previously. Is sulfide generally an important player for thermodynamic calculations done here?"
  **Sulfide influences AOM it appears in the formulation of $F_T$, which controls the AOM in the bioenergetic approach as shown in Eq. 11. Bicarbonate appears as well, but it is rarely a limiting factor.**

Page 33:

- "The wording should be reversed. An AOM biomass accounts for an AOM filter, not the other way round"

**Response: we agree but we wanted to stress that in order to have an efficient AOM filter a minimum AOM biomass is needed and this quantity has been estimated to be $> 10^{10}$ cells cm$^{-3}$, which is of the same order of magnitude as the value we found.**

- "Overall, this is irrelevant. The supply from below is what counts for the Siberian shelf, not the in-situ production, which is negligible in almost all settings except for the Eastern East Siberian Sea and the Chukchi Sea. In addition, the statement is also irrelevant in a general sense. As the supply from below is increased, so must the proportional contribution of in-situ produced methane decrease. This is not worth mentioning."
  **Response: We will edit this sentence accordingly in the final version of the paper.**

- "typo here: from ... to.."
  **Response: Corrected**

- "I am getting lost with the abbreviations"
  **$[CH_4]_-$ is the methane concentration at the bottom of the sediment column.**

- "As stated this is not true and must be corrected. Never did you investigate ESAS shelf sediments in this study. Modeling scenarios were investigated, of which some conditions may apply to selected environmental setting on the ESAS. The passive/active terminology strictly applies to theoretical scenarios of system behavior.[...] Seriously, the authors have not investigated these sediments directly at all and should not make a claim to have investigate them."
  **Response: This is a misunderstanding. The focus of this study is not a regional simulation of ESAS shelf sediments, but to develop a robust, quantitative framework that can be used to evaluate the potential for non-turbulent methane escape driven by thawing subsea permafrost and/or disintegrating methane gas hydrates on the warming Siberian shelf. We would again like to repeat our response to one of the general comments.**

  "*This comment reflects a string of misunderstandings. We do not aim at quantifying, "with certainty" the exact evolution of present and future methane emissions from the Siberian shelf. As highlighted in the title, abstract, introduction, the presented study assesses the potential for non-turbulent methane escape (derived from deep sediment sources such as permafrost/gas hydrates) from Siberian shelf sediments. As pointed out in the results and conclusion section, it thus provides a robust, quantitative framework suitable to make first order estimates and draw conclusions with respect to present and potential future emissions, as well as methane gas emissions required to support previous estimates of Arctic Ocean methane emissions to the atmosphere. Given the urgent need to assess this potentially ticking*

*time bomb, but the paucity of observational data, it represents a feasible and robust quantitative first step towards a better assessment of the threat methane emissions from thawing subsea permafrost/ disintegrating methane hydrates pose for our climate.*

*Therefore, we are convinced that the title, as well as the approach of the presented study adequately reflect its scope and do not give a false impression.*

*However, we also modified this section accordingly to avoid misunderstandings."*

- "first or first-order?"
  **Response: Actually both first and first-order. Modified accordingly.**

**References**

Aloisi, Giovanni, Klaus Wallmann, RR Haese, and J-F Saliege (2004). "Chemical, biological and hydrological controls on the 14C content of cold seep carbonate crusts: numerical modeling and implications for convection at cold seeps". In: *Chemical Geology* 213.4, pp. 359–383.

Bohlen, Lisa, Andrew W Dale, and Klaus Wallmann (2012). "Simple transfer functions for calculating benthic fixed nitrogen losses and C: N: P regeneration ratios in global biogeochemical models". In: *Global Biogeochemical Cycles* 26.3.

Bourgeois, Solveig, Philippe Archambault, and Ursula Witte (2017). "Organic matter remineralization in marine sediments: A Pan-Arctic synthesis". In: *Global Biogeochemical Cycles* 31.1, pp. 190–213.

Bowles, Marshall W, José M Mogollón, Sabine Kasten, Matthias Zabel, and Kai-Uwe Hinrichs (2014). "Global rates of marine sulfate reduction and implications for sub–sea-floor metabolic activities". In: *Science* 344.6186, pp. 889–891.

Bröder, Lisa, Tommaso Tesi, August Andersson, Timothy I Eglinton, Igor P Semiletov, Oleg V Dudarev, Per Roos, and Örjan Gustafsson (2016). "Historical records of organic matter supply and degradation status in the East Siberian Sea". In: *Organic Geochemistry* 91, pp. 16–30.

Brüchert, Volker, Lisa Bröder, Joanna E Sawicka, Tommaso Tesi, Samantha P Joye, Xiaole Sun, Igor P Semiletov, and Vladimir A Samarkin (2018). "Carbon mineralization in Laptev and East Siberian sea shelf and slope sediment". In: *Biogeosciences* 15.2, pp. 471–490.

Capet, Arthur, Filip JR Meysman, Ioanna Akoumianaki, Karline Soetaert, and Marilaure Grégoire (2016). "Integrating sediment biogeochemistry into 3D oceanic models: a study of benthic-pelagic coupling in the Black Sea". In: *Ocean Modelling* 101, pp. 83–100.

Crémière, Antoine, Aivo Lepland, Shyam Chand, Diana Sahy, Daniel J Condon, Stephen R Noble, Tonu Martma, Terje Thorsnes, Simone Sauer, and Harald Brunstad (2016). "Timescales of methane seepage on the Norwegian margin following collapse of the Scandinavian Ice Sheet". In: *Nature communications* 7, p. 11509.

Crémière, Antoine, Aivo Lepland, Shyam Chand, Diana Sahy, Kalle Kirsimäe, Michael Bau, Martin J Whitehouse, Stephen R Noble, Tõnu Martma, Terje Thorsnes, et al. (2016). "Fluid source and methane-related diagenetic processes recorded in cold seep carbonates from the Alvheim channel, central North Sea". In: *Chemical Geology* 432, pp. 16–33.

Dale, Andrew W, Michelle Graco, and Klaus Wallmann (2017). "Strong and dynamic benthic-pelagic coupling and feedbacks in a coastal upwelling system (Peruvian shelf)". In: *Frontiers in Marine Science* 4, p. 29.

Dale, Andrew W, Levin Nickelsen, Florian Scholz, Christian Hensen, Andreas Oschlies, and Klaus Wallmann (2015). "A revised global estimate of dissolved iron fluxes from marine sediments". In: *Global Biogeochemical Cycles* 29.5, pp. 691–707.

Dudarev, Oleg Victorovich, Igor Petrovich Semiletov, Alexander Nikolaevich Charkin, and AI Botsul (2006). "Deposition settings on the continental shelf of the East Siberian Sea". In: *Doklady earth sciences*. Vol. 409. 2. Springer, pp. 1000–1005.

Gypens, Nathalie, Christiane Lancelot, and Karline Soetaert (2008). "Simple parameterisations for describing N and P diagenetic processes: Application in the North Sea". In: *Progress in oceanography* 76.1, pp. 89–110.

Karaca, Deniz, Christian Hensen, and Klaus Wallmann (2010). "Controls on authigenic carbonate precipitation at cold seeps along the convergent margin off Costa Rica". In: *Geochemistry, Geophysics, Geosystems* 11.8.

Luff, Roger, Jens Greinert, Klaus Wallmann, Ingo Klaucke, and Erwin Suess (2005). "Simulation of long-term feedbacks from authigenic carbonate crust formation at cold vent sites". In: *Chemical Geology* 216.1-2, pp. 157–174.

Marquardt, Mathias, Christian Hensen, Elena Pinero, Klaus Wallmann, and Matthias Haeckel (2010). "A transfer function for the prediction of gas hydrate inventories in marine sediments". In: *Biogeosciences* 7.9, pp. 2925–2941.

Meister, Patrick, Johanna Wiedling, Christian Lott, Wolfgang Bach, Hanna Kuhfuß, Gunter Wegener, Michael E Böttcher, Christian Deusner, Anna Lichtschlag, Stefano M Bernasconi, et al. (2018). "Anaerobic methane oxidation inducing carbonate precipitation at abiogenic methane seeps in the Tuscan archipelago (Italy)". In: *PloS one* 13.12, e0207305.

Pierre, Catherine, Marie-Madeleine Blanc-Valleron, Jérôme Demange, Omar Boudouma, Jean-Paul Foucher, Thomas Pape, Tobias Himmler, Noemi Fekete, and Volkhard Spiess (2012). "Authigenic carbonates from active methane seeps offshore southwest Africa". In: *Geo-Marine Letters* 32.5-6, pp. 501–513.

Regnier, Pierre, Andy W Dale, S Arndt, DE LaRowe, J Mogollón, and P Van Cappellen (2011). "Quantitative analysis of anaerobic oxidation of methane (AOM) in marine sediments: a modeling perspective". In: *Earth-Science Reviews* 106.1-2, pp. 105–130.

Strobl, C, V Schulz, S Vogler, S Baumann, Heidemarie Kassens, PW Kubik, M Suter, and A Mangini (1998). "Determination of depositional beryllium-10 fluxes in the area of the Laptev Sea and beryllium-10 concentrations in water samples of high northern latitudes". In: *Land-Ocean Systems in the Siberian Arctic*. Springer, pp. 515–532.

Tesi, Tommaso, Igor Semiletov, Gustaf Hugelius, Oleg Dudarev, Peter Kuhry, and Örjan Gustafsson (2014). "Composition and fate of terrigenous organic matter along the Arctic land–ocean continuum in East Siberia: Insights from biomarkers and carbon isotopes". In: *Geochimica et Cosmochimica Acta* 133, pp. 235–256.

---

## Referee Report (RR1)

Puglini et al comments comments to revised version

- 1. The revised version has addressed the following points:
  - The authors now consider a larger number of dated sediment cores as the basis for the extrapolation for their sedimentation rate map. The new model still considers sedimentation rate to be the main variant and not variation in sediment type. While the model makes clear that sedimentation is one of the most important driver in regulating AOM, the authors should still make reference to the uncertainties arising from variable sedimentary environments, in particular in the near-shore where the emissions are expected to be highest.
  - Porewater data from the Laptev Sea are now taken as reference instead of data from the Barents Sea.
  - More consideration is given to studies that investigated the reactivity of organic matter on the Siberian shelf, e.g., Wild, Tesi, Brüchert, Bröder, etc.) than in the first version.
  - The revised version explicitly considers transient response due to upward flow from gas hydrates and thawing permafrost simulating various potential scenarios.
  - The authors use the same extrapolation of their data to the whole Laptev Sea that is based on a sedimentation rate map, but also calculate a Laptev Sea-wide transient seepage flux of 2.6 to 4.5Tg/yr-1 for the whole Laptev Sea without taking localized seepage into account. At least, this is my understanding from reading the section on the extrapolation. It is not clear, how the authors integrate the transient response into the spatial model.

Overall, this is a very good contribution and a substantial modification of the first manuscript with significant improvements. That said, the section on the model performance is still very extensive and not very closely related to the ESAS (although the authors have already moved a lot of material to the supplemental file). My recommendation would still be to further condense some of the sections 3.2. so that this part does not become too extensive before the direct application to the Siberian Sea is made. Please also address the small comments in the attached pdf file.

With best regards, Volker Brüchert

[revised manuscript text omitted]

(9)

15 with  $K_m^{SO_4^{2-}}$  half saturation constant of  $SO_4^{2-}$  and  $K_m^{CH_4}$  half saturation constant of  $CH_4$ , according to a typical Michaelis-Menten for enzymatically-catalyzed reactions.  $F_T$  represent the thermodynamic limitation and is given by

$$\begin{cases} 1 - \exp\left(\frac{\Delta G_r + \Delta G_{BQ}}{\chi RT}\right), & \text{if } \frac{\Delta G_r + \Delta G_{BQ}}{\chi RT} < 0\\ 0, & \text{if } \frac{\Delta G_r + \Delta G_{BQ}}{\chi RT} > 0 \end{cases}$$
(10)

where R is the gas constant, T is the absolute temperature,  $\chi$  is the average number of electrons transferred per reaction per mole of ATP produced (Jin and Bethke, 2005),  $\Delta G_r$  is the Gibbs free energy of the reaction and  $\Delta G_{BQ} = 20$  kJ (mol e-)-1 is the minimum energy needed to support synthesis of  $\sim \frac{1}{3} - \frac{1}{4}$  mol ATP (Dale et al., 2008c). In order to be thermodynamically favorable the total energy  $\Delta G_r + \Delta G_{PQ}$  has to be negative, meaning the that Gibbs free energy provided by the catabolic

favorable the total energy  $\Delta G_r + \Delta G_{BQ}$  has to be negative, meaning the that Gibbs free energy provided by the catabolic reaction is sufficient to sustain the microbial biomass growth.  $\Delta G_r$  is given by

$$\Delta G_r = \Delta G_r^0 + RT \ln \left( \gamma \frac{[\text{HS}^-] \cdot [\text{HCO}_3^-]}{[\text{CH}_4] \cdot [\text{SO}_4^{2^-}]} \right)$$
(11)

with  $\Delta G_r^0$ : standard free energy of the reaction, the second term: deviations from standard conditions (temperature and reaction quotient) on Gibbs free energy and  $\gamma$ : a parameter representing departure from ideal beahviour.

The link between substrate consumption and microbial growth (anabolism) is given by Dale et al. (2006):

$$13.8SD \cdot SO_4^{2-} + 14.3SD \cdot CH_4 + 0.2SD \cdot NH_4^+ + 0.3SD \cdot H^+ \rightarrow 0.2B + 13.3SD \cdot HCO_3^- + 13.8SD \cdot HS^-$$
(12)

[revised manuscript text omitted]

---

## Author Response (AR3)

**Response to Review n.1**

November 25, 2019

**General comments**

General comments: The abstract is very long and contains too many information. Suggest to re-write it in a more concise way. The same comment is valid for the chapter 3.3.1 Window of opportunities, here there are interesting observations, but sometimes slightly verbose. The authors indicate that the active sediments are influenced by "deep methane source", then at the end of the paper they define that the deep methane source is ca 3 m below the seafloor, which is not exactly very deep. Would it be possible to find another term instead of "deep"? In any case, this has to be better defined at the beginning of the manuscript

**Response**

We would like to thank the reviewer for the overall positive comment and suggestions. We will revise the abstract and the section 3.3.1 *Window of opportunity* for the final version of the paper.

In addition, we will also clarify the term "deep". We used the term "deep" to refer to methane sources below the simulated sediment column (*i.e.* $> 3$ m) not investigating the precise origin of this methane (permafrost/hydrates/thermogenic sources/in situ production) at the base of the sediment column (which could also come from even deeper depths). But we do agree that we must refer more clearly to the base of the sediment column.

**Specific comments**

1. Page 2 Lines 17-18: "Under these conditions, permafrost aggraded on the shelf and was subsequently submersed when rising sea level flooded the shelf during the Holocene sea transgression (12 and 5 kyr BP)". Reference is needed
   **Response: We added a reference to Romanovskii and Hubberten, 2001; Romanovskii, Hubberten, et al., 2005, for the thickness after submersion and Bauch et al., 2001 for the sea transgression.**

2. Page 2 Line 19: explain what is"gas hydrate"
   **Response: a state of matter in which a low molecular weight gas (like $CH_4$) is trapped in a "cage" of water molecules and whose structure is thermodynamically stable under specific temperature-pressure-salinity conditions that are found either in oceanic depths or beneath the permafrost (Sloan Jr et al., 2007). We will integrate a definition in the revised version of the manuscript..**

3. Page 2 Lines 29-30:"The increasing influx of warmer Atlantic water into the Arctic Ocean - the so-called Atlantification". This term need to be explained and relevant papers need to be cited. In both "Zhang et al., 1998; Biastoch et al., 2011" the term Atlantification is not mentioned.
   **Response: the influence of warmer and saltier waters of Atlantic origins has been identified and brought up to the attention of the scientific community already in Biastoch et al., 2011; Carmack et al., 1995; Zhang et al., 1998, but the term "Atlantification" appears only in Polyakov et al., 2017 and Barton et al., 2018. These reference will be added in the revised version of the manuscript.**

4. "Page 2 Line 2: what destabilize gas hydrate? Pressure changes or temperature increase? Or what?"
   **Response: both pressure and temperature change are responsible of gas hydrates destabilization as reported in paragraph 3.3 of Shakhova, Semiletov, and Chuvilin, 2019. It has been suggested that in the case of subsea permafrost associated gas hydrates, temperature plays a more important role gas hydrate destabilization (Chuvilin et al., 2018; Makogon et al., 2007).**

5. Page 4 Line 6: which are the"changes in environmental condition" mentioned here?
   **Response: The transient change in lower $CH_4$ boundary conditions and, in case of the seasonal scenario n.2, also the change in the upper boundary conditions of $SO_4{}^{2-}$. We will clarify this point in the revised version of the manuscript.**

6. Page 4 Line 12: for methane emissions and fractures, it might be useful to read a recently published paper in Biogeosciences "Yao et al., 2019". Biogeosciences, 16, 2221-2232, 2019.
   **Response: Thanks for the suggestion. The recommend paper indeed supports our understanding of methane transport and biogeochemistry in fracture-affected sediments and we will add a reference to the revised version of the manuscript.**

7. Rage 4 Line 19: What are the "passive and active sediment"? Although there is some explanation later in the manuscript, these concepts need to be explained here, as soon as they are mentioned in the text.

Response: "Passive sediments" are sediments characterized by the absence of an advective water flow. In contrast, "active sediments" are subject to a non-zero water flow pointing upwards towards the sediment-water interface. The definition in the paper is reported at page 5, line 18-19. We will define these terms earlier in the revised version of the manuscript.

8. Page 6 Line 15: what about the anaerobic oxidation of methane?
   Response: The aerobic and the anaerobic oxidation of methane have been regarded as secondary redox reaction, as they are not directly involved in the degradation of the organic matter. They are described in detail later on (page 6, line 32 and page 7).

9. Page 9 Line 10: why the authors have assumed both baseline scenarios a water depth of 30 m when the average water depth of the ESAS is ∼45 m (data from James et al., 2016)?
   Response: mainly for two reasons:

   - We do not expect a large difference in the results between 30 or 45 meters, as well as if we had used 60 m. The mechanisms we identify and the sensitivity we explore is expected to be largely unaffected by such small changes in the water depth. Results indicate that one of the main controls on non-turbulent methane escape is the sedimentation rate $\omega$. Applying the formulation of Burwicz et al., 2011, $\omega$ has basically the same value for 30 m and 45 m water depth. The only factor which is sensitive to water depth is the saturation value of methane ($[CH_4]^*$). At a water depth of 30 m, $[CH_4]^* = 5.45$ $\mu$M as opposed to $\sim 10$ $\mu$M at 45 m. This last value might increase even more the efficiency of the biofilter, leading in case simply to a reduction of the maximum $CH_4$ we identified.

   - The observed increase in summer temperature (Dmitrenko et al., 2011) occurs at shallower depths ($\sim 10$ m). We wanted to investigate even shallower shelves, as they are the ones expected to be more delicate and active from the biogeochemical point of view. For this reason we set a depth halfway between the average value of 45 m (which takes into account also deeper depths, not really important for methane emissions) and shallower shelves closer to the coast.

10. Page 10 Line 28: is the trawling in the area affecting gas hydrate stability also? Is the gas hydrate close to the seafloor? Where is the real sediment depth? Which is the thickness of the sediments that is affected by trawling? Few cm or maybe 1 meter?
    Response: On the Siberian shelf, gas hydrates are often associated with subsea permafrost (the so called subsea permafrost associated gas hydrates, Ruppel et al., 2017) and are located below the subsea

permafrost. Trawling can affect sediments: from centimeters to meters to a few meters (Shakhova, Semiletov, Gustafsson, et al., 2017) and, thus, is not expected to exert a significant effect on hydrate stability. In any case, we do not simulate subsea permafrost thawing or hydrate destabilization explicitly, but rather explore the fate of plausible methane fluxes from such deep sources and therefore do not make assumptions about release mechanisms and drivers.

11. Page 17 Line 13:"rapidely".
    **Response: Thanks. Typo corrected**

12. Page 23 lines 26-29: Would it be possible to better explain this concept here? I found very difficult to follow the reasoning here and related gas saturation concentration with precipitation of authigenic carbonates.
    **Response: Thanks. We will revise this section to clarify these aspects.**

13. Page 24 Line 28: Lena river and Moustakh Island in the Buor-Khaya Gulf need to be included in Figures and captions. As a general rule, all the locations that are mentioned in the main text need to be reported in location maps and relative captions.
    **Response: The revised version of the manuscript will include a map reporting the mentioned locations.**

14. Page 26 Lines 16-17: The authors indicate that Additional physical reworking such as ice scouring or dredging, or the absence of bioirrigation, which is known to be patchy in Arctic sediments could even further reduce estimated methane efflux. I would assume that these processes might enhance the methane fluxes instead since they remobilize sediments. More elaboration is needed here.
    **Response: The effects of non-local mixing processes are complex. They can indeed increase fluxes by enhancing transport through the sediment. However, they can also reduce fluxes of methane (and other reduced species) by increasing the flux of oxygen and sulfate into the sediment. We will revise this section to clarify this point.**

15. Page 26 Line 25: "Artic's".
    **Response: Thanks. Typo corrected**

16. How does it happen that "increasing sedimentation rates occur through coastal erosion"? please clarify.
    **Response: Coastal erosion and the erosion of coastal ice complex provide an input of debris and sediments which are sink rapidly to the sea floor (Vonk et al., 2014). Areas close to the coast are affected by coastal erosion and will thus receive a higher input of terrigeneous material.**

17. Page 28 Lines 33-34: "we show that methane from deep sources (ca. 3 m) reaches the sediment water interface within 7 to 20 years." A comment on the fact that

3 meters is considered deep has been previously reported.
**Response: see comment above**

18. Page 29 Line 29: wording "which is in turn is determined".
    **Response: Thanks. Corrected.**

19. Chapter 3.3.1 this chapter is not very well organized and it is difficult to follow.
    **Response: We will carefully revise this section.**

20. Page 33 Lines 25-26: "On the ESAS, AOM is a transport-limited process and transport parameters thus exert an important control on the efficiency of the AOM biofilter and, thus, on methane efflux". Please rewrite in a more clear way.
    **Response: Since AOM is a transport-limited process, transport processes and parameters exert a dominant control on the efficiency of the AOM biofilter and, ultimately, on the methane efflux at the SWI. We will revise the section accordingly.**

21. Page 33 line27: what does "sedimentation and active fluid flow" in brackets mean respect the advective transport?
    **Response: We simply list the two possible types of advective transport considered.**

**References**

Barton, Benjamin I, Yueng-Djern Lenn, and Camille Lique (2018). "Observed Atlantification of the Barents Sea causes the Polar Front to limit the expansion of winter sea ice". In: *Journal of Physical Oceanography* 48.8, pp. 1849–1866.

Bauch, Henning A, Thomas Mueller-Lupp, Ekaterina Taldenkova, Robert F Spielhagen, Heidemarie Kassens, Peter M Grootes, Jörn Thiede, J Heinemeier, and VV Petryashov (2001). "Chronology of the Holocene transgression at the North Siberian margin". In: *Global and Planetary Change* 31.1-4, pp. 125–139.

Biastoch, Arne, Tina Treude, Lars H Rüpke, Ulf Riebesell, Christina Roth, Ewa B Burwicz, Wonsun Park, Mojib Latif, Claus W Böning, Gurvan Madec, et al. (2011). "Rising Arctic Ocean temperatures cause gas hydrate destabilization and ocean acidification". In: *Geophysical Research Letters* 38.8.

Burwicz, Ewa B, LH Rüpke, and Klaus Wallmann (2011). "Estimation of the global amount of submarine gas hydrates formed via microbial methane formation based on numerical reaction-transport modeling and a novel parameterization of Holocene sedimentation". In: *Geochimica et Cosmochimica Acta* 75.16, pp. 4562–4576.

Carmack, Eddy C, Robie W Macdonald, Ronald G Perkin, Fiona A McLaughlin, and Richard J Pearson (1995). "Evidence for warming of Atlantic water in the southern Canadian Basin of the Arctic Ocean: Results from the Larsen-93 expedition". In: *Geophysical Research Letters* 22.9, pp. 1061–1064.

Chuvilin, Evgeny, Boris Bukhanov, Dinara Davletshina, Sergey Grebenkin, and Vladimir Istomin (2018). "Dissociation and self-preservation of gas hydrates in permafrost". In: *Geosciences (Switzerland)* 8.12. ISSN: 20763263. DOI: 10.3390/geosciences8120431. URL: https://www.mdpi.com/2076-3263/8/12/431.

Dmitrenko, Igor A, Sergey A Kirillov, L Bruno Tremblay, Heidemarie Kassens, Oleg A Anisimov, Sergey A Lavrov, Sergey O Razumov, and Mikhail N Grigoriev (2011). "Recent changes in shelf hydrography in the Siberian Arctic: Potential for subsea permafrost instability". In: *Journal of Geophysical Research: Oceans* 116.C10.

Makogon, Y. F., S. A. Holditch, and T. Y. Makogon (2007). "Natural gas-hydrates - A potential energy source for the 21st Century". In: *Journal of Petroleum Science and Engineering* 56.1-3, pp. 14–31. ISSN: 09204105. DOI: 10.1016/j.petrol.2005.10.009. URL: https://www.sciencedirect.com/science/article/pii/S0920410506001859.

Polyakov, Igor V, Andrey V Pnyushkov, Matthew B Alkire, Igor M Ashik, Till M Baumann, Eddy C Carmack, Ilona Goszczko, John Guthrie, Vladimir V Ivanov, Torsten Kanzow, et al. (2017). "Greater role for Atlantic inflows on sea-ice loss in the Eurasian Basin of the Arctic Ocean". In: *Science* 356.6335, pp. 285–291.

Romanovskii, Nikolai N and H-W Hubberten (2001). "Results of permafrost modelling of the lowlands and shelf of the Laptev Sea region, Russia". In: *Permafrost and periglacial processes* 12.2, pp. 191–202.

Romanovskii, Nikolai N, H-W Hubberten, AV Gavrilov, AA Eliseeva, and GS Tipenko (2005). "Offshore permafrost and gas hydrate stability zone on the shelf of East Siberian Seas". In: *Geo-marine letters* 25.2-3, pp. 167–182.

Ruppel, Carolyn D and John D Kessler (2017). "The interaction of climate change and methane hydrates". In: *Reviews of Geophysics* 55.1, pp. 126–168.

Shakhova, Natalia, Igor Semiletov, and Evgeny Chuvilin (2019). "Understanding the Permafrost–Hydrate System and Associated Methane Releases in the East Siberian Arctic Shelf". In: *Geosciences* 9.6, p. 251.

Shakhova, Natalia, Igor Semiletov, Orjan Gustafsson, Valentin Sergienko, Leopold Lobkovsky, Oleg Dudarev, Vladimir Tumskoy, Michael Grigoriev, Alexey Mazurov, Anatoly Salyuk, et al. (2017). "Current rates and mechanisms of subsea permafrost degradation in the East Siberian Arctic Shelf". In: *Nature communications* 8, p. 15872.

Sloan Jr, E Dendy and Carolyn Koh (2007). *Clathrate hydrates of natural gases.* CRC press.

Vonk, Jorien E, Igor P Semiletov, Oleg V Dudarev, Timothy I Eglinton, August Andersson, Natalia Shakhova, Alexander Charkin, Birgit Heim, and Örjan Gustafsson (2014). "Preferential burial of permafrost-derived organic carbon in S iberian-A rctic shelf waters". In: *Journal of Geophysical Research: Oceans* 119.12, pp. 8410–8421.

Zhang, Jinlun, D Andrew Rothrock, and Michael Steele (1998). "Warming of the Arctic Ocean by a strengthened Atlantic inflow: Model results". In: *Geophysical Research Letters* 25.10, pp. 1745–1748.

**Response to Review n.2: Volker Brüchert**

November 25, 2019

**General comment**

"I have a lot of respect for the sophisticated details of the diagenetic reaction-transport model BRNS described in the manuscript by Puglini et al. It is a sophisticated, well-established model framework and has been used in many important publications, not the least already in the sensitivity analysis of anaerobic oxidation of methane in many different marine settings. This study takes advantage of the long developmental work that has been done previously with respect to AOM with this model. Here it is used to simulate sediment methane cycling for one of the big hotspots for potential future marine methane emissions - the East Siberian shelf sea, with its potential for thawing submarine permafrost and the potential presence of gas hydrates (although the presence of both is often contested in the literature for good reasons)."

**Response: We would like to thank the reviewer for his appreciative, extremely constructive and insightful comment that not only sheds light on some critical aspects of our manuscript and helps to improve the quality of the manuscript, but also provides an opportunity to provide important clarifications and/or further detail.**

**Here we would like to stress that we included in the model a methane source from below (assuming different methane concentration spanning the range from $0$ to the saturation concentration) which is supposed to resemble any kind underlying source. Our focus is in the upper $3$ m of the sediments and we do not investigate and/or specify any explicit origin of the methane coming from below nor the model is, in such a version, sensitive to this origin. Since the area of interest is the ESAS, we hypothesize that subsea permafrost or gas hydrates may be the origin of such methane, but no results rely on this specific assumption. In fact we just wanted to stress the potential character of the non-turbulent methane emissions we found.**

"The model uses the conventional setup of a network of biogeochemical reactions directly or indirectly coupled to the degradation of organic matter deposited at the sea floor. The paper is mostly not about the Siberian shelf, but is a very thorough assessment of AOM dynamics with explicit treatment of upward flow, bioenergetics controls of AOM, and a complex reaction network of biogeochemical redox reactions

as they may occur in Siberian shelf sediment"

**Response: While the reviewer is absolutely right in pointing out that the results of the comprehensive sensitivity study described in the manuscript are universally valid, we would like to stress that the model setup and the sensitivity study have been specifically designed with the aim of assessing the fate of dissolved methane released from a deep source (*e.g.* dissociating hydrates or thawing subsea permafrost) in warming Siberian Shelf sediments. More specifically:**

- **The model is forced with a variable flux of dissolved methane potentially originating from dissociating methane hydrates and/or thawing permafrost in the deeper sediment. The methane flux is constrained by assuming lower model boundary methane concentrations ranging from $0$ to a maximum concentration that is constrained by the saturation of dissolved $CH_4$ under pressure, temperature and salinity conditions encountered on the Siberian shelf.**

- **All model boundary conditions, forcings and parameters (Tables S5 and S6) are chosen to be representative of environmental conditions encountered on the Siberian shelf.**

- **The range of boundary conditions and parameters tested in the steady state sensitivity study are constrained based on data compiled for the Siberian shelf.**

**As a consequence, the study presented here does not cover the entire range of possible conditions (*e.g.* methane fluxes, active fluid flow, organic carbon concentrations etc.) encountered at the global ocean seafloor, but is representative for conditions (likely) encountered on the present and future Siberian Shelf.**

"The manuscript is well written up section 3.3.1., after which it deteriorates conspicuously"

**Response: We agree that the logical structure of section 3.3.1 could be improved and have carefully revised this part.**

"In principle, there were two objectives: 1. Broadscale simulation of AOM dynamics: It does a very good job at simulating a range of broadly set environmental conditions with direct impact on the filter efficiency of anaerobic methane-oxidizing microbial consortia that use methane and sulfate. The range of the environmental conditions is set broad enough to encompass conditions that may be encountered on the East Siberian shelf. However, this part is not very novel and AOM dynamics and filter efficiency have been reviewed by Regnier et al. (2011) previously. Therefore all sections of the manuscript that relate to the simulation tests should be significantly shortened."

**Response: We strongly disagree with this comment. Regnier et al., 2011 present a comprehensive review of previously developed models that have**

been applied to investigate a large employed to simulate a large set of diverse depositional environments affected by intense methane cycling, ranging from mud volcanoes and active seeps to passive sediments experiencing groundwater discharge or high organic matter inputs. The review explicitly explores how different model implementations/formulations (with increasing complexity of the biogeochemical network) perform in simulating methane-affected sediments, as well as explore simulated AOM efficiency in response to a discrete, non-specific set of environmental conditions considered in these models.

However, the analysis of AOM filter efficiency and $CH_4$ effluxes presented has a completely different focus and goes well beyond the analysis presented in Regnier et al., 2011. As pointed out above, the main aim of this model study is to specifically investigate the potential escape of dissolved methane released from a deep source (*e.g.* dissociating hydrates or thawing subsea permafrost) from warming Siberian Shelf sediments. It thus assesses the efficiency of the microbial AOM filter in attenuating potential dissolved permafrost/hydrate methane fluxes under a continuous and specifically chosen range of environmental conditions/scenarios (likely) encountered on the present and (idealized) future Siberian shelf using an identical model set-up and thus offering not only more robust theoretical consistency and comparability. The main focus of the presented sensitivity analysis lies on identifying environmental conditions (and thus potential areas on the Siberian Shelf) that favor non-turbulent dissolved methane fluxes across the sediment-water interface.

We further emphasized this point in the manuscript by modifying the introduction and abstract accordingly.

"2. Regional application: The second part of the manuscript is the application of the model to the East Siberian shelf. I found this part the more relevant one, given the title, but unfortunately also less well constrained due to the paucity of data used to constrain their model in face of the diversity and size of the targeted marine region. For reference, my guess is that the authors would certainly not model the whole of the North Sea or the Baltic Sea with this model, two marginal seas of similar size or even smaller than the Laptev Sea"

Response: We also disagree with this statement. One strength of a models is that it can provide the explorative means to assess dynamics at spatial/temporal scales that cannot easily be assessed by observations alone. In particular, transfer functions, simple look-up tables and/or neural networks that are derived from or trained on a large ensemble of individual model simulations over a broad range of plausible boundary conditions have been frequently and successfully used to investigate regional and even global dynamics.

For instance, Gypens et al., 2008, Dale, Nickelsen, et al., 2015, Dale, Graco, et al., 2017, Capet et al., 2016 use simple transfer functions derived from a large ensemble of 1D diagenetic model simulations to predict benthic

nutrient recycling fluxes for the coastal North Sea (Gypens et al., 2008), the Peruvian Upwelling system (Dale, Graco, et al., 2017), the entire global ocean (Bohlen et al., 2012; Dale, Nickelsen, et al., 2015) or the entire Black Sea (Capet et al., 2016). Marquardt et al., 2010 used a transfer function to estimate the global gas hydrate inventory in marine sediments. In addition, Bourgeois et al., 2017 used a generalized additive model to calculate oxygen fluxes through the sediment-water interface for the entire Arctic Ocean and Artificial Neural Networks have been used to estimate sulfate (Bowles et al., 2014) fluxes through the sediment-water interface on a global scale.

These approaches are similar to the regional assessment presented here and illustrate the power of such transfer functions. We now highlight this in the introduction.

"My specific critique relates to the following points, which to my opinion are important in controlling the biogeochemical rates and flux output of the model, but that are not or too poorly constrained in the model to substantially further our understanding of how efficient anaerobic methane oxidation is and will be in the Siberian shelf sediments. Even with the reduction of the investigated area to the Laptev Sea only, the depositional environments and geological settings are so much more variable that a simple sedimentation rate/bathymetry-based prediction of present-day organic carbon accumulation gives a starting condition for the model that is too simplifying to be acceptable."

Response: The results of the extensive sensitivity study presented here clearly indicate the sedimentation rate and active fluid flow exert the dominant control on the escape of methane derived from thawing permafrost and/or disintegrating methane gas hydrates through the Siberian shelf sea floor across a wide range of contrasting environmental conditions encountered in this depositional environment. Results show that additional environmental conditions, such as OM content or AOM efficiency (*i.e.* $k_{AOM}$) play a minor or negligible role. Sedimentation rate can thus be used to predict the non-turbulent of methane escape on the Siberian Shelf.

The extensive sensitivity study presented here, thus also confirms the general approach that underlies the ensemble of studies listed in the previous response: single benthic biogeochemical characteristics, such as seafloor fluxes, redox horizons or inventories are often controlled by a limited set (1-2) of dominant factors that can then be used to robustly predict these characteristics on a regional/global scale.

"For example, the authors rely on a selected handful of Pb-210 data (there are more available in the literature for better coverage (see Bröder et al., 201; Strobl et al., 1988) for sedimentation rates"

Response: We thank the reviewer for the suggestions. Bröder et al., 2016 reports values for two sites in the East Siberian Sea and can thus unfortunately not be used to improve data coverage in the Laptev Sea. However, the reported linear sedimentation rate $(0.14 - 0.15 \text{ cm yr}^{-1})$ is not only similar to the sedimentation rate used in our local model application ($0.12$ cm yr$^{-1}$), but would also not change flux calculations if applied (see sensitivity study). We now include the values reported by Strobl et al., 1998. They show that sedimentation rate in the Laptev sea is of the same order ($0.15$ cm yr$^{-1}$)- a value that falls well in the range we explored.

"The model doesn't consider the regionally diverse sediment types, permeabilities and rates in the Siberian Shelf Sea (see for example Dudarev et al., 2006 Oceanology; Rekant et al., 2015). The model doesn't consider known clay/sand/sand grain size variation and their influence of carbon concentration, permeability, transport, and resulting biogeochemical rates."

Response: We would like to stress again that the presented study does account for the regional variability of sedimentation rate: 1) in the sensitivity study considering a large range spanning almost two orders of magnitude ($0.03 - 1.5$ cm yr$^{-1}$), and 2) in the regional analysis that applies a spatially variable sedimentation rate. In addition, the influence of the amount of degradable OM has also been tested in the sensitivity study and, because it is of secondary importance, is qualitatively discussed in the regional study.

It is however correct that we assume a porosity profile, which is representative for fine-grained shelf sediments. This is in agreement with Dudarev et al., 2006 (although they focus on the East Siberian Sea and not the Laptev Sea). They suggest that: "*The distribution of sediments demonstrates that they sustain fine-grained texture in the major part of the continental shelf regardless of the distance from the shore*". Considering that the overall geomorfological characteristics of the East Siberian Sea and Laptev Sea are similar, we can assume that a $3$ m sediment column with a prescribed porosity (dependent on depth) and a uniform texture and sediment type might be a decent representative for a large setting of the ESAS. We added a comment to the methods section.

"The model assumes Barents Sea depositional conditions as a good analog, however, these are unlike those of the Siberian shelf, since the Barents Sea is much deeper, has higher marine productivity, less ice cover, and much less input of terrestrial organic matter. In addition, it does not have terrestrial permafrost underneath the recent Holocene sediments. It is therefore not a particularly good analog. If the authors are interested, I can provide porewater methane, sulfate and ammonium data from this region."

Response: We would like to thank the reviewer for this offer. We have been in contact with the reviewer for porewater methane, sulfate and ammonium data and now include an additional model test case for this Laptev Sea site. We would however also like to stress that we do not consider the Barents Sea shelf offshore Versterålen as a good analog for the ESAS. Due to the paucity of observational data from the Laptev Sea for model testing, we used this Arctic site to illustrate the performance of our model set-up in simulating biogeochemical dynamics in high-latitude shelf sediments.

"The reactive continuum approach employed here probably overestimates the reactive organic carbon amount that is available to organic carbon degradation at depth. In reality, the reactivity of the organic matter below the oxic horizons is one to two orders of magnitude lower than commonly observed in marine shelf sediments (see Figure 9, Brüchert et al., 2018). Given the very low reactivity of carbon in these sediments (See Brüchert al., 2018; Bröder et al., 2016; Tesi et al., 2014), sulfate is likely never exhausted and methanogenesis and AOM may not even take place in these sediments at all. I am therefore not surprised at all that the authors arrive at such low regional dissolved benthic methane fluxes, seemingly at odds with the broadly published claims of extensive methane emission from the Siberian shelf."

**Response: This is a misunderstanding which we would like to clarify. First of all, we would also like to emphasize again that, according to our findings, the organic matter reactivity only exerts a secondary effect on our conclusions and therefore does not alter the overall picture of our results. In addition, we would like to stress again that the focus of the presented analysis centers on the fate of methane fluxes from thawing permafrost and/or disintegrating methane gas hydrates and not in-situ biogenically produced methane for which OM reactivity may play a more important role. The presence of a deep methane flux from thawing permafrost and/or disintegrating methane gas hydrates also ensures the presence of an AOM and the depletion of sulfates.**

**However, apart from this, we also disagree with the overall comment that the reactive continuum model (RCM) overestimates reactivity in these sediments. In fact, the RCM accounts for the decrease of OM reactivity with sediment depth/degradation state. Here, we test a wide range of RCM parametrizations (*i.e. a*) including those that result in a rapid decrease of OM reactivity by 1-2 orders of magnitude. Moreover the two papers cited actually support the use of a reactive-continuum model.**

1. **Bröder et al., 2016 show that the half-life of the organic matter deposited at two sites in the East Siberian Sea is $19 - 27$ yr. These half-life are represented by our RCM parametrizations in the intermediate range. Assuming $\nu = 0.125$ the corresponding a for the two samples would be $a = 3.4 - 4.8$ yr - values that are well within the range explored in our sensitivity analysis.**

2. **Tesi et al., 2014 in their conclusions clearly state: "*Therefore our results suggest that TerrOC is made of several allocthonous pools each with distinct reactivity toward the oxidation (i.e., reactive continuum)*".**

**We modified the method section to clarify this point and also added the two references.**

"In fact, these fluxes confirm my own direct measurements of porewater methane concentrations and methane fluxes from a range of stations investigated in the summer of 2014 during the SWERUS expedition with the Swedish icebreaker Oden. If the authors are interested, I am willing to share these data with them to better constrain their model."

**Response: We are really thankful for this offer and have been in contact with the reviewer.**

"The model doesn't consider Holocene sealevel change to elaborate on the mass of sediment available for methane generation since the last glacial maximum, which is the time since reactive sedimentary organic carbon accumulation began."

**Response: This is a misunderstanding. Again, the focus of the presented paper is on the fate of methane released from subsea permafrost/gas hydrates on the present-day and future Siberian shelf. We do not intend to simulate the historical evolution of the SSPF and of related historical methane emission, but only a plausible range of current/future ones. Furthermore, our model analysis is based on the simulation of the first $3$ meters of sediment and the Holocene sedimentation rates we explored ($0.03 - 1.5$ cm yr$^{-1}$) indicate that the sediment layer overlying the subsea permafrost always exceeds $3$ m.**

"The model design relies on a sequence of thermodynamically regulated terminal electron acceptor reactions driven by fresh carbon accumulation at the top of the model domain. In reality, non-biogenic or old Pre-Holocene-produced methane transport from below (of thermogenic or Pleistocene age, i.e., terrestrial) is the key unique characteristic of the Siberian shelf with respect to methane cycling. This carbon is old and uncoupled to recent carbon accumulation. In addition, carbon accumulation varied greatly through time on the Siberian shelf. The model appears to assume continuity of recent depositional conditions back in time and space, which is most certainly incorrect."

**Response: This is a misunderstanding. In fact, the model analysis focus on this "non-biogenic or old Pre-Holocene-produced methane transport from below (of thermogenic or Pleistocene age, i.e., terrestrial)" and not on the in-situ produced biogenic methane. Because it is impossible to reconstruct depositional conditions over the Holocene for the entire region, we indeed assume broadly similar depositional conditions during the Holocene. This is an acceptable simplification, in particular because:**

1. **Early diagenetic rates are highest in the shallow, young sediment layers and decrease rapidly with depth. As a consequence, biogeochemical dynamics are mostly affected by recent depositional conditions. This is especially true in the light of the fast decrease in OM reactivity reported by broder 2016; Brüchert et al., 2018; Tesi et al., 2014.**

2. **Our comprehensive sensitivity study indicates that OM degradation and biogenic methane production in the Holocene sediment layer ex-**

**erts a minor control on non-turbulent methane fluxes across the sediment-water interface. Holocene fluctuations in environmental conditions will thus exert a negligible effect on our results.**

**We clarify this throughout the manuscript (see previous replies).**

"Only the section with the transient model scenarios therefore applies to the Siberian shelf and only scenarios with an explicit upward flux of methane are relevant for investigating AOM dynamics in these sediments. However, because of the difficulties in constraining the regional distribution of seeps, flux rates cannot be reliably extrapolated and one should refrain from a regional flux estimate."

**Response: This is a misunderstanding. All steady-state simulations also apply an upward flux of methane (as outlined in the method section for details). They are thus relevant for investigating the fate of permafrost/hydrate derived methane in the Holocene sediment column and its possible escape through the sediment water interface. They also allow to derive the transfer function for possible non-turbulent methane escape that has been used to establish a regional estimate. We clarify this point throughout the manuscript (see previous replies).**

**Because our steady state analysis shows that AOM acts as an efficient biofilter and mostly prevents non-turbulent methane escape from the sediment, we also explored a number of plausible transient scenarios to explore if microbial dynamics could possibly create "windows of opportunity" for methane escape and assess their importance. We further clarify this in the introduction and method section. in the transient analysis we performed we actually refrained from an upscale estimate and we just explained the result of the flux out of simulated sediment column.**

"My objections to the present manuscript are therefore not whether the model's capabilities are useful to the scientific community in general, which it certainly is, but a critique of the attempt to mimic biogeochemical as well as recent and past depositional conditions on the Siberian shelf to better predict sediment methane emissions from this region."

**Response: see responses above.**

"I am fully aware of the infected discussion of the relevance of the Siberian shelf sea's role as a potentially huge methane source to the atmosphere put forward by Shakhova and co-authors. The outcome of the model simulations presented here, even in their most generous state (high advective upward flow and moderately to high sedimentation rates), would imply that the emissions proposed by Shakhova and coauthors are very hard to achieve without invoking massive gas emissions (which are not seen regionally in atmospheric measurements)."

**Response: This is indeed one of the conclusions of our analysis.**

"However, the inability of this 1D model to encapsulate environmental conditions

that are found in the Laptev and East Siberian Sea make it impossible to use its scaled model output to the current system or to use the model to make reliable assessments of how the shelf environment may change methane fluxes in the future. Particularly the latter requirement is key to the use of a reaction transport model such as this one in climate science. [...] The study and conclusions give the false impression that this particular model is capable, with certainty, to predict the non-gaseous methane flux emanating from this 1.5 million square kilometer large region, if one only knows the sedimentation rate and water depth. The authors may therefore consider a new title for their manuscript for the first section and resubmit it under this new title without much reference to dissolved methane emissions on the East Siberian shelf, since this is not what they can model reasonably with the data they have available. [...] Alternatively, the model simulations can be tested with actual data from the Siberian shelf, which I am willing to share. In this case, I would suggest to reduce the first part of the manuscript and focus on the application of the BRNS to the Siberian shelf sea rather than a broad treatment of the model's performance."

**Response: This comment reflects a string of misunderstandings. We do not aim at quantifying, "*with certainty*" the exact evolution of present and future methane emissions from the Siberian shelf. As highlighted in the title, abstract, introduction, the presented study assesses the potential for non-turbulent methane escape (derived from deep sediment sources such as permafrost/gas hydrates) from Siberian shelf sediments. As pointed out in the results and conclusion section, it thus provides a robust, quantitative framework suitable to make first order estimates and draw conclusions with respect to present and potential future emissions, as well as methane gas emissions required to support previous estimates of Arctic Ocean methane emissions to the atmosphere. Given the urgent need to assess this potentially ticking time bomb, but the paucity of observational data, it represents a feasible and robust quantitative first step towards a better assessment of the threat methane emissions from thawing subsea permafrost/ disintegrating methane hydrates pose for our climate.**

**Therefore, we are convinced that the title, as well as the approach of the presented study adequately reflect its scope and do not give a false impression. However, we have adapted the abstract, introduction, method and conclusion sections to further clarify these points. In addition, we have also included a new case study for the Laptev sea site based on the data provided by the reviewer.**

**Specific comments**

Page 8: "This is a crude overgeneralization. The authors must provide more references on the physical oceanography of the Laptev Sea and its sediment distribution and bathymetry to justify this comparison. The Norwegian setting has much higher primary productivity, is up to 8 times deeper and has substantially less ice cover over the year. If anything, the Vesterålen site shares very few similarities with the Laptev

Sea or the East Siberian Shelf Sea."

**Response: This is a misunderstanding. As pointed out in the response to general comments, we used the Hola trough sediments merely to assess the ability of the model to simulated carbon and sulfur dynamics in high latitude shelf sediments porewater profiles in a Northern shelf. No calibration of the BRNS or other following results relies on the simulations performed to reproduce the Vesterålen site, nor do we claim any similarity with the shelf areas of the East Siberian Arctic shelf. However, we do agree that our statement could be misunderstood and have now modified this section accordingly.**

Page 12: "Please correct, not for methane"
**Response: "Simulation results show an overall satisfactory agreement with measurements except for methane."**

Page 13:

- "It is not correct to make reference to the ESAS, since the range of the environmental conditions applied here is sufficiently broad to be applied to a wide range of shelf and slope margin settings with possible AOM. One condition worthwhile exploring and not done here is whether at low OM reactivities, the consumption of sulfate may not be completed for the time span of Holocene sediment accumulation on the ESAS (i.e., since ca 7000 years ago)."
  **This is a misunderstanding. As stated earlier, we investigate the fate of methane from deep sources (permafrost/hydrate) rather than in-situ produced methane (although the model also accounts for biogenic production in the Holocene sediment layer). As a consequence, we apply a range of methane fluxes from below that ensure a consumption of sulfate. With respect to the comment on the environmental conditions, we would like to repeat our response to a similar general comment here.**

  "*While the reviewer is absolutely right in pointing out that the results of the comprehensive sensitivity study described in the manuscript are universally valid, we would like to stress that the model setup and the sensitivity study have been specifically designed with the aim of assessing the fate of dissolved methane released from a deep source (e.g. dissociating hydrates or thawing subsea permafrost) in warming Siberian Shelf sediments. More specifically:*

  - *The model is forced with a variable flux of dissolved methane potentially originating from dissociating methane hydrates and/or thawing permafrost in the deeper sediment. The methane flux is constrained by assuming lower model boundary methane concentrations ranging from $0$ to a maximum concentration that is constrained by the saturation of dissolved $CH_4$ under pressure,*

*temperature and salinity conditions encountered on the Siberian shelf.*

– *All model boundary conditions, forcings and parameters (Tables S5 and S6) are chosen to be representative of environmental conditions encountered on the Siberian shelf.*

– *The range of boundary conditions and parameters tested in the steady state sensitivity study are constrained based on data compiled for the Siberian shelf.*

*As a consequence, the study presented here does not cover the entire range of possible conditions (e.g. methane fluxes, active fluid flow, organic carbon concentrations etc.) encountered at the global ocean seafloor, but is representative for conditions (likely) encountered on the present and future Siberian Shelf."*

- "Please correct to : 'to the SWI' The model does not provide any constraint on the SWI flux, i.e., the benthic flux itself, because here other processes play an important that are modelled here."
**Response: We are not sure which processes the reviewer refers to, but in addition to diffusion and advection, the model explicitly accounts for bioturbation and non-local transport (through bioirrigation or ice scouring). It thus provides a robust representation of transport through the SWI.**

- "Referencing this study to other studies that show a range of 5 orders of magnitude in methane fluxes to justify its applicability seems odd. Please clarify how exactly each of the referenced studies supports the model findings in your simulation."
**Response: The referenced studies offer a comparison with respect to the fluxes, as well as the flux variability in response to different environmental conditions we simulated.**

- "Which value was that? Not clear from the text. Apart from that, I deeply object to the use of one value to the whole of the ESAS. What is the purpose of this upscaled value? The original model value doesn't gain any more legitimacy from upscaling and the fact that the upscaled value may be in the range of expected values neither. Please delete this section"
**The maximum value we found was 27.48 $\mu$molCH$_4$ cm$^{-2}$ yr$^{-1}$. We added the exact value to the respective section. As pointed out in the earlier response, model results provide a robust quantitative framework to evaluate the potential for non-turbulent methane escape from the Siberian Shelf. The purpose of upscaling the maximum value to the ESAS is simply to offer an upper limit for this possible non-turbulent methane flux and show that, even if the most favorable conditions for methane escape were to be found over large shelf areas (note, this is**

different from claiming that they are), non-turbulent methane fluxes would still be negligible and would not be able to support earlier estimates of methane emissions to the atmosphere.

Page 14:

- "This is an interesting conclusion. How can one reconcile the observation that methane concentrations in the methanogenic zone generally tend to increase with depth, i.e., their transport away from the zone of formation is too slow relative to the methanogenesis rate?"
  **Response: The Damköhler numbers are defined in such a way that the transport process considered occurs in the same region as the reaction, *i.e.* we considered the methane transport within the methanogenic zone for the evaluation of $Da_{MG}$ and the SMTZ for the evaluation of the $Da_{AOM}$ . Simulation results reveal that methane transport is efficient within the methanogenetic zone. However, comparison with $Da_{AOM}$ shows that methane consumption within SMTZ is slower than its transport. In other words, methane can be efficiently transported to SMTZ but it is not quickly consumed there. As a consequence, methane accumulates below the SMTZ because at the SMTZ level it is not consumed and below the SMTZ no AOM occurs.**

- "This is a curious assertion for the Siberian shelf system. It is wellknown that the sediments of the Siberian shelf are not reactive enough to yield significant methane. It is instead supposed that externally introduced methane from the thawing permafrost that serves as the methane source. The current model does not take external sources into account and this is the major flaw of this paper. It is actually not suited in the current version to model the processes on the Siberian shelf."
  **Response: Deep (external) sources of methane are the main focus of the presented study. See response to general comments for details on biogenic methane production, methane fluxes from permafrost/hydrates.**

- "This introduction paragraph is rather wordy and doesn't say much. Can it be shortened?"
  **Response: we will shorten it in the finalized version of the paper, although we value the fact that an introduction might already provide the main message of what is described in detail later.**

- "Please provide a reference to the 'traditional views'. The view proposed here is not new."
  **Response: We replaced "traditional" with "intuitive". Our findings give further evidence of the dominant role of transport processes for non-turbulent methane effluxes also in modeling scenario compatible with ESAS settings.**

Page 15:"What is meant by 'margin'?"

**Response: the continental margin. We could replaced "margin" with "shelf" to avoid confusion.**

Page 17: "The authors should avoid trivial sentences such as this one."

**Response: it is not necessarily trivial, since a high methanogenesis might also be expected to foster a higher oxidation process and therefore accumulation of methane is not necessarily a triviality**

Page 19: "I wonder whether the reactivity of organic matter in large parts of the Siberian Shelf isn't even lower than 100 years. More 1000 years."

**Response: we also explored the $a \geq 100$ yr. As already stated in the reply to the general comment, the reactivity of the organic matter reported in other studies (*e.g.* Bröder et al., 2016) shows that $a$ is $< 5$, not far from the value $a = 10$ yr we used for the baseline simulation. In addition, $a$-values $>1000$ years are characteristic for deep sea sediments underlying extremely oligotrophic gyres, such as the deep South Pacific. Shelf, slope and most deep sea environments are generally characterized by $a < 1000$ years.**

Page 23: "The authors are conflating to independent processes into one."

**It is not clear which processes the reviewer refers to. We guess they are, on one hand, the actual AOM and, on the other hand, the precipitation of authigenic carbonate. We do not claim or mix them up and we are aware that they are two different processes but it is well established that they are not independent, since the alkalinity produced during the AOM can drive precipitation of authigenic carbonates as reported in many site all over the globe (*e.g.* Aloisi et al., 2004; Crémière, Lepland, Chand, Sahy, Condon, et al., 2016; Crémière, Lepland, Chand, Sahy, Kirsimäe, et al., 2016; Karaca et al., 2010; Luff et al., 2005; Meister et al., 2018; Pierre et al., 2012). We are simply hinting at an indirect effect supporting our findings, aware that the two processes are however well distinct and not trivially connected.**

Page 24: "These calculated active and passive fluxes are so low that they are empirically not verifiable with currently available measurement techniques."

**Response: We are aware of this limit and acknowledge it in the study. However, we would also like to point out that the exact quantity of these small fluxes is of minor importance. What is important here is that the potential for non-turbulent methane fluxes from Siberian Shelf sediments, even under the most favorable environmental conditions, is extremely limited and previous estimates of methane emissions to the atmosphere would thus require the build up of large quantities of methane gas.**

Page 26: "The question is more, whether biogenic methane ever forms in these sediments, as the authors likely overestimate the reactivity of the organic matter. Altogether I think that the authors arrive at the right conclusion for the wrong reasons."

**As stated previsouly, we disagree with this comment. Please see reply to general comment for details.**

Page 28: "From this section on the manuscript becomes distinctly less well written, more typographic errors and less succinct writing. At the same time, the discussion of transient conditions is most relevant to the Siberian shelf system. This section needs to be carefully revised and improved in its writing."
**Response: We will carefully revise and improve this section.**

Page 29: "A better way of explaining the discrepancy between the two methane fluxes at steady state and the transient condition would be to show the AOM rate for the two rate laws."
**Thanks for the suggestion. We add the AOM rate profile to fig. 11.b**

Page 31:

- "This is hard to understand. It should be possible to extract the instantaneous apparent kAOM value throughout the simulation. Ultimately of relevance is not what the kAOM is at the end of the simulation, but its time-integrated AOM rate throughout the modelled transient run."
  **It is actually possible to extract the $k_{AOM}$ at each simulated time step. However, here we wanted to explain why the final, new steady-state flux in the bioenergetic formulation is different from the simulation with the bimolecular formulation and that is the reason we focused on the final $k_{AOM}$, its shape and values.**

- "Poor English makes this paragraph hard to understand, most importantly it is not clear how the authors arrive at their conclusion with this argument"
  **Response: We will carefully revise and improve this section.**

- "thermodynamical"
  **Response: Corrected**

Page 32:

- "19 years"
  **Response: Corrected**

- "The role of sulfide was not mentioned previously. Is sulfide generally an important player for thermodynamic calculations done here?"
  **Sulfide influences AOM it appears in the formulation of $F_T$, which controls the AOM in the bioenergetic approach as shown in Eq. 11. Bicarbonate appears as well, but it is rarely a limiting factor.**

Page 33:

- "The wording should be reversed. An AOM biomass accounts for an AOM filter, not the other way round"

**Response: we agree but we wanted to stress that in order to have an efficient AOM filter a minimum AOM biomass is needed and this quantity has been estimated to be $> 10^{10}$ cells cm$^{-3}$, which is of the same order of magnitude as the value we found.**

- "Overall, this is irrelevant. The supply from below is what counts for the Siberian shelf, not the in-situ production, which is negligible in almost all settings except for the Eastern East Siberian Sea and the Chukchi Sea. In addition, the statement is also irrelevant in a general sense. As the supply from below is increased, so must the proportional contribution of in-situ produced methane decrease. This is not worth mentioning."
  **Response: We will edit this sentence accordingly in the final version of the paper.**

- "typo here: from ... to.."
  **Response: Corrected**

- "I am getting lost with the abbreviations"
  **$[CH_4]_-$ is the methane concentration at the bottom of the sediment column.**

- "As stated this is not true and must be corrected. Never did you investigate ESAS shelf sediments in this study. Modeling scenarios were investigated, of which some conditions may apply to selected environmental setting on the ESAS. The passive/active terminology strictly applies to theoretical scenarios of system behavior.[...] Seriously, the authors have not investigated these sediments directly at all and should not make a claim to have investigate them."
  **Response: This is a misunderstanding. The focus of this study is not a regional simulation of ESAS shelf sediments, but to develop a robust, quantitative framework that can be used to evaluate the potential for non-turbulent methane escape driven by thawing subsea permafrost and/or disintegrating methane gas hydrates on the warming Siberian shelf. We would again like to repeat our response to one of the general comments.**

  *"**This comment reflects a string of misunderstandings. We do not aim at quantifying, "with certainty" the exact evolution of present and future methane emissions from the Siberian shelf. As highlighted in the title, abstract, introduction, the presented study assesses the potential for non-turbulent methane escape (derived from deep sediment sources such as permafrost/gas hydrates) from Siberian shelf sediments. As pointed out in the results and conclusion section, it thus provides a robust, quantitative framework suitable to make first order estimates and draw conclusions with respect to present and potential future emissions, as well as methane gas emissions required to support previous estimates of Arctic Ocean methane emissions to the atmosphere. Given the urgent need to assess this potentially ticking**"*

*time bomb, but the paucity of observational data, it represents a feasible and robust quantitative first step towards a better assessment of the threat methane emissions from thawing subsea permafrost/ disintegrating methane hydrates pose for our climate.*

*Therefore, we are convinced that the title, as well as the approach of the presented study adequately reflect its scope and do not give a false impression.*

*However, we also modified this section accordingly to avoid misunderstandings."*

- "first or first-order?"
  **Response: Actually both first and first-order. Modified accordingly.**


   - Porewater data from the Laptev Sea are now taken as reference instead of data from the Barents Sea.

   - More consideration is given to studies that investigated the reactivity of organic matter on the Siberian shelf, e.g., Wild, Tesi, Brüchert, Bröder, etc.) than in the first version.

   - The revised version explicitly considers transient response due to upward flow from gas hydrates and thawing permafrost simulating various potential scenarios."

**Response: We would like to thank the reviewer for his comments which allowed us to better clarify and corroborate our findings.**

"The authors use the same extrapolation of their data to the whole Laptev Sea that is based on a sedimentation rate map, but also calculate a Laptev Sea-wide transient seepage flux of 2.6 to 4.5Tg/yr-1 for the whole Laptev Sea without taking localized seepage into account. At least, this is my understanding from reading the section on the extrapolation. It is not clear, how the authors integrate the transient response into the spatial model." **Response: we did not include transient response in the spatial extrapolation which only relies on steady-state results.**

**Specific comments**

All the comments concerning the wording, typos and text extension/removal have been addressed and not listed below. If a different choice has been made the comment is reported below with a response.

Page 3:

- "wording: the shelf is not a basin, it is essentially a broad flat plain."
  **Response: we follow the lines of use of this expression in literature, considering that a sedimentary basin like the Siberian shelf is basin which has been filled in with sediments. Examples of similar use are in Drachev, 2016; Franke et al., 2005; Gramberg et al., 1983; Herman, 1989; Malyshev et al., 2012.**

- "wording: what is a typical sedimentation rate in the ocean?; shouldn't you use a typical shelf sedimentation rate, if at all?"
  **Response: we mainly wanted to stress the difference in sedimentation rate on shelves with respect to other marine settings. As reported in Burwicz et al., 2011, the accumulation rate on shelves is of the order of $0.1$ cm yr$^{-1}$, 5-10 times faster than in the rest of the Ocean ($<= 0.01$ cm yr$^{-1}$), although highly spacial dependent.**

- "glacial interglacial changes do not affect the sulfate concentration significantly"
  **Response: For sulfate we mean only the seasonal changes, not referring to interglacial states. We simply linked salinity to sulfate concentration employing the empirical relation provided in Dickson et al., 1994 and the seasonal salinity measured in Dmitrenko et al., 2011, which showed a seasonal variation in bottom salinity ($20.68 \pm 1.80$ psu in summer *vs.* $26.61 \pm 0.92$ psu in winter).**

Page 9:

- "I don't understand this. Fe(OH)3 and iron reduction is very apparent in these sediments in the upper 10 cm?"
  **Response: we wanted to highlight all the reactions involving $PO_4^{3-}$, and the sorption on $Fe(OH)_3$ among them. However, because of the small reaction constant and the amount of $Fe(OH)_3$, it is expected to be relevant only in the narrow horizon where iron reduction occurs.**

- "This is a very high value and unlikely found here, 2.2 - 2.4 g cm-3 are more realistic."
  **Response: this is typo. The real value which has been used is reported in Table S6 and is $2.41$ g cm$^{-3}$.**

Page 10:

"I wonder why you didn't use the sedimentation rates given by BrÃűder et al 2016 or Vonk et al 2012 for the working area?"

**Response: Sedimentation rates given by Bröder et al., 2016 ($0.15 \pm 0.04$ cm yr$^{-1}$ and $0.14 \pm 0.03$ cm yr$^{-1}$) and Vonk et al., 2012 ($0.11 - 0.16$ cm yr$^{-1}$) do not differ from the one we used in our baseline simulation, *i.e.* $0.123$ cm yr$^{-1}$.**

Page 12:

"There are distinct discrepancies that the model appears not to capture very well. I wonder whether forced concave CH4 profile of the model reflects reality. This model predicts very high CH4 cocentrations beyond the data domain, not in line with an extrapolated trend of the measured data."

**Response: we do agree that there are species not well reproduced. Based on the results for $NH_4^+$ we try to give a plausible explanation for such discrepancies. A local change in organic matter quantity and/or reactivity beyond the data domain is the most plausible explanation for the disagreement in $NH_4^+$ profile, considering the reaction network implemented in BRNS. In fact, $NH_4^+$ is only affected by the degradation of the organic matter, nitrification (which occurs only in the upper and thin oxic layer) and adsorption. Even in case the adsorption process were misreproduced, it would mainly cause a horizontal shift in the profile which is however not compatible with the data profile. For such a reason a change in the OM reactivity and/or quantity remains as the most plausible interpretation.**

**Such a change in organic matter properties may cause also the disagreement in $CH_4$ profiles, especially concerning the concavity. The model well reproduces the other species ($SO_4^{2-}$, DIC, $PO_4^{3-}$ and $O_2$) only if methane concentration gets high enough to oversaturate pore water at depth. A reduction of organic matter quantity and/or reactivity or a local dishomogeneity might account for a milder (linear) increase in $CH_4$ concentration with depth. It must also be stressed the crude and simplistic parametrization of gas biogeochemistry and physics in the sediments: a factor capable of altering the methane profile at depth. However, the qualitative behaviour in the upper part of the sediments, directly determining the diffusive flux into the water column, is grasped and this was the main aim of the section.**

Page 18-19:

"this sentence is not clear to me. What other advection fluxes do you mean here? Bioirrigation? And this doesn't even account for the bubble flux? And what do you mean by both the advection and molecular diffusion flux to the total flux. Shouldn't this say the relative contribution of the molecular diffusion to the advection flux? The terminology is confusing."

**Response: we made explicit the 4 mechanisms affecting the non-turbulent (*i.e.* non-bubble mediated) flux at the Sediment-Water Interface:**

1. molecular diffusion

2. bioturbation: described as a diffusion-like process

3. advection: due to the imbalance of the sedimentation rate $\omega$ and the flow velocities $v_{up}$)

4. bioirrigation: described as a non-local transport of dissolved species, hence bubbles are not accounted.

**We refer to the section S1 for a thorough description of the mathematical modeling of these flux components, which are however at the base of most of the reactive-transport models (see Boudreau, 1997).**

Page 20:

"at least for the range of OM degradation rate constnts chosen here. The chosen range is narrow, though."
**The range of our investigations cover 4 orders of magnitude $[0.1 - 1000]$ yr. We disagree that the chosen range of investigation is narrow.**

Page 27:

- "This manuscript comes to the conclusion that methanogensis from transported terrestrial organic matter and deposited on the ESAS as sediment can berce of methane. This is a new conclusion, because all previous work on the ESAS has focused on mobilisation of previous drowned in-situ terrestrially organic matter as methane source. This is a totally different mechanism, for which so far no field data exist to my knowledge. In fact, previous investigations, e.g., Koch et al. 2008; Overrduin et al., 2015; 2016; Thornton et al., 2015; shown that the interface with submarine permafrost is where AOM is prevalent. These data also show no signfcant gradient in sulfate despite the near-shore locations of the drill cores. To some extent, the model results should also be consistent with these observations and not propose methanogenesis in Holocene sediments."

  **Response: we disagree with such a conclusion, which is not in line with our statements. We would like to clarify that, although present, the methanogenesis in Holocene sediments (*i.e* those sediments above the drowned *in-situ* terrestrial organic matter) does not appear indeed to be the main source of methane according to our results. In fact, if we consider the simulations with zero input of methane from below, the flux of methane is generally almost negligible (see for reference Fig. 3). The only cases where the methane flux is relevant, even with 0 input of methane from below, and therefore the only cases where the methanogenesis occurring in upper sediments is crucial to have a sizable methane flux, are the simulations with a high upward flow velocity**

$(v_{up} > 5$ cm yr$^{-1}$) and/or a high sedimentation rate (roughly $\omega > 0.6$ cm yr$^{-1}$ but very likley higher, looking again at Fig. 3). There is no evidence that either of these conditions could be found in the ESAS: considering that the highest sedimentation rate found is $\sim 0.4$ cm yr$^{-1}$, even if we cannot **a priori** exclude that on small scales they might be met, although we presume that currently they would not contribute largely enough to alter the methane flux we found. However, we have chosen to include the results of the simulations considering these "extreme" conditions in the discussion of the results for two reasons: i) they are genuine and plausible model results (supported in evidence for instance by similar results by Egger et al., 2016), ii) changes of the environmental conditions of Arctic coasts are quick and there is no reason to exclude that there might be regions or future scenarios where these conditions might be present, and this is exactly what we do not want to rule out in our conclusions.

- "do you have a reference for this?"
  Response: **the most direct reference is Brüchert et al., 2018: "Evidence for bioturbation and bioirrigation based on multiple micro-electrode profile measurements per core was rare", "The good fit between the two methods also supports the notion that bioirrigation and bioturbation effects from meiofauna and macro-fauna were minor." and "For the other stations, optimal fits required no sediment mixing by bioturbation or advective pore-water transport by bioirrigation". But the lack of evidence and report of faunal activity in the description of other cores (Miller et al., 2017; Overduin et al., 2016; Shakhova, I. Semiletov, Gustafsson, et al., 2017; Shakhova, I. Semiletov, Leifer, et al., 2014; Wild et al., 2018) and the rough conditions due to ice scouring give indications in this direction.**

- "How is this number extrapolated? Simply by multiplying? But active seepage would be highly localized. How can one derive at this number?"
  Response: **yes this is just a simple multiplication time the area of interest and it just offers an upper constraint to the transient flux to the water column**

- "But transient scenarios where the SMTZ is closer to the SWI, would respond seasonally, I presume?"
  Response: **based on our simulation there is no room to infer that this occurs. Looking for instance at Fig. S16 and S17 there is no sign of any seasonal variation affecting either the methane efflux or the SMTZ level. They seem to respond only to the longer trend. And this is actually also expected considering the shallowest SMTZ we found**

($\ell_{SMTZ} \simeq 16$ cm) and the diffusivity of $SO_4{}^{2-}$ ($D \simeq 174$ cm$^2$ yr$^{-1}$). A back-of-the-envelope calculation estimates that the time scale ($\tau$) required for a variation of sulphate concentration at the sea bottom to propagate down to about 16 cm is roughly $1.5$ year ($D \cdot \tau \simeq \ell_{SMTZ}^2$). The system is definitely responding on longer times than seasonal variations.

[revised manuscript text omitted]